# Cell-fate transition and determination analysis of mouse male germ cells throughout development

Jiexiang Zhao [1,8], Ping Lu [2,3,8], Cong Wan [1,8], Yaping Huang [1,8], Manman Cui[1], Xinyan Yang [1], Yuqiong Hu[2,3], Yi Zheng [1], Ji Dong [2,3], Mei Wang [1], Shu Zhang [2,3], Zhaoting Liu[1], Shuhui Bian [2,4], Xiaoman Wang [1], Rui Wang [2,4], Shaofang Ren [1], Dazhuang Wang [1], Zhaokai Yao [1], Gang Chang [5,8 ✉], Fuchou Tang [2,3,4 ✉] & Xiao-Yang Zhao [1,6,7 ✉]

Mammalian male germ cell development is a stepwise cell-fate transition process; however, the full-term developmental profile of male germ cells remains undefined. Here, by interrogating the high-precision transcriptome atlas of 11,598 cells covering 28 critical time-points, we demonstrate that cell-fate transition from mitotic to post-mitotic primordial germ cells is accompanied by transcriptome-scale reconfiguration and a transitional cell state. Notch signaling pathway is essential for initiating mitotic arrest and the maintenance of male germ cells' identities. Ablation of HELQ induces developmental arrest and abnormal transcriptome reprogramming of male germ cells, indicating the importance of cell cycle regulation for proper cell-fate transition. Finally, systematic human-mouse comparison reveals potential regulators whose deficiency contributed to human male infertility via mitotic arrest regulation. Collectively, our study provides an accurate and comprehensive transcriptome atlas of the male germline cycle and allows for an in-depth understanding of the cell-fate transition and determination underlying male germ cell development.

[1] State Key Laboratory of Organ Failure Research, Department of Developmental Biology, School of Basic Medical Sciences, Southern Medical University, 510515 Guangzhou, Guangdong, P. R. China. [2] Beijing Advanced Innovation Center for Genomics (ICG), School of Life Sciences, Peking University, 100871 Beijing, P. R. China. [3] Biomedical Pioneering Innovation Center, Ministry of Education Key Laboratory of Cell Proliferation and Differentiation, 100871 Beijing, P. R. China. [4] Peking-Tsinghua Center for Life Sciences, Academy for Advanced Interdisciplinary Studies, Peking University, 100871 Beijing, P. R. China. [5] Department of Biochemistry and Molecular Biology, Shenzhen University Health Science Center, 518060 Shenzhen, Guangdong, P. R. China. [6] Guangdong Key Laboratory of Construction and Detection in Tissue Engineering, Southern Medical University, 510515 Guangzhou, Guangdong, P. R. China. [7] Guangzhou Regenerative Medicine and Health Guangdong Laboratory (GRMH-GDL), 510700 Guangzhou, Guangdong, P. R. China. [8] These authors contributed equally: Jiexiang Zhao, Ping Lu, Cong Wan, Yaping Huang, Gang Chang. ✉email: changgang@szu.edu.cn; tangfuchou@pku.edu.cn; zhaoxiaoyang@smu.edu.cn

In species relying on sexual reproduction, germ cells are essential for the transmission of parental genetic information to subsequent generations. The full-term developmental trajectory of male germ cells in mammals entails primordial germ cell (PGC) specification, PGC migration, germ cell sex determination, gonadal PGC proliferation arrest, prospermatogonia-spermatogonia transition, and spermatogenesis[1]. The current understanding of mammalian germ cell development, of the germ cell niche, and of the critical roles of master regulators in controlling initiation and maintenance of germ cells emerged from pivotal studies in vivo and in vitro[2–9].

In mammals, the precursors of gametes are PGCs, which derive from the post-implantation epiblasts and gradually acquire the potential to further differentiate into prospermatogonia and finally mature into spermatozoa or oocytes. While PGCs have been well studied in many aspects such as PGC specification, epigenetic reprogramming, and the transition from PGCs to prospermatogonia[10,11], it is still poorly understood how stepwise cell-fate transitions orchestrate full-term male germ cell development. For instance, it is well known that the commitment of fetal male germ cells to mitotic arrest is crucial for subsequent development. However, the biological function and the regulatory mechanisms of mitotic arrest in male germline need further characterization albeit previous studies have accumulated much knowledge about this event[1,12–14]. On the other hand, a major bottleneck of human germ cell research is the difficulty in the continuous sampling of human germ cells, impeding our understanding of the full-term developmental trajectory and the underlying regulatory networks. Therefore, a single-cell transcriptome atlas to compare mouse and human male germ cell development programs is instructive to uncover evolutionarily conserved regulators. These insights are crucial to recapitulate the complete male gametogenesis process in vitro.

Previous genome-wide transcriptomic and epigenetic studies have offered valuable information about germ cell development[15–17]. With the help of single-cell RNA sequencing (scRNA-seq)[18], it is now possible to reveal the transcriptome signatures of rare cell populations with limited cell numbers. As a consequence, a cell census can be constructed for entire organs or even the whole organisms, providing novel avenues to decoding the molecular events underlying key developmental programs including germ cell development. Major breakthroughs have been achieved for mouse[19–23] and human germ cell development[24–30]. These studies illuminated specific stages of germ cell development but none of them monitored a full-term program of germ cell development in a single study. Considering the contribution of genetic background to the global gene-expression variations[31], and difficulties in integrating different sets of single-cell data due to the "batch effect" caused by different sequencing technologies, library preparation protocols, and data interpretation strategies across different platforms[32], it has profound significance to establish the full-term transcriptome atlas of male germ cell development at single-cell resolution.

In the present study, high-precision scRNA-seq was used to characterize the transcriptome atlas throughout the whole developmental trajectory of mouse male germ cells. We reconstructed the cell-fate transition landscapes of male germ cells and revealed the cell-type-specific transcriptome signatures as well as undefined cellular states. We further revealed the regulatory heterogeneity and key markers across male germ cell development. A transitional PGC stage was uncovered to be important for the cell-fate transition from mitotic to mitotic arrest PGCs. Moreover, we revealed that the importance of Notch signaling pathway in proper cell-fate transition from mitotic to mitotic arrest PGCs in mouse male germline development. And the biological function of mitotic to mitotic arrest transition of male PGCs was further verified using *Helq* knockout

mouse model, wherein DNA damage-associated cell-cycle regulation was found to be essential for prenatal development of male germ cells, as exemplified by the involvement of HELQ in controlling the normal development of transitional PGCs and post-arrest PGCs as well as proper transcriptome reconfiguration. Finally, we established the systematic comparison dataset of germ cell development between human and mouse and elicited human-mouse conserved gene regulatory networks. To enable public access to our data, we constructed the mouse male germ cell landscape website at https://tanglab.shinyapps.io/Mouse_Male_Germ_Cells/.

## Results

**A single-cell map of mouse male germ cell development.** To robustly trace germ cell development in mice, we crossed C57BL/6 females with double transgenic *Prdm1*-mVenus and *Dppa3*-ECFP (*Blimp1-mVenus* and *Stella-ECFP*, BVSC) males. The offspring of these animals were used to explore the transcriptome profiles and developmental program underlying male germ cell development. We performed high-precision scRNA-seq on 11,896 individual cells from 28 critical time-points across the whole male germ cell development process. For sample collection, early embryos or male gonads were collected spanning embryonic day (E) 6.5 and up to postnatal day (PND) 8 at one-day intervals, PND10-14 at two-day intervals, juvenile stages of PND30 and PND35 as well as adults (8–10 weeks). At least two independent biological replicates were performed for each time-point (Fig. 1a, Supplementary Table 1).

A total of 11,598 single-cell transcriptomes passed stringent quality control measures (Fig. 1b, c, Supplementary Fig. 1b and Supplementary Data 1), with an average of 9,413 detected genes and 302,772 mRNA molecules detected per cell (Supplementary Fig. 1a), indicating the STRT-seq method (based on SMART-seq2, see "Methods") that we used here was with high gene detection rate and sensitivity[33]. Subsequently, dimension reduction and clustering analysis were performed to group cells according to the global gene-expression profiles in an unbiased manner[34,35]. 10 major cell clusters, including one E6.5–E8.5 mix cell cluster containing specification PGCs, 7 germ cell clusters, and 2 somatic cell clusters, were identified (Fig. 1b–d and Supplementary Fig. 1b) (excluding contaminating blood cells and outliers). In particular, there were: (1) E6.5–E8.5 mix cell cluster (cluster 1) which contains specification PGCs expressing *Pou5f1* (*Oct4*), *Prdm1* (*Blimp1*), *Dppa3* (*Stella*), *Alpl*, and *T*[2,36–39] (Fig. 1d and Supplementary Fig. 1d, e); (2) migrating PGCs (cluster 2, E8.5–E10.5) expressing *Kit* and *Pecam1*[40]; (3) mitotic PGCs (cluster 3, E11.5–E13.5) expressing *Ddx4*, *Dazl*, and *Mki67*[41,42]; (4) mitotic arrest PGCs (cluster 4, E13.5–E16.5) expressing *Nanos2* and *Piwil4*[43,44]; (5) prospermatogonia (ProSPG, cluster 5, E16.5-PND4) expressing *Tspan8*[45]; (6) spermatogonia (SPG, cluster 6, PND0-14) expressing *Etv5* and *Zbtb16* (*Plzf*)[46–48]; (7) spermatocytes (SPC, cluster 7, PND5-Adult) expressing *Sycp3* and *Spo11*; (8) round spermatids (RS, cluster 8, PND30-Adult) expressing *Prm1* and *Tex36*[19,21]. To better understand the cell heterogeneity of specific cell clusters, unbiased cell clustering was further performed on clusters 5–6 and clusters 7–8, respectively. For ProSPG (cluster 5) and SPG (cluster 6), five subgroups corresponding to the five sequential developmental stages were further identified (Supplementary Fig. 1f, g). These included quiescent prospermatogonia (Q-ProSPG, expressing *Dnmt3l* and *Tspan8*), transitional prospermatogonia (T-ProSPG, expressing *Id4* and *Pecam1*), undifferentiated spermatogonia (undiff.ed SPG, expressing *Etv5* and *Gfra1*), differentiating spermatogonia (diff.ing SPG, expressing *Nanos3*, *Lin28a*, and *Upp1*) and differentiated spermatogonia (diff.ed SPG, expressing *Dmrtb1* and *Esx1*) (Supplementary Fig. 1f, g). Similarly, nine subgroups (excluding

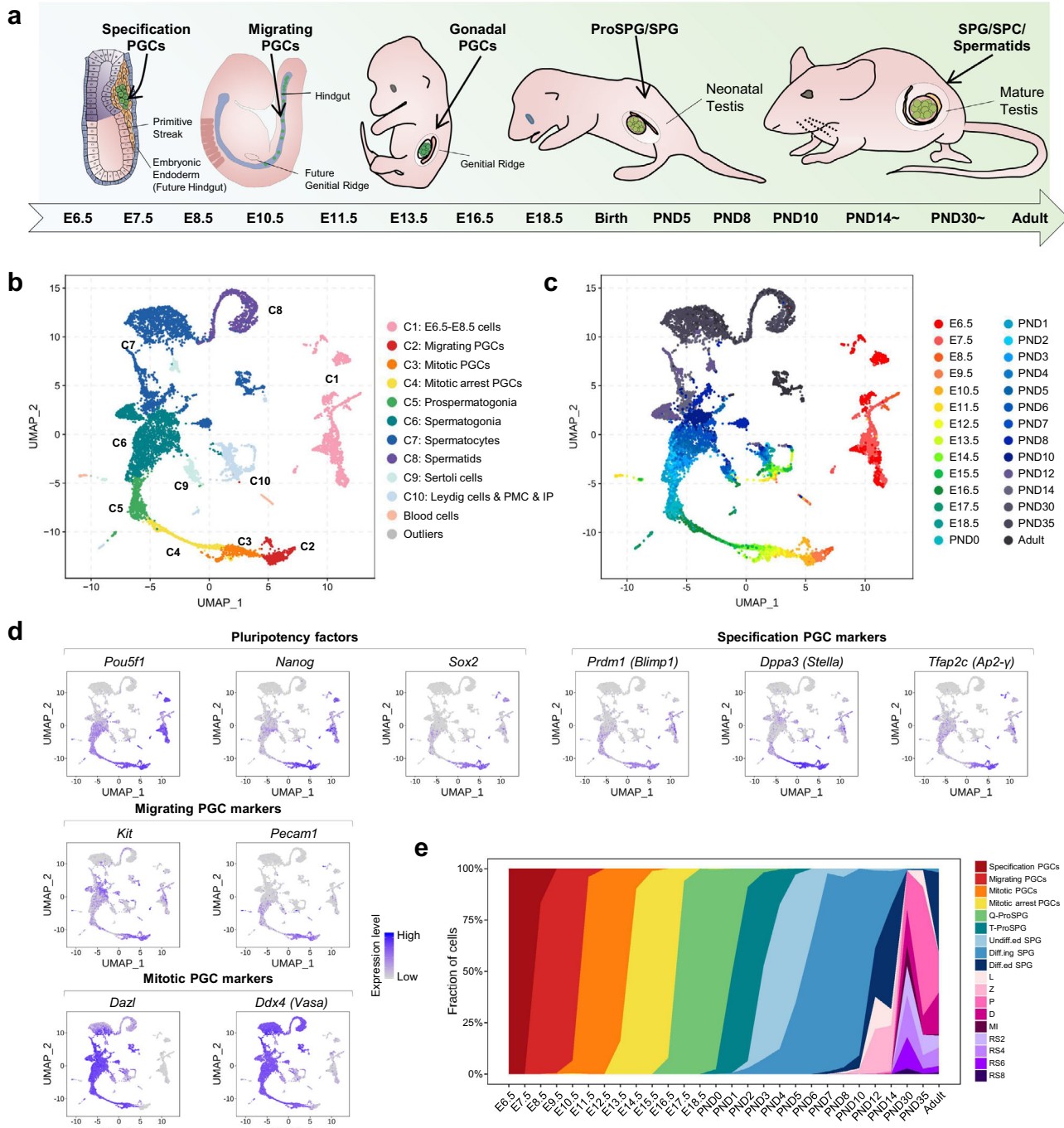

**Fig. 1 A single-cell map of mouse male germ cell development. a** Schematic overview of the sampled time-points across mouse male germ cell development. **b** UMAP (Uniform Manifold Approximation and Projection) plot of the 11,598 mouse germ cells and somatic cells included in this study. Cells are colored by indicated cell types. **c** UMAP plot of 11,598 mouse germ cells and somatic cells included in this study. Cells are colored by indicated developmental time-points. **d** UMAP plots are colored by expression levels for selected marker genes. Aliases of some genes are also shown. The color key from gray to blue indicates low to high expression levels. **e** Proportion of germ cell types across the sampled time-points. L, leptotene; Z, zygotene; P, pachytene; D, diplotene; MI, metaphase I; RS, round spermatids.

some residual SPG) were further defined from SPC (cluster 7) and RS (cluster 8) according to the dynamic expression of stage-specific markers (Supplementary Fig. 1h, i). The succession of all the germ cell types matched the progression along the densely sampled time-points. Also, four major somatic cell clusters were annotated as progenitor cells, peritubular myoid cells & interstitial progenitors (PMC & IP), Leydig cells and Sertoli cells based on previously determined cell-type markers (Supplementary Fig. 1j, k).

To examine the cell distribution across male germ cell development, the cellular composition of each sampled time-point was recorded based on the cell clustering results (Fig. 1e). Specifically, germ cells from E6.5 and E7.5 embryos consisted of only specification PGCs. Migrating PGCs colonized the gonads mainly at E10.5. At E13.5, mitotic arrest PGCs began to emerge and accordingly the proportion of mitotic PGCs declined, followed by sharply decreasing at E14.5. The conversion from Q-ProSPG to T-ProSPG occurred mainly from PND0-3. From

E16.5 to PND3, Q-ProSPG developed into T-ProSPG, while undiff.ed SPG appeared as early as PND2. From PND7 onwards, the commitment to spermatogonia differentiation and subsequent male meiosis occurred. This indicates that our dataset captures all of the known critical stages and cell-fate transition of mouse male germ cells, constituting the full-term single-cell transcriptome atlas of male germline cycle.

**Transcriptome signatures of mouse male germ cells.** To further dissect the transcriptome dynamics along the developmental trajectory of male germ cell development, global gene-expression patterns of differentially expressed genes (DEGs) were analyzed. A total of 23,204 DEGs (9,922 unique DEGs) were identified among the entire set of 8,581 germ cells in 18 major germ cell stages (Supplementary Fig. 2a and Supplementary Data 1). Gene ontology (GO) analysis on these DEGs demonstrated distinct biological processes in each germ cell type (Fig. 2a). Cell-cycle analysis revealed that a remarkable fraction of a majority of PGC and spermatogonia cell types were actively dividing (Fig. 2b, c). In contrast, mitotic arrest PGCs, Q-ProSPG, and spermatids were predominantly quiescent.

Cell-fate transition during male germ cell development involves the participation of a set of critical regulators; however, the expression dynamics of these genes in the context of the whole male germ cell development is hitherto unknown. To this end, the full-term transcriptional patterns of known essential regulators were analyzed across mouse male germ cell development (Fig. 2d; Supplementary Fig. 2d). As anticipated, core pluripotency factors were prominently enriched in the PGCs. According to the self-organizing maps (SOM), we found that each cell cluster enriched a different set of pluripotency factors (Supplementary Fig. 2c, Supplementary Data 1). And the well-defined spermatogonia factors were upregulated in ProSPG and SPG stages as expected. Notably, RS stage exhibited an upregulated expression of several pluripotency markers (*Klf2*, *Klf4*, *Gfra1*) and spermatogonia factors (*Etv5*, *Lhx1*) (Fig. 2d).

In search for markers with potential roles as drivers in mouse PGC development, we searched our data for genes specifically elevated at different developmental stages by DEG analysis. For example, it is observed that a prominent and cell-type-specific expression of *Elf3*, *Zic3*, *Rbm47*, *Irf1*, *Meios3*, *Porcn*, and *Slpi* in specification PGCs (Supplementary Fig. 3a, Supplementary Data 2). Among these candidate genes, *Irf1* was highly expressed in E7.5 PGCs; however, the emergence of IRF1 protein was first detected in migrating PGCs (BLIMP1-EGFP$^+$, E9.5), but not E7.5 PGCs (Supplementary Fig. 3b, c). This might be due to delayed translation of *Irf1* mRNA. Thus, our data indicate that IRF1 might participate in the regulation of PGC maintenance rather than PGC specification. Notably, we found that the population of migrating PGCs was highly heterogeneous and could be further divided into three states (Supplementary Fig. 3d). GO analysis showed that DEGs involved in "Wnt signaling pathway", "Ribosome", and "piRNA metabolic process" distinguished the three states of migrating PGCs, indicating that Wnt signaling pathway, translation, and piRNAs are sequentially involved in regulating the migration process of PGCs (Supplementary Fig. 3e, Supplementary Data 2).

With respect to the defined stage-specific markers identified in this study (discuss later) (Supplementary Fig. 2b), *Zic3* was highly expressed in both specification and migrating PGCs. Upregulated expression of *Hesx1* was detected in mitotic arrest PGCs. Subsequently, *Zxdc* and *Tie1* were primarily expressed in Q-ProSPG, while *Krt18* was expressed in Q-ProSPG, T-ProSPG as well as specification PGCs. Furthermore, prominent expression of *Hhex* was observed in both T-ProSPG and undiff.ed SPG. These stage-specific markers could potentially be used to further characterize the regulatory networks of male germ cells.

**Mitotic to mitotic arrest transition is companied by global transcriptome reconfiguration.** Upon arrival at genital ridges, PGCs undergo rapid proliferation and subsequently commit to male or female developmental trajectories[1]. In accordance with previous reports[49,50], we verified that almost all of the mouse male mitotic PGCs were indeed actively proliferating, but a part of mitotic arrest PGCs expressed cell-cycle-related genes highly, indicating that the heterogeneity and asynchronism in mouse male gonadal PGC development (Fig. 2b). Immunostaining analysis showed that the number of MKI67$^+$ PGCs decreased dramatically from 94.8% to 0.6% between stages E11.5 to E16.5, corroborating the final fate of male germ cells towards quiescence (Supplementary Fig. 4a, b). The mitotic arrest PGCs were then subdivided into two sub-populations based on the expression of cell cycle-related genes by unsupervised hierarchical clustering analysis (Fig. 3a). Coincided with the findings in human male fetal germ cells[29], the group with higher expression of cell-cycle genes were at the transitional stage of the two types of gonadal PGCs and were accordingly named as transitional PGCs. ROGUE-guided analysis[51] (see "Methods") proved that the purity score of mitotic arrest PGCs was increased from 0.26 to 0.50 when sub-dividing was performed. Besides, DEG analysis showed that the transitional PGCs exhibited its unique gene-expression signature (Fig. 3b). Based on the results using unsupervised dimension reduction and clustering analysis, the transitional PGCs were further confirmed to be an independent cell population (Supplementary Fig. 4c), which was comparable with the aforementioned results by unsupervised hierarchical clustering analysis.

Further analysis showed that transitional PGCs shared the transcriptome signatures of both mitotic PGCs and mitotic arrest PGCs; for instance, both of the early PGC's- marker gene *Nanog* and late PGC's- marker gene *Nanos2* were expressed in the transitional PGCs (Fig. 3c). The cellular characteristics of the transitional PGCs were analyzed by comparing with mitotic PGCs and mitotic arrest PGCs. We found that transitional PGC's signature genes were enriched in GO terms such as "Cell-cycle regulation", "Cell-cycle phase transition", and "APC/C-mediated degradation of cell-cycle proteins" (Supplementary Data 2), suggesting that cell cycle might be involved in the regulation of this stage of PGC development. Among the genes associated with these terms, *Wee1* (encoding WEE1) was selected as a specific maker of the transitional PGCs for further study (Supplementary Fig. 4d). By simultaneous immunofluorescence of MKI67 antibody and RNAscope assay of *Wee1* mRNA probe in situ, the MKI67$^+$ *Wee1*$^{High}$ cells were identified to be the transitional PGCs (Fig. 3d). In addition, the expression patterns of WEE1 protein and *Wee1* mRNA were generally consistent (Supplementary Fig. 4e–g). Based on these results, the developmental behavior of transitional PGCs could be traced by MKI67 antibody co-stained with *Wee1* mRNA probe or WEE1 antibody (discussed below). In addition, gene-expression dynamics of transitional PGCs along with mitotic PGCs and mitotic arrest PGCs were examined (Supplementary Data 2). GO analysis showed that the downregulated genes from mitotic PGCs to transitional PGCs were markedly associated with "Glucose catabolic process", while genes downregulated from transitional PGCs to mitotic arrest PGCs were related to "Cell cycle" and "DNA replication". In contrast, male gamete generation-related GO terms were enriched in the upregulated genes (Supplementary Data 2). Furthermore, pluripotency factors underwent dramatic expression changes (Supplementary Fig. 4h). Based on these data, transitional PGCs represented the transitional cellular state in the conversion from mitotic PGCs to mitotic arrest PGCs, and its activity was highly associated with cell-cycle regulation. *Ascl2*, *Hesx1*, *Hes1*, *Tgif1*, and *Sp5* were identified to be the top transcription

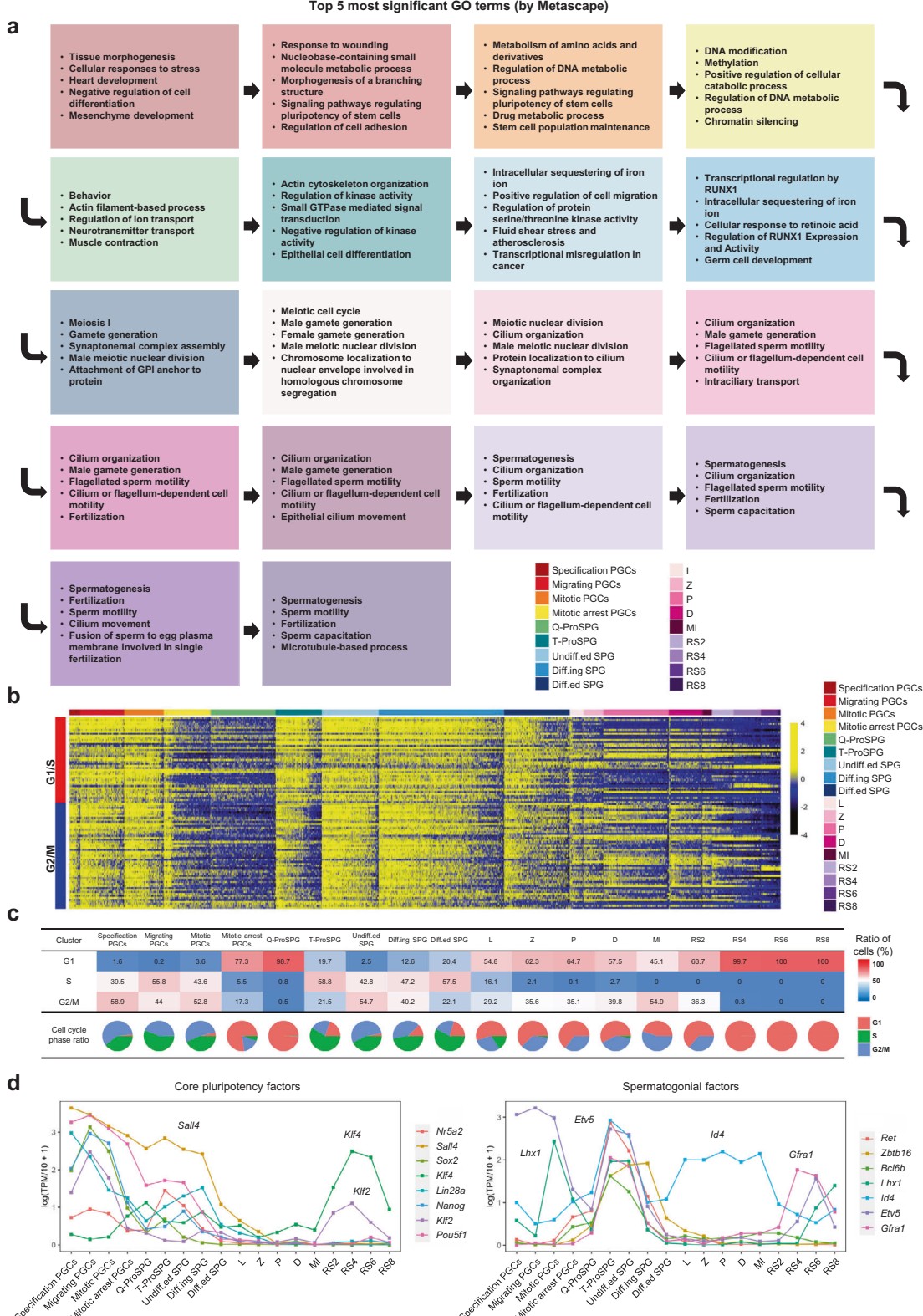

**Fig. 2 Transcriptome signatures of mouse male germ cells. a** Top 5 most significant GO terms and *P* values (-LogP) of 18 germ cell clusters are indicated by distinct colors, respectively. **b** Heatmap of 97 cell-cycle-related genes in mouse male germ cells. The color key from black to yellow indicates low to high expression levels. L, leptotene; Z, zygotene; P, pachytene; D, diplotene; MI, metaphase I; RS, round spermatids. **c** Proportion germ cells in one of three cell-cycle phase. **d** Line plots showing the relative expression levels (log(TPM/10+1)) of previously known core pluripotency factors, and spermatogonial factors in each cell cluster.

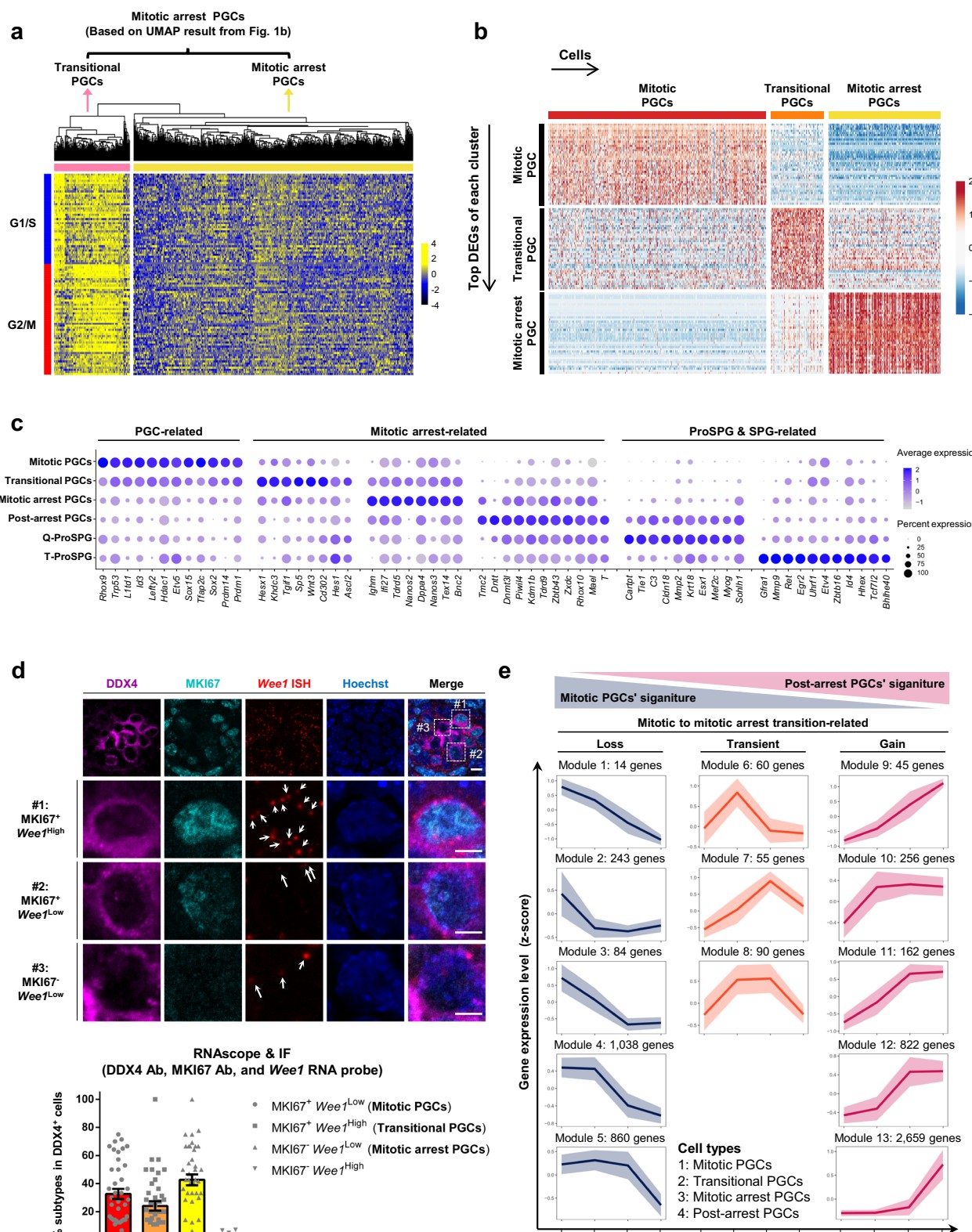

factors specific for the transitional PGCs (Supplementary Fig. 4i, Supplementary Data 2).

To comprehensively understand how cell-fate evolved from PGCs to spermatogonia, dimension reduction and clustering analysis were further performed and an overlapping sub-

population between mitotic arrest PGCs and ProSPG was identified, which was termed the post-arrest PGCs (Supplementary Fig. 5a). GO analysis showed that genes specifically expressed in post-arrest PGCs were associated with "DNA methylation involved in gamete generation", "Cilium assembly", "Cellular

**Fig. 3 Mitotic to mitotic arrest transition is companied by global transcriptome reconfiguration. a** Heatmap showing cell-cycle-related genes based hierarchical clustering result of the mitotic arrest PGCs identified from Fig. 1b. The transitional PGCs are clearly subdivided as an independent cell cluster. **b** Heatmap of the top DEGs of each stage (identified among the mitotic PGCs, transitional PGCs and mitotic arrest PGCs). The color key from blue to red indicates low to high expression levels. **c** Dotplot showing the expression patterns of gene sets in the mitotic PGCs, transitional PGCs, mitotic arrest PGCs, post-arrest PGCs, Q-ProSPG, and T-ProSPG. Dot size indicates the fraction of cells with detectable expression for a given marker gene and the color key indicates average gene-expression levels in each cell type. **d** RNAscope of *Wee1* co-stained with immunofluorescence of DDX4 and MKI67 antibodies in E14.5 mouse male gonadal sections. Arrows indicate the dots of the probe signals, each representing one copy of the *Wee1* mRNA. Scale bar, 10 μm. Detailed images of the indicated germ cells are also shown. Scale bar, 5 μm. Images are obtained using ZEISS LSM880 confocal microscope under a C-Apochromat 63×/1.20 W korr M27 objective lens with 2× scan zoom. Relative proportions of mitotic- (MKI67$^+$ *Wee1*$^{Low}$), transitional- (MKI67$^+$ *Wee1*$^{High}$), and mitotic arrest PGCs (MKI67$^-$ *Wee1*$^{Low}$) in the WT mouse male gonads at E14.5 are also shown below. Mean ± SEM, $n = 457$ cells examined over four biologically independent experiments. **e** Clustering analysis of dynamic gene expression during mitotic to mitotic arrest transition. Thirteen modules of genes form three distinct categories according to the expression patterns. Gene number per cluster was followed by the module number. Mean of scaled gene-expression level of each module (solid line) was shown with 95% confidence interval (shadow).

process involved in reproduction in multicellular organism", and "G1 to G0 transition" (Supplementary Fig. 5b, Supplementary Data 2), demonstrating that the cell-cycle transition of post-arrest PGCs was completed; instead, it was undergoing epigenetic reprogramming for the formation of spermatogonial stem cells (SSCs) pool (Supplementary Fig. 5c, Supplementary Data 2). Besides, the cell identity of post-arrest PGCs was also confirmed by the expression of known and the identified marker genes (Fig. 3c). Accordingly, the proportion of mitotic arrest PGCs increased from 4% to 85%, along with decreased population of mitotic PGCs, which was from 84% to 1% between E13.5 and E15.5 (Supplementary Fig. 5d).

To delineate the transcriptome reconfiguration from mitotic PGCs to post-arrest PGCs, gene-expression dynamics of the four types of PGCs along with their former- and later-stage cells were examined. Thirteen modules of DEGs with distinct expression patterns were classified into three categories among these four consecutive types of PGCs (Fig. 3e, Supplementary Fig. 5e, and Supplementary Data 2). The genes in modules 1–5 (Category 1), which exhibited down-regulated trends, were associated with "Cell cycle" and "mRNA metabolic process". The genes in modules 6–8 (Category 2) were transiently upregulated in transitional PGCs, mitotic arrest PGCs or both, which were enriched in GO terms of "Regulation of cell-cycle process", "Regulation of reproductive process", and "Gamete generation". While genes in modules 9–13 (Category 3) were gradually upregulated from mitotic PGCs to post-arrest PGCs. Interestingly, there were 2,659 genes (module 13) specifically upregulated from mitotic arrest PGCs to post-arrest PGCs, and these genes were enriched in GO terms associated with intercellular communication and cell movement. In contrast, genes in modules 9–12 were involved in the GO terms of "piRNA metabolic process" and "DNA methylation involved in gamete generation", which were essential for male germ cell development[44,52–55] (Supplementary Data 2). Based on these results, the global transcriptome reconfiguration of these gene modules might be important for the cell-fate transition of prenatal male germ cells.

Taken together, male germ cells undergo a comprehensive transcriptome reconfiguration during the mitotic to mitotic arrest transition, wherein transitional PGCs might be essential for the proper progression of this process through cell-cycle regulation.

**Notch signaling pathway regulates the mitotic arrest of male PGCs.** Exit from mitosis is an indispensable developmental process of late-stage prenatal male germ cells[56,57]. As aforementioned, the transitional PGCs were identified as an undefined sub-population before mitotic PGCs entered into mitotic arrest status, implying its essential roles in male germ cell-cycle regulation. Among the top transcription factors enriched in transitional PGCs (Fig. 3c, Supplementary Fig. 4i), *Hes1* (encoding HES1), a canonical downstream effector of the Notch signaling pathway,

has caught our attention for its crucial roles in stem cell quiescence, cell-cycle control and maintenance through direct DNA-binding activity[58,59]. The previous studies illustrated the roles of Sertoli cell-Notch signaling pathway in regulating male germ cell fate[60,61]. Given that *Hes1* was highly expressed in transitional PGCs, we speculated that Notch signaling pathway might also play functions in mitotic- to mitotic arrest transition of prenatal male germ cells. Using immunostaining, a gradual increase of HES1$^{Bright}$/HES1$^{Dim}$ cell ratio was observed from E13.5 to E16.5, which followed the activation of NOTCH1 around E12.5 (Fig. 4a–d). Further immunostaining results of NOTCH2, NOTCH3, and NOTCH4 showed that the NOTCH1/3-HES1 axis was activated in mouse male prenatal germ cells from E13.5 to E15.5, while NOTCH2/4 might not involve in the process (Supplementary Fig. 6a–d). The quite strict correlation of mitotic arrest progression and the activation timing of Notch signaling pathway implied that the Notch signaling pathway might play important roles in PGC development.

To test if the Notch signaling pathway was involved in regulating PGC mitotic arrest, we performed γ-Secretase inhibitor −DAPT (N-[N-(3,5-difluorophenacetyl)-L-alanyl]-S-phenylglycine t-butyl ester) treatment experiment, which can effectively block the Notch signaling pathway. Pregnant mice were intraperitoneally injected with DAPT twice at E13.5 and E14.5, and the embryos were dissected and genital ridges were analyzed at E15.5 onward (Fig. 4e). We found a dramatic decrease of NOTCH1 and HES1 signal in DAPT-treatment groups compared with the mock controls (Fig. 4f, g). Notably, Notch signaling pathway blocking impaired the mitotic arrest progression of PGC development, wherein the number of MKI67$^+$ PGCs was significantly increased compared with the mock controls (~20% versus ~3%, Fig. 4h). These data indicated that the Notch signaling pathway could promote mitotic arrest through cell-cycle regulation. To further confirm this finding, another Notch signaling pathway inhibitor-LY411575 (a chemical γ-Secretase inhibitor, N-[(1S)-2-[[(7S)-6,7-dihydro-5-methyl-6-oxo-5H-dibenz[b,d]azepin-7-yl]amino]-1-methyl-2-oxoethyl]-3,5-difluoro-αS-hydroxy-benzeneacetamide) was used to block Notch signaling pathway and we evaluated its function accordingly (Supplementary Fig. 6g-l). Our results showed that the treatment by LY411575 could also result in a significant decrease of NOTCH1 and HES1 in mouse male germ cells, wherein the number of MKI67$^+$ PGCs was significantly increased compared with the mock controls (Supplementary Fig. 6i, l). Short-term treatment of the in vitro cultured mouse male PGCs by DAPT or LY411575 could also lead to the decreased expression of *Hes1* (Supplementary Fig. 6e, f). These results demonstrated that the Notch signaling pathway was indeed activated in mouse male gonadal germ cells, and played critical roles in the cell-fate transition of PGCs.

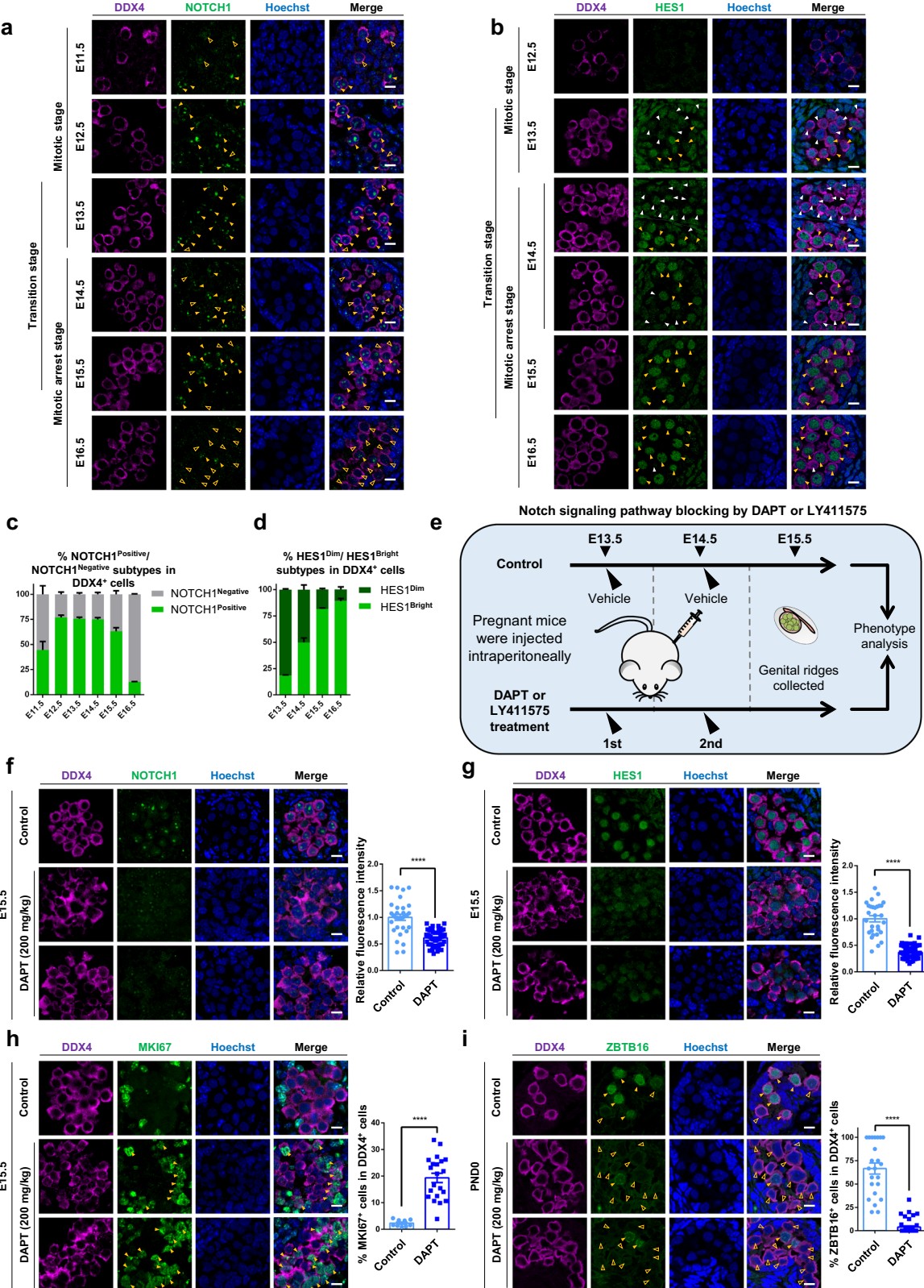

Further experiment demonstrated that in vivo Notch signaling pathway blocking led to a lower proportion of ZBTB16$^+$ germ cells compared with the mock controls at PND0 (Fig. 4i), suggesting that abnormal cell cycle switch might further impair the postnatal cell-fate transition and eventually disturb SSC pool development. Based on these results, the Notch signaling pathway

was uncovered to be crucial for the mitotic arrest initiation and the subsequent male germ cell fate transition.

**Proper mitotic to mitotic arrest transition is crucial for the prenatal development of male germ cells.** By mining the dynamic transcriptome database delineated from mitotic PGCs to

**Fig. 4 Notch signaling pathway regulates the mitotic arrest of male PGCs. a** Immunofluorescence of NOTCH1 co-stained with DDX4 in E11.5–E16.5 mouse male gonads. Hollow and solid yellow arrowheads indicate NOTCH1[Negative] and NOTCH1[Positive] subtypes in DDX4[+] cells, respectively. Scale bar, 10 µm. **b** Immunofluorescence of HES1 co-stained with DDX4 in E11.5–E16.5 mouse male gonads. White and yellow arrowheads indicate HES1[Dim] and HES1[Bright] subtypes in DDX4[+] cells, respectively. Scale bar, 10 µm. **c** Proportion of NOTCH1[Negative] and NOTCH1[Positive] subtypes in DDX4[+] cells in E11.5–E16.5 mouse male gonads. Mean ± SEM, $n$ = at least 2 biologically independent samples for each time-point. **d** Proportion of HES1[Dim] and HES1[Bright] subtypes in DDX4[+] cells in E11.5–E16.5 mouse male gonads. Mean ± SEM, $n$ = at least three biologically independent samples for each time-point. **e** Schematic showing the workflow of treatment of γ-Secretase inhibitor–DAPT or LY411575 for Notch signaling pathway blocking. **f** Immunofluorescence of NOTCH1 co-stained with DDX4 and the quantification of relative fluorescence intensity in DAPT-treatment and control mouse male gonads at E15.5. Scale bar, 10 µm. Mean ± SEM, $n$ = 3 per group, **** $P < 0.0001$, $P = 9.5E-13$, unpaired two-tailed $t$ test. **g** Immunofluorescence of HES1 co-stained with DDX4 and the quantification of relative fluorescence intensity in DAPT-treatment and control mouse male gonads at E15.5. Scale bar, 10 µm. Mean ± SEM, $n$ = 3 per group, **** $P < 0.0001$, $P = 4.0E-25$, unpaired two-tailed $t$ test. **h** Immunofluorescence of MKI67 co-stained with DDX4 and proportion of MKI67[+] cells in DDX4[+] cells in DAPT-treatment and control mouse male gonads at E15.5. Solid yellow arrowheads indicate MKI67[Positive] subtypes in DDX4[+] cells. Scale bar, 10 µm. Mean ± SEM, $n$ = 3 per group, **** $P < 0.0001$, $P = 8.8E-07$, unpaired two-tailed $t$ test. **i** Immunofluorescence of ProSPG and SPG marker ZBTB16 co-stained with DDX4 and proportion of ZBTB16[+] cells in DDX4[+] cells in DAPT-treatment and control mouse male gonads at PND0. Hollow and solid yellow arrowheads indicate ZBTB16[Negative] and ZBTB16[Positive] subtypes in DDX4[+] cells, respectively. Scale bar, 10 µm. Mean ± SEM, $n$ = 3 per group, **** $P < 0.0001$, $P = 1.2E-18$, unpaired two-tailed $t$ test.

post-arrest PGCs (Supplementary Data 2), we found a set of genes were highly enriched in GO terms such as "Cell cycle", "Cellular response to DNA damage stimulus", "DNA repair", "DNA replication", and "Cell-cycle phase transition", suggesting that the cell-fate transition of PGCs might be connected to these biological processes. During the mitotic to mitotic arrest transition, we observed a high expression of $Helq$, which was one of "DNA repair" gene and encoded a DNA helicase (Supplementary Fig. 7a). Previous works have emphasized the critical role of the DNA helicase HELQ in male reproductive system, wherein hypogonadism and germ cell reduction were observed in the postnatal $Helq$ mutant male mice[62,63]; however, the fundamental questions about when and how HELQ regulated germ cell development prenatally remained unexplored. To this end, $Helq^{-/-}$ mice were generated in the context of the BVSC double transgenic reporter (Supplementary Fig. 7b, c). We found that $Helq^{-/-}$ male mice exhibited a clear decrease of germ cell number than the wild-type controls from E13.5 onward (Fig. 5a, Supplementary Fig. 7d). Immunostaining of cleaved PARP1 (cPARP1) showed that apoptosis was upregulated in $Helq^{-/-}$ PGCs relative to the wild-type controls (Fig. 5b, Supplementary Fig. 7e). Notably, the germ cell attrition in $Helq^{-/-}$ male gonads was mainly coincident with the emergence of transitional PGCs at ~E14.5, suggesting that HELQ might potentially participate in the regulation of cell-fate transition from the mitotic to the mitotic arrest stage.

Dissecting the temporal and mechanistic roles of stably expressed genes like $Helq$ (Supplementary Fig. 7a) in male germ cells is challenging for both in vitro germ cell induction and bulk sequencing-based study of in vivo isolated samples. To circumvent these difficulties, we used our atlas as a reference for the analysis of germ cells isolated from $Helq$ mutant male mice to elucidate whether HELQ was critical for the cell-fate transition of prenatal male germ cells. scRNA-seq was performed for 724 individual germ cells isolated from $Helq^{-/-}$ gonads from E11.5 to E18.5 (Fig. 5c, Supplementary Fig. 8a and Supplementary Data 3). Interestingly, unbiased cell cluster and developmental trajectory analysis showed a developmental arrest of $Helq^{-/-}$ PGCs during E15.5 and E18.5. Of note, the proportion of transitional PGCs increased dramatically in $Helq^{-/-}$ PGCs (45% in $Helq^{-/-}$ versus 12% in wild-type at E13.5; 41% in $Helq^{-/-}$ versus 7.0% in wild-type at E15.5); by contrast, we did not detect post-arrest PGCs in $Helq^{-/-}$ mice at E15.5 due to developmental arrest (0.0% in $Helq^{-/-}$ versus 7.6% in wild-type at E15.5) (Fig. 5d). Further validation demonstrated that $Helq^{-/-}$ PGCs exhibited aberrant MKI67 expression

patterns during E13.5–E15.5 (Supplementary Fig. 7f, g), indicating the aberrant mitotic arrest transition in $Helq^{-/-}$ PGCs.

To further explore how HELQ regulates the mitotic arrest transition, we analyzed the cell-cycle status based on CellCycleScoring of scRNA-seq. The results showed that there were more cells in the S phase of cell cycle in E11.5–E15.5 $Helq^{-/-}$ mitotic- or transitional PGCs (Supplementary Fig. 8b), which could explain the higher percentage of MKI67[+] cells in $Helq^{-/-}$ mice. In support of this, $Helq^{-/-}$ mESCs also exhibited a higher S population (47.84%) than the wild-type control (41.00%) (Supplementary Fig. 8c). Based on the model established previously, it has been proposed that HELQ was involved in DNA replication and facilitated the loading of factors required for DNA damage processing[62–64]. As expected, ATR (ataxia-telangiectasia and Rad3-related) was hyperactivated in $Helq^{-/-}$ mice (Fig. 5e, h), consistent with the fact that ATR couples DNA replication with mitosis and preserves genome integrity by enforcing the S/G2 checkpoint. Then, immunostaining results showed that there was a remarkable increase of p-CHK1 (phospho-checkpoint kinase 1) positive cells in the $Helq^{-/-}$ group compared with the wild-type control (Fig. 5f, i, Supplementary Fig. 8d). p-CHK1 acts as the downstream effector of p-ATR in cell-cycle checkpoint that could inhibit CDKs (cyclin-dependent kinases)[65,66]. Next, we checked the behavior of the transitional PGCs by co-staining of MKI67 and WEE1 (mentioned before, Fig. 3d, Supplementary Fig. 4d–g). The MKI67 and WEE1 antibodies co-staining results indicated that the MKI67-positive and WEE1 lowly expressed cells (MKI67[+] WEE1[Low]), the MKI67-positive and WEE1 highly expressed cells (MKI67[+] WEE1[High]), and the MKI67-negative and WEE1 lowly expressed cells (MKI67[−] WEE1[Low]) should refer to as the mitotic PGCs, transitional PGCs, and mitotic arrest PGCs, respectively (Fig. 5g). As a result, there was a significant increase of transitional PGCs (MKI67[+] WEE1[High]) at E14.5 when $Helq$ was knocked out (Fig. 5g, j). Moreover, we observed a significant increase of transitional PGCs in the $Helq^{-/-}$ group at E15.5 (Supplementary Fig. 8e), which was rare in the wild-type control. Based on these data, HELQ depletion could disrupt the mitotic to mitotic arrest transition of PGCs, wherein HELQ deficiency might hyperactivate the p-ATR/p-CHK1 regulatory axis, which then prolonged the cell-cycle checkpoint and resulted in PGC development arrest.

To understand the molecular consequences of HELQ deficiency, we analyzed the transcriptome of $Helq^{-/-}$ PGCs and found that 194, 519, and 438 genes changed their expression pattern in mitotic PGCs, transitional PGCs, and mitotic arrest

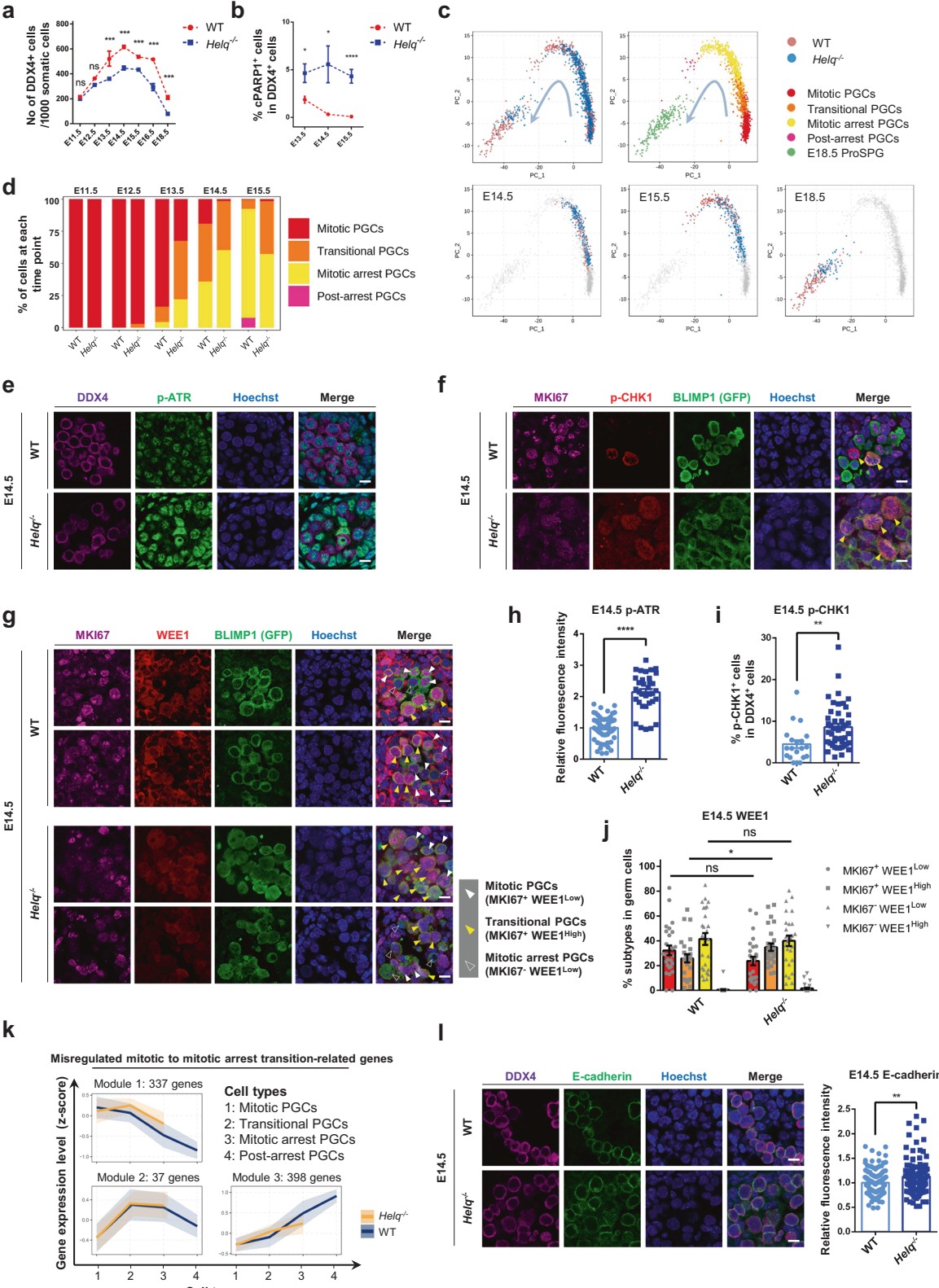

PGCs, respectively (Supplementary Fig. 9a, Supplementary Data 3). As expected, genes involved in cell-cycle checkpoint-related processes were upregulated in $Helq^{-/-}$ PGCs (Supplementary Data 3), confirming that HELQ might act on cell-cycle checkpoint through maintaining DNA integrity during the mitotic arrest process in male PGCs. In contrast, the genes downregulated in $Helq^{-/-}$ mitotic arrest germ cells were mostly

associated with male germ cell development such as "Cilium organization" and "Flagellated sperm motility" (Supplementary Fig. 9b, Supplementary Data 3). In accordance with the abnormal developmental trajectory of PGCs in $Helq^{-/-}$ mice, we found that the expression dynamics of the three categories of genes, which were necessary for the proper cell-fate transition from mitotic to post-arrest PGCs, were dysregulated (Fig. 5k). Among them,

**Fig. 5 Proper mitotic to mitotic arrest transition is crucial for the prenatal development of male germ cells. a** Numbers of DDX4$^+$ cells from E11.5 to E16.5, and E18.5 WT versus $Helq^{-/-}$ mouse male gonads. Mean ± SEM, $n = 4$ per time-point, ns, not significant, *** $P < 0.001$, E11.5 ($P = 0.5731$), E12.5 ($P = 0.0706$), E13.5 ($P = 0.0004$), E14.5 ($P = 0.0001$), E15.5 ($P = 0.0004$), E16.5 ($P = 0.0002$), E18.5 ($P = 0.0002$), unpaired two-tailed $t$ test. **b** Scatter diagram showing the proportions of cPARP1$^+$ cells in DDX4$^+$ cells from E13.5 to E15.5 in the WT- and $Helq^{-/-}$ mouse male gonads. Mean ± SEM, n = at least 2 per time-point, * $P < 0.05$, **** $P < 0.0001$, E13.5 ($P = 0.0176$), E14.5 ($P = 0.0358$), E15.5 ($P = 7.7E−05$), unpaired two-tailed $t$ test. **c** PCA (Principal component analysis) plots showing the developmental trajectory of 985 WT- and 724 $Helq^{-/-}$ mouse male germ cells sampled between E14.5 and E18.5. The same PCA is also plotted, highlighting only cells from each stage (cell numbers: $n = 187$, WT; $n = 191$, $Helq^{-/-}$ for E14.5; $n = 185$, WT; $n = 187$, $Helq^{-/-}$ for E15.5; $n = 140$, WT; $n = 72$, $Helq^{-/-}$ for E18.5, also see Supplementary Fig. 8a). **d** Relative proportions of mitotic-, transitional-, mitotic arrest-, and post-arrest PGCs from E11.5 to E15.5 in the WT- versus $Helq^{-/-}$ mouse male gonads. **e** Immunofluorescence of p-ATR co-stained with DDX4. Scale bar, 10 μm. **f** Immunofluorescence of p-CHK1 co-stained with MKI67 and BLIMP1 (GFP) in the WT- and $Helq^{-/-}$ mouse male gonads at E14.5. Arrowheads indicate p-CHK1$^+$ subtypes in MKI67$^+$ germ cells. Scale bar, 10 μm. **g** Immunofluorescence of WEE1 co-stained with MKI67 and BLIMP1 (GFP) in the WT- and $Helq^{-/-}$ mouse male gonads at E14.5. Scale bar, 10 μm. **h** The quantification of relative fluorescence intensity of p-ATR in the WT- and $Helq^{-/-}$ mouse male germ cells at E14.5. Mean ± SEM, $n = 4$ per group, **** $P < 0.0001$, $P = 9.1E-19$, unpaired two-tailed $t$ test. **i** Relative proportions of of p-CHK1$^+$ germ cells in the WT- and $Helq^{-/-}$ mouse male germ cells at E14.5. Mean ± SEM, $n = 4$ per group, ** $P < 0.01$, $P = 0.0029$, unpaired two-tailed $t$ test. **j** Relative proportions of mitotic- (MKI67$^+$ WEE1$^{Low}$), transitional- (MKI67$^+$ WEE1$^{High}$), and mitotic arrest PGCs (MKI67$^-$ WEE1$^{Low}$) in the WT- versus $Helq^{-/-}$ mouse male gonads at E14.5. Mean ± SEM, $n = 4$ per group, ns, not significant, * $P < 0.05$, MKI67$^+$ WEE1$^{Low}$ ($P = 0.1113$), MKI67$^+$ WEE1$^{High}$ ($P = 0.0488$), MKI67$^-$ WEE1$^{Low}$ ($P = 0.8135$), unpaired two-tailed $t$ test. **k** Clustering analysis of the misexpression of mitotic to mitotic arrest transition-related genes expression. Three distinct modules according to the expression patterns were shown. Misregulated gene number per cluster was followed by the module number. Mean of scaled gene-expression level of each module (solid line) was shown with 95% confidence interval (shadow). **l** Immunofluorescence of E-cadherin co-stained with DDX4 and the quantification of relative fluorescence intensity in the WT- and $Helq^{-/-}$ mouse male gonads at E14.5. Scale bar, 10 μm. Mean ± SEM, $n = 4$ per group, ** $P < 0.01$, $P = 0.0029$, unpaired two-tailed $t$ test.

several genes have been reported to be important for early- or late-stage fetal germ cell development, such as *Prdm1*, *L1td1*, *Rhox10*, and *Dnmt3l*[67–70] (Supplementary Fig. 9c). In the DEG list when *Helq* was knocked out, we found $Helq^{-/-}$ germ cells highly expressed *Cdh1* (encoding E-cadherin) (Supplementary Fig. 9c), which was gradually downregulated in the wild-type control (involved in the module of "Loss" in Fig. 3e and Supplementary Data 2; Supplementary Fig. 9d), and this downregulation trend was verified by immunostaining of E-cadherin (Supplementary Fig. 9e). Immunostaining results further showed the higher expression of E-cadherin protein in $Helq^{-/-}$ germ cells compared with the wild-type control (Fig. 5l). Moreover, GO analysis showed that *Cdh1* and other abnormally expressed genes were enriched in the processes such as "Apoptotic cleavage of cellular proteins", "Apoptosis", "Tight junction assembly", "Hippo signaling pathway", "Pathways in cancer", and "Negative regulation of cell differentiation" (Supplementary Data 3), which were highly related to the phenotypes of $Helq^{-/-}$ germ cells as we observed. These results suggest that *Helq* knockout might also interrupt germ cell development by maintaining the high expression level of E-cadherin (encoded by *Cdh1*).

Taken together, these results proved that HELQ deficiency could lead to the deceleration of PGC development at the transitional stage. Our studies based on HELQ ablation demonstrated that the proper cell-cycle switch was crucial for the cell-fate transition from mitotic to post-arrest PGCs (Supplementary Fig. 9f).

**Formation of the spermatogonial stem cell pool.** In mammals, prospermatogonia, also called prespermatogonia or testicular gonocytes, are the fetal/neonatal precursors of SSCs. SSCs are the foundation for life-long production of spermatozoa. To dissect the transcriptional regulation driving the cell-fate transition from ProSPG to spermatogonia, further unbiased clustering and cell-type assignments were performed. ProSPG, and spermatogonia can be clearly divided into five clusters, including Q-ProSPG, T-ProSPG, undiff.ed SPG, diff.ing SPG, and diff.ed SPG characterized by well-defined markers (Supplementary Fig. 1f, g). Of note, the identified T-ProSPG exhibited the transcriptome signatures of both Q-ProSPG and undiff.ed SPG, indicating that T-ProSPG might act as the intermediate cell state during this process. In support of this finding, GO analysis revealed that the DEGs in T-ProSPG were associated

with terms such as "cell movement and cell-cell junction assemble", consistent with their roles in the establishment of the SSC pool (Supplementary Fig. 10a, Supplementary Data 4). In addition, T-ProSPG could be further divided into two sub-populations mainly based on the characteristics of cell proliferation. The population with lower expression of proliferation genes was intermediate prosper-matogonia (I-ProSPG), consistent with a previous study[23]. The developmental trajectory based on our data could be also recapitulated by integrating other perinatal germ cells scRNA-seq datasets[71,72] (Supplementary Fig. 11 and 12).

Decoding the regulatory network leading to SSCs is of critical relevance for the potential utilization of these cells in infertility treatment. Therefore, we sought to find regulators crucial for the PGC-ProSPG-SSC transition and the maintenance of the SSC pool. GO analysis showed that cell-cycle-related terms were significantly associated with this process. Notably, undiff.ed SPG was highly enriched with "Positive regulation of canonical Wnt signaling pathway", suggesting potential functions of Wnt signaling pathway for this population (Supplementary Fig. 10b, Supplementary Data 4). Moreover, a critical set of markers of these two processes were identified for ProSPGs (*Zxdc*, *Tie1*, *Myog*, *Mef2c*, and *Krt18*) and SPGs (*Tmc6*, *Hhex*, *Nefm*, and *Tagln2*) (Fig. 6a, Supplementary Data 4). The expression patterns of several markers were further evaluated by immunostaining. In particular, ZXDC was shown to be highly expressed in ProSPG, but its expression level began to decline as ProSPG developed into SSCs (Fig. 6b, Supplementary Fig. 10c). Similarly, a strong expression of KRT18 protein was observed in ProSPG from E16.5 to PND1, but began to decline from PND3 onward (Fig. 6c, Supplementary Fig. 10d). Similarly, TIE1 protein was also detected at E18.5 (Fig. 6d). Prominent expression of HHEX and NEFM was identified in PND5 SSCs, implying a potential function in the establishment of the SSC pool (Fig. 6e, f, Supplementary Fig. 10e, f).

To test whether HHEX had a causal role in SSC development, *Hhex* expression was depleted in mSSC cells using short hairpin RNAs (shRNAs) (Fig. 6g). Knockdown (KD) efficiencies were validated by q-PCR and Western blot (Supplementary Fig. 10g, h). Knockdown of *Hhex* impaired the mSSC clone formation and decreased cell proliferation (Fig. 6h, i). Genome-wide RNA-seq analysis revealed that *Hhex* knockdown cells comprised a distinctive gene-expression profile with 1,028 upregulated and

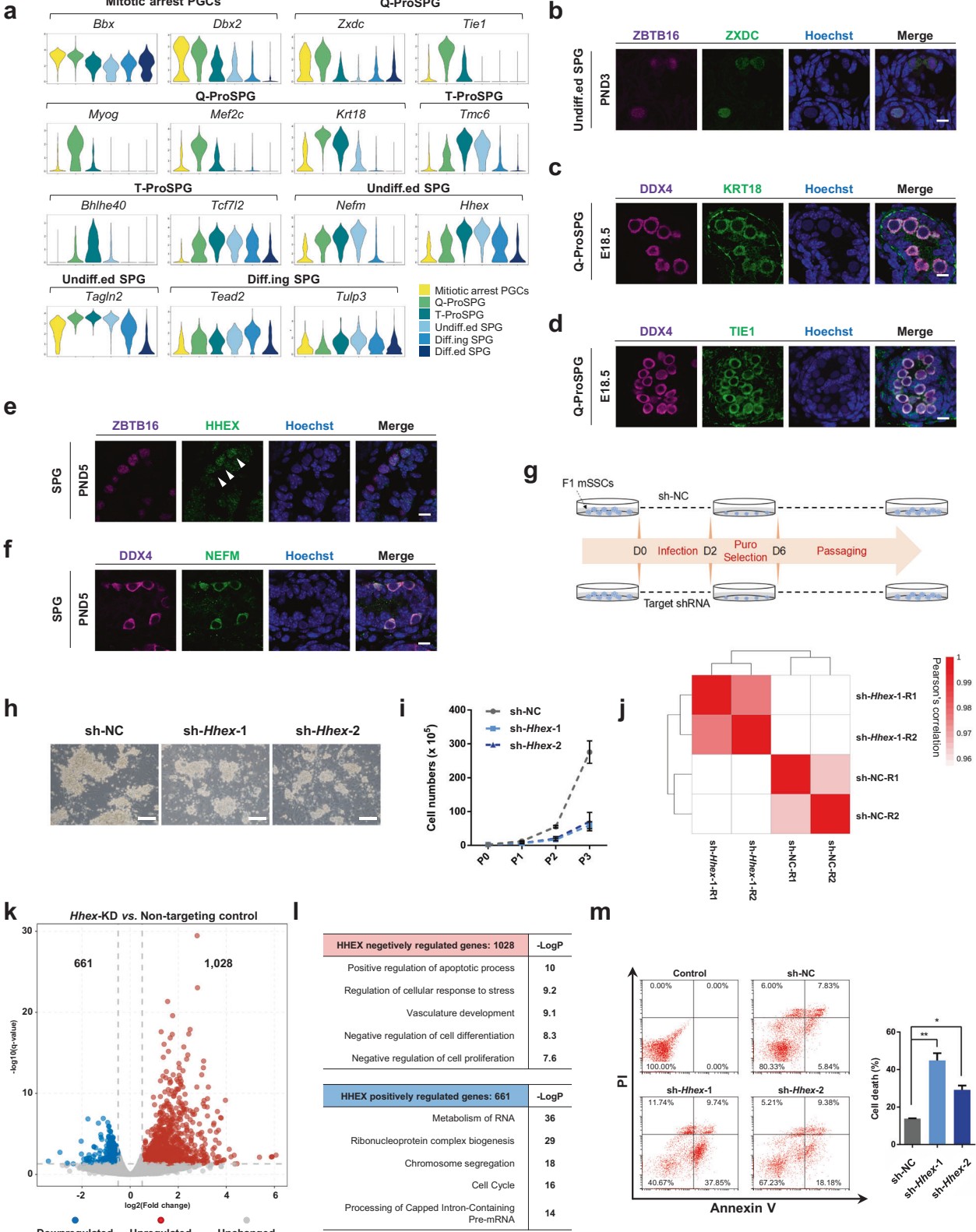

661 downregulated genes (Fig. 6k, Supplementary Data 5). GO analysis revealed that HHEX positively regulated genes were associated with "Metabolism of RNA", "Chromosome segregation", "Cell cycle", "DNA replication", "Mitotic G1 phase, and G1/S transition" (Fig. 6l, Supplementary Data 5). While HHEX negatively regulated genes were enriched in terms such as "Positive regulation of apoptotic process" and "Negative

regulation of cell differentiation", as well as "Negative regulation of cell proliferation", consistent with the previous studies[73–75]. In support of these findings, knockdown of *Hhex* promoted the cell death of mSSCs (Fig. 6m). Meanwhile, we found a clear cell-cycle arrest of mSSCs induced by *Hhex* knockdown (Supplementary Fig. 10i), along with a dramatic downregulation of cyclin genes (*Ccna2*, *Ccnb1*, and *Ccnb3*) and cyclin-dependent-kinase genes

**Fig. 6 Formation of the spermatogonial stem cell pool. a** Violin plots showing the relative expression levels (log(TPM/10 + 1)) of the identified marker genes of mitotic arrest PGCs, ProSPG and SPG. **b** Immunofluorescence of ZXDC co-stained with ZBTB16 in PND3 mouse testis. Scale bar, 10 μm. **c** Immunofluorescence of KRT18 co-stained with DDX4 in E18.5 mouse testis. Scale bar, 10 μm. **d** Immunofluorescence of TIE1 co-stained with DDX4 in E18.5 mouse testis. Scale bar, 10 μm. **e** Immunofluorescence of HHEX co-stained with ZBTB16 in PND5 mouse testis. Scale bar, 10 μm. Arrowheads indicate the HHEX+ SPG. **f** Immunofluorescence of NEFM co-stained with DDX4 in PND5 mouse testis. Scale bar, 10 μm. **g** Schematic showing the workflow of gene knockdown in mSSCs. **h** Representative cell morphology of *Hhex*-knockdown mSSCs compared to control mSSCs transduced with non-targeting shRNA (shNC). Scale bar, 200 μm. **i** Proliferation of mSSCs upon *Hhex* knockdown. Data are shown as the mean ± SEM, $n$ = at least 3 biologically independent samples for each group. **j** Heatmap showing the hierarchically clustered Pearson's correlation coefficient among different samples based on global gene-expression profiling in bulk cell populations. **k** Volcano plots showing DEGs (performed by DESeq2 with likelihood ratio test) between the cells transfected with non-targeting shRNA or with shRNAs against *Hhex*. The number of DEGs ($|\log_2(\text{fold change})| \geq 0.5$, $q$-value < 0.05) are shown above the plots. **l** Representative GO terms of DEGs in *Hhex*-knockdown mSSCs. **m** Cell death of *Hhex*-knockdown mSSCs. Quantitative data are shown as the mean ± SEM to the right, $n$ = 2, * $P$ < 0.05, ** $P$ < 0.01, sh-NC vs. sh-*Hhex*-1 ($P$ = 0.0074), sh-NC vs. sh-*Hhex*-2 ($P$ = 0.0109), unpaired two-tailed $t$ test.

(*Cdk1* and *Cdk2*) as well as an upregulation of cyclin-dependent-kinase inhibitor genes such as *Cdkn1c*, which is a tight-binding inhibitor of several G1 cyclin/Cdk complexes and a negative regulator of cell proliferation (Fig. 6l, Supplementary Data 5)[76]. Altogether, these results suggest that HHEX plays a critical role for the maintenance of mSSCs through prevention of cell death and promotion of their proliferative capacity.

**Male germ cell meiosis and spermiogenesis.** To characterize a dynamic developmental and cellular sequence of events during the conversion processes from SPG to SPC, and eventually to RS, we analyzed the global differentially expressed genes and GO terms during this process. As expected, "Cell cycle", "Gamete generation", "Synaptonemal complex assembly", "Cilium organization", and "Sperm axoneme assembly" were significantly enriched in different stages of spermatogenic cells (Supplementary Fig. 13a). Meanwhile, DEG analysis and GO analysis of each spermatogenic cell stage with their former- and later-stage cells were conducted (Supplementary Data 4). Furthermore, we found that transcriptional inactivation and reactivation occurred on sex chromosomes in the conversion from SPG to SPC, and SPC to RS, respectively (Supplementary Fig. 13b). The minimum transcriptional state on sex chromosomes was observed in SPC and the gene-expression level gradually increased, reaching chrX+Y/chrA ratios comparable to SPG in RS, which was consistent with a previous study[19].

**Interspecies comparison of germ cell development between human and mouse.** We next sought to utilize our high-precision reference map of full-term mouse germ cell development to infer the evolutionary conservation of the transcriptional regulation of this program in humans. To this end, we integrated the scRNA-seq datasets of mouse PGCs in this study and our previously reported scRNA-seq datasets of human PGCs[29]. Standardized approaches were used to process the raw data, and dimension reduction showed that migrating PGCs, mitotic PGCs, and mitotic arrest PGCs followed similar trajectories in human and mouse (Fig. 7a, b). This finding was supported by the expression patterns of well-known germ cell markers (Supplementary Fig. 14a). Next, the DEGs of each cell type were independently inspected in human and mouse. Overall, the 8,106 DEGs with clear orthologues can be further divided into three subsets —human-specific DEGs (850), mouse-specific DEGs (6,209), and human-mouse-shared DEGs (1,227) (Fig. 7c). To identified regulatory principles that are evolutionarily conserved, we focused our subsequent analysis on these 1,227 human-mouse-shared DEGs of which 701 were stage-matched (104 in migrating PGCs, 75 in mitotic PGCs, 522 in mitotic arrest PGCs) and 526 non-stage-by-stage matched DEGs (Fig. 7c, Supplementary Data 6).

To analyze the regulatory networks involved in human and mouse PGC development, single-cell regulatory network inference and clustering (SCENIC) analysis was performed to score the activity of gene regulatory networks at each stage of PGC development. Gene modules exhibiting co-expression patterns with transcription factors were first established. Next, the binding motifs were analyzed to select the prominently enriched modules for evaluating the activity of each subnetwork. Based on the SCENIC results, 224 common active regulators were finally identified between human and mouse (Fig. 7d, Supplementary Data 6). They were defined as the potential determinants for four mouse male PGC stages and three human male PGC stages (without specification stage of human PGCs). We then over-lapped these 224 common active regulators with the 1,227 human-mouse-shared DEGs. This analysis led to the identification of 28 core regulators including a set of 14 highly conserved regulators (common & stage-by-stage matched) (Fig. 7d). ZFP42, one of the 14 inferred highly conserved regulators of mitotic PGCs in human and mouse, is indispensable for the normal development of male germ cells in mice according to the Mouse Genome Informatics (MGI) database. The second set of 14 regulators (common & non-stage-by-stage matched) were conserved in both species but they did not function at the same developmental stages (Fig. 7d). In detail, *Sox15/SOX15*, *Pou5f1/POU5F1*, and *Msc/MSC* were active in mouse/human migrating PGCs, *Rest/REST*, *Mybl2/MYBL2*, *Xbp1/XBP1*, and *Zfp42/ZFP42* were active in mouse/human mitotic PGCs, while another 7 highly conserved regulators such as *Taf7/TAF7*, *Hdac2/HDAC2*, and *Klf11/KLF11* were active in mouse/human mitotic arrest PGCs (Fig. 7d, Supplementary Fig. 14b–e). *Tfap2a* was highly expressed in mouse specification PGCs, but *TFAP2A* was highly expressed in human mitotic arrest PGCs. *Tfap2c* was highly expressed in the stages of mouse specification PGCs and mitotic PGCs, while *TFAP2C* was highly expressed in human mitotic PGCs (Supplementary Fig. 14b–e). Notably, *Hes1*, a marker of transitional PGCs (Fig. 4), was identified to be active in human mitotic arrest PGCs (Supplementary Fig. 14b, c). ZXDC, a marker of the mouse Q-ProSPG (Fig. 6a), was found to be quite active in both mouse and human PGCs albeit acting as a mouse-specific DEG (Supplementary Fig. 14f, g). Thus, a set of key regulators with potential contribution to PGC development were identified to be conserved between human and mouse (Fig. 7d, e).

We next integrated our mouse data with our previously published datasets of human spermatogenesis[26]. Clustering analysis showed that mouse undiff.ed SPG, Diff.ing SPG, and Diff.ed SPG were generally well-merged with human SSCs, Diff.ing SPG, and Diff.ed SPG, respectively (Fig. 7f, g). With the same analytical criterion, we present a list of species-specific (7,135 in mouse, 773 in human) and human-mouse-shared DEGs (1,373) for each cell type (Fig. 7h, Supplementary Data 6). Further analysis of these 1,373 shared DEGs revealed that a set of stage-

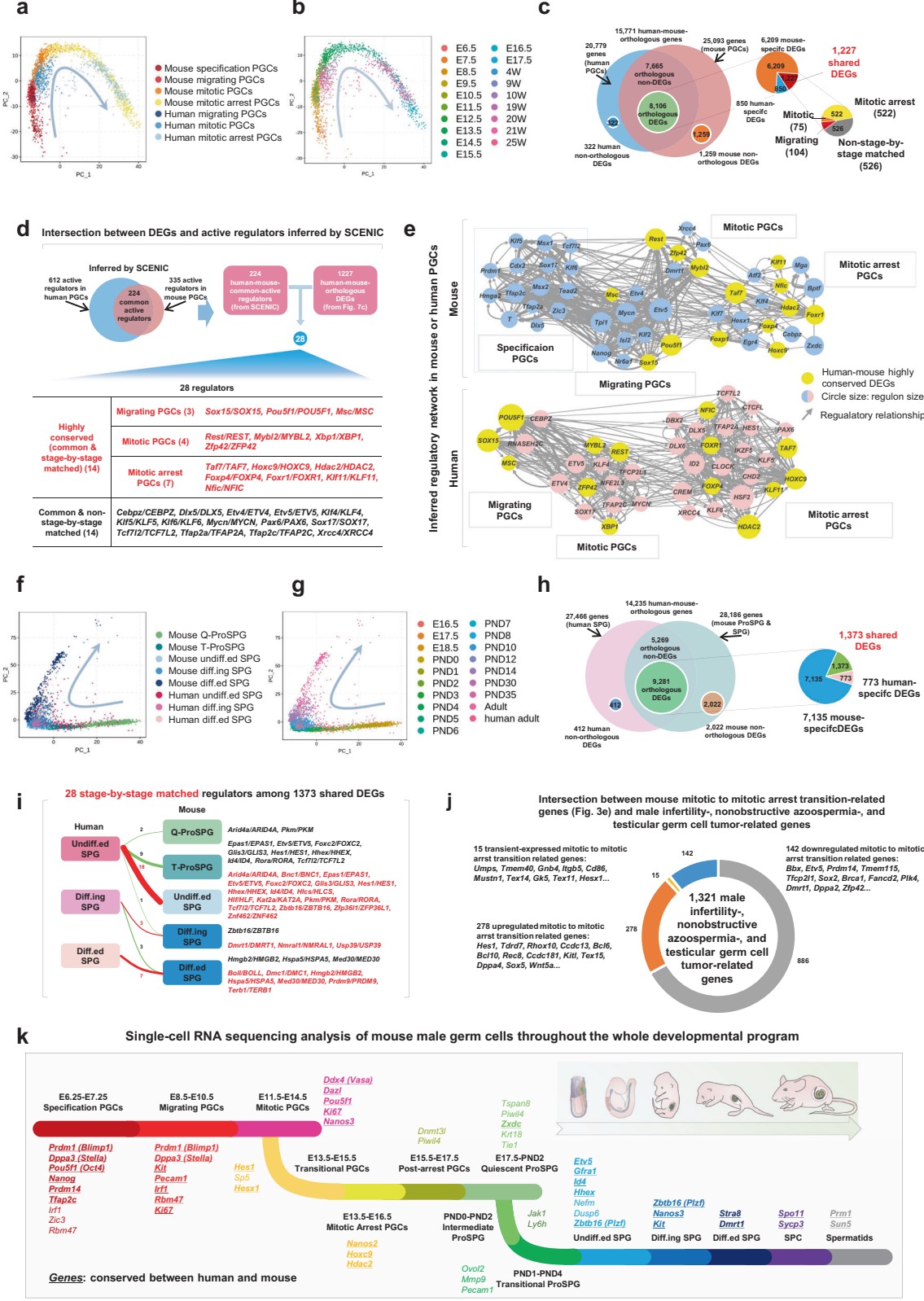

by-stage matched regulators were highly conserved in undiff.ed SPG (18), diff.ing SPG (3), and diff.ed SPG (7), respectively (Fig. 7i). In particular, HHEX, a critical regulator of mSSCs as functionally validated in this study, was highly expressed in both mouse and human undiff.ed SPG. FOXC2, and ETV5, two highly conserved regulators of undiff.ed SPG, were identified to be with regulatory potential in mouse and human (Supplementary

Fig. 14h), consistent with their functions in mSSC maintenance[77,78]. These data indicate that the inferred conservative regulators such as ETV5, FOXC2, and HHEX might contribute to hSSC maintenance.

Finally, we screened out 435 male infertility and testicular germ cell tumor-related genes from the public database (Gene Ontology Resource, OMIM, and NCBI ClinVar), which were

**Fig. 7 Interspecies comparison of male germ cell development between human and mouse. a** PCA plots of four phases of mouse male PGCs and three phases of human male PGCs. Cells are colored based on the cell types are shown. **b** PCA plots of four phases of mouse male PGCs and three phases of human male PGCs. Cells are colored based on the sampled time-points shown. **c** Pie chart showing the distribution of human-mouse orthologous- and non-orthologous genes in the transcription profiles of mouse male PGCs and human male PGCs. The distribution of stage-by-stage matched and mismatched DEGs are also shown. **d** Intersection between DEGs and active regulators inferred by SCENIC in mouse male PGCs and human male PGCs. **e** Inferred regulatory networks of representative regulators in mouse male PGCs (Top) and human male PGCs (Bottom) are shown. Nodes labeled by yellow indicate the human-mouse highly conserved DEGs (common & stage-by-stage matched regulators). **f** PCA plots of five phases of mouse male ProSPG and SPG and three phases of adult human male SPG. Cells are colored based on the cell types are shown. **g** PCA plots of five phases of mouse male ProSPG and SPG and three phases of human adult male SPG. Cells are colored based on the sampled time-points shown. **h** Venn diagram showing the distribution of human-mouse orthologous- and non-orthologous genes in the transcription profiles of mouse male ProSPG and SPG and adult human male SPG. **i** 28 stage-by-stage matched regulators among 1,373 shared DEGs are shown. **j** Intersection between mouse mitotic to mitotic arrest transition-related genes (Fig. 3e) and male infertility-, nonobstructive azoospermia-, and testicular germ cell tumor-related genes. **k** The cell stages determined by scRNA-seq during mouse male germ cell development are shown in different colors. Genes conserved in humans and mice are bolded and underlined.

also identified to be critical for the mitotic to mitotic arrest transition in mice based on the aforementioned results (Figs. 3e, 7j, and Supplementary Data 6). Together, these disease-related human-mouse orthologous genes facilitate us to decipher human male infertility and testicular germ cell-related diseases.

## Discussion

A fundamental question in developmental biology is how mammalian PGCs are specified from the post-implantation epiblasts and how these cells sequentially transit in a stepwise manner to become mature sperm. Here, we systematically characterize the single-cell transcriptome atlas of full-term male germ cell development with unprecedented resolution and precision. Our deep scRNA-seq method enabled us to capture rare transitional cell types and identified the specific and exclusive detection markers for each cell state. Based on the evidence in this study, a cascade of cellular states with hallmark transcriptome signatures was discovered in mouse male germ cell development (Fig. 7k, Supplementary Fig. 14i, and Supplementary Fig. 15).

It is well-known that due to the "batch effect" which caused by different sequencing technologies, library preparation protocols, and data interpretation strategies[32], it is quite difficult to integrate different single-cell sequencing data and generate an unbiased and high-quality atlas, which is critical for faithfully discovering undefined sub-populations and performing analysis such as calculating differentially expressed genes without inaccuracy[79]. Therefore, to our knowledge, we carried out so far the most comprehensive and accurate scRNA-seq analysis of essentially all stages of male germ cells. The unbiased single-cell transcriptome atlas of male germline cycle established here allowed for an accurate overview of the cell-fate transition and determination underlying male germ cell development. The developmental trajectory and the identified cell sub-populations based on our data could be mostly recapitulated by integrating other germ cells scRNA-seq datasets[71,72], although certain cell stages were not covered in these studies (Supplementary Fig. 11, 12). Additionally, the map we constructed here represents a reference framework that facilitates the integration with other layers of omic information to characterize the epigenetic regulation in male germ cells. Recent studies using multi-omics analyses revealed the regulatory elements associated with germ layer emergence[80]. Thus, it will be interesting to explore the epigenomic regulatory principles throughout male germ cell development using single-cell multi-omics profiling. Further, the full-term expression dynamics of key regulators will guide the functional evaluation of proposed candidate genes using inducible gene knockout and overexpression strategies.

The development from specified PGCs to mature haploid gametes is crucial for the maintenance of a species. In this study, we revealed that the Notch signaling pathway was highly enriched

in mitotic arrest PGCs, suggesting its potential roles. Further study via Notch signaling pathway blocking demonstrated its essential roles in the regulation of mitotic arrest in male PGCs as exemplified by the increase of MKI67+ cells. Previous studies found that Notch signaling pathway was involved in PGC or SPG development in mouse, chicken, sea urchin, Xenopus, *Drosophila*, and *C. elegans*, or in mouse gonadal somatic progenitor cells to Leydig/interstitial cells differentiation, but its direct functions in mouse male gonadal PGCs have not been characterized[81–87]. In this study, we showed that the Notch signaling pathway was intrinsically activated in mouse male gonadal PGCs. Anyway, our Notch signaling pathway blocking results and the reported phenotypes of *Hes1*−/− mice[81] together proved the importance of this signaling pathway in male germ cell development. It should be noted that *Notch1*−/−, *Notch2*−/−, and *Hes1*−/− mice all exhibited either complete embryonic or complete perinatal/postnatal lethal phenotype[88–90], making it difficult to use these genetic mouse models for the study of their roles in male reproductive function. In addition, *Notch3*−/− and *Notch4*−/− mice are viable and fertile with no apparent lack of reproductive function[91,92]. Taking these factors into account, the inhibitors of Notch signaling pathway-DAPT and LY411575 were therefore used to block the Notch signaling pathway activation in the present study. Previous studies show that Sertoli cell-Notch signaling pathway plays the role in fetal germ cell development[60,61]; therefore, a future study using conditional knockout of key regulators of this pathway is aimed to confirm the detailed functions of Notch signaling pathway in fetal germ cells. Similarly, cell-cycle regulators have been identified to be highly associated with male germ cell arrest[12,13]. HELQ is crucial for male germ cell development and HELQ deficiency is associated with male subfertility[62], but when and how HELQ regulate germ cell development prenatally are still unexplored. Here we showed that *Helq* knockout mice exhibited decreased proportion of PGC numbers and aberrant mitotic to mitotic arrest transition. Mechanistically, the p-ATR/p-CHK1 regulatory axis was hyperactivated in *Helq*−/− PGCs, suggesting that *Helq* knockout enforced the S/G2 checkpoint and resulted in the cell-cycle arrest in S/G2 phase; as a result, a clear developmental arrest was uncovered. In addition, an aberrant upregulation of E-cadherin (encoded by *Cdh1*) was observed in *Helq*−/− germ cells. This finding was consistent with previous studies that the expression of E-cadherin (encoded by *Cdh1*) in male germ cells should be downregulated once sexual differentiation completed, which was critical for the cell fate transition of prenatal male germ cells[93,94]. Based on these data, it is most likely that HELQ also functions in the regulation of mouse male PGC differentiation. More broadly, HELQ ablation assays suggested that the phenotype of some infertile or subfertile male patients carrying mutated genes like *Helq* might be caused by developmental defects at the mitotic to

mitotic arrest transition stage, which eventually led to an unrecoverable reduction of SSC pool. Thus, these findings uncovered by scRNA-seq and functional validations should provide valuable advice for prenatal diagnostic testing and manipulating PGCs at the transitional stage might be a promising strategy to treat infertility or subfertility. Nevertheless, it should be noted that the present lack of stable prenatal germ cell culture system essentially limits functional validations of the identified regulators. To summarize, our study on the Notch signaling pathway and HELQ demonstrated the biological importance of cell-cycle switch for the proper transition from mitotic to post-arrest PGCs, emphasizing the biological significance of transitional PGCs. Multiple lines of evidence support the notion that developmentally defective germ cells are eliminated through apoptosis before birth[95]. Therefore, healthy PGCs might be discriminated from developmental incompetent cells via a differential surveillance of DNA damage.

Due to the difficulties to get access to certain stages of human germ cells such as specification stage of human PGCs, the full-term developmental program of human male germ cells remains unexplored. To overcome these shortcomings, we performed a comprehensive human-mouse comparison at the PGC and spermatogonia stages. Overall, the developmental trajectory from gonadal PGCs to differentiated spermatogonia was found to be very similar in human and mouse. Our data revealed a set of conserved regulators involved in germ cell development of human and mouse, highlighting the value of the accumulated knowledge of mouse model to understand human germ cell development. Of note, a set of potential regulators (435), which were identified to be critical for the mitotic to mitotic arrest transition of mouse PGCs, were highly associated with human male infertility and testicular germ cell tumor. Given that scRNA-seq datasets used here were all constructed by one laboratory via a consistent method, the present comparison between human and mouse germ cell development could reliably represent the human-mouse differences and similarities.

In conclusion, our work offers an unprecedented high-precision single-cell transcriptome atlas of full-term development of mouse male germ cells, illuminating the cell-fate transition landscape of male germ cells. In this context, we uncovered several critical developmental stages/regulators for prenatal germ cell fate determination. Collectively, our study demonstrates that densely sampled large-scale high-precision single-cell profiling has the potential to advance our understanding of male germ cell development in mammals. Further studies will be aimed to establish the regulatory networks of postnatal male germ cells and explore other key events, such as epigenetic reprogramming.

## Methods

**Animals**. All animal studies were performed in accordance with the ethical guidelines of South Medical University ethics committee (L2016149). Animal-related research protocols were performed in accordance with the U.S. Public Health Service Policy on Use of Laboratory Animals, and were approved by the Ethics Committee on Use and Care of Animals of Southern Medical University.

Male BVSC (*Blimp1*-mVenus and *Stella*-ECFP) transgenic mice were obtained from Mitinori Saitou laboratory (Kyoto, Japan). Female C57BL/6 mice were purchased from the Guangdong Medical Laboratory Animal Centre (Guangzhou, China). *Helq*-KO mice were generated by CRISPR/Cas9 technology (see below). Mice were kept in a standard 12 h light-dark cycle in the specific-pathogen free conditions, and were permitted for free access to water and food. The ambient temperature was 20–25°C and the humidity was 40-70%. All the mice we used were healthy and immune-normal.

Notch signaling pathway was inhibited by intraperitoneally injection of 200 mg/kg DAPT (*N*-[*N*-(3,5-difluorophenacetyl)-L-alanyl]-S-phenylglycine t-butyl ester, Selleck, S2215), and 10 mg/kg LY411575 (N-[(1S)-2-[[(7S)-6,7-dihydro-5-methyl-6-oxo-5H-dibenz[b,d]azepin-7-yl]amino]-1-methyl-2-oxoethyl]-3,5-difluoro-αS-hydroxy-benzeneacetamide, Selleck, S2714) dissolved in 10% ethanol and 90% corn oil. Pregnant mice were injected twice, at E13.5 and E14.5, the embryos were dissected and genital ridges were analyzed at E15.5 onward.

**Cell culture**. mSSC lines were derived from the testis of 5.5 days postpartum (dpp) mice with the genetic background of B6D2F1 (C57BL/6 × DBA). mSSCs were cultured as previously described[96,97] with some modifications: StemPro-34 SFM (Invitrogen) supplemented with StemPro supplement (Invitrogen), 1% FBS, N2 (100×), 6 mg/ml D-(1)-glucose, 5 mg/ml BSA, 0.1 mM NEAA, 1 mM sodium pyruvate, 0.1 mM 2-mercaptoethanol, 2 mM L-glutamine, $10^{-4}$ M ascorbic acid, 10 mg/ml biotin, 30 ng/ml β-estradiol, 60 ng/ml progesterone (Sigma), 20 ng/ml mouse EGF, 10 ng/ml human bFGF, 10 ng/ml recombinant rat GDNF (R&D Systems) and $10^3$ U/ml LIF (Merck Millipore). SSCs were passaged every 6 days by dissociating with 0.05% trypsin and changed the medium every 2 days.

BVSC ESCs (embryonic stem cells) were established and maintained under the N2B27 '2i-LIF' condition on feeder cells as previously described[98]. Briefly, the blastocysts were plated in N2B27 '2i-LIF' medium. Several days later, the cell masses were dissociated using 0.05% trypsin and replated into culture dished with feeder cells. ESCs were passaged every two days using 0.05% trypsin and the medium was changed every day.

**Single-cell collection**. Five- to six-week-old female C57BL/6 mice in natural estrus were mated with male homozygous BVSC (*Blimp1*-mVenus and *Stella*-ECFP) mice. Female mice with vaginal suppository were retained for further use and noon of the day was set as E0.5. The collection of germ cells from E6.5 and E7.5 embryos was strictly performed in accordance with the previous studies[67,99].

The germ cells ranged from E8.5 to E18.5 were collected with the assistance of fluorescent active cell sorting (FACS) using Blimp1-mVenus and Stella-ECFP signals as shown in Supplementary Fig. 16. Whole embryos at the stages from E8.5 to E10.5 were digested using the TrypLE™ Express enzyme to isolate the BVSC+ cells through FACS. For stages E10.5 and E11.5, male embryos and gonads were selected by PCR for the genomic DNA of male specific *Sry* gene. Of note, gonads from E13.5 to E16.5 embryos were divided into three parts and defined as the proximal, central and distal according to the relative positions with respect to the mesonephros. Due to the downregulation of the Blimp1-mVenus signal from E14.5 to E18.5, single positive Stella-ECFP cells were collected from E14.5. After FACS, the collected cells were examined under a fluorescence microscope to further confirm the fluorescence signal.

After birth, male gonads were treated as previously described[71] and testicular cells with typical morphology of germ cells were randomly selected.

**scRNA-seq library preparation and sequencing**. After sample collection, a mouth pipette was used to immediately transfer the single cell into prepared lysis buffer with an 8-nt barcode. The preparation of scRNA-seq library was performed using the STRT-seq method[100]. In brief, samples were incubated at 72°C for 3 min after vortexing for 60 s. Then, the first-strand cDNA was reverse-synthesized using oligo(dT) anchored with cell-specific barcode, unique molecular identifiers (UMIs) and template-switching oligonucleotides (TSO). After that, the second-strand cDNAs were synthesized, and the cDNAs were amplified by 18 cycles of PCR. Subsequently, the products of single cells were pooled and purified. Next, biotinylated pre-indexed primers were used to further amplify the PCR product by an additional four cycles of PCR to introduce biotin tags to the 3′ ends of the amplified cDNAs. About 300 ng of cDNA was then sheared with a Covaris M220 to obtain fragments with an average length of approximately 300 bp. Dynabeads® MyOne Streptavidin C1 beads (Thermo Fisher) were used to enrich for the 3′ terminal cDNAs. The RNA-seq library was then constructed using a Kapa Hyper Prep Kit (Kapa Biosystems) and subjected to 150 bp paired-end sequencing on an Illumina HiSeq XTEN platform (sequenced by Novogene).

**scRNA-seq data processing**. Reads 1 of sequencing read pairs were added the 8-nt cell barcode sequence and unique molecular identifiers (UMIs) based on reads 2 with "extract" tool in UMI-tools (version 1.0.0)[101]. Subsequently, template switch oligo (TSO) sequence, ployA tail sequence and low-quality bases in reads 1 were removed via costumed script and "trimfq" command in seqtk (version 1.3)[102]. Clean reads 1 were aligned to mm10 mouse reference (References-3.0.0, download from https://support.10xgenomics.com/single-cell-gene-expression) using STAR (version 2.7.1a) with options "–outFilterMultimapNmax 3" and "–outFilterMismatchNmax 4"[103]. To count uniquely aligned reads with genes, genomic features were then added to reads in BAM file using featureCounts of subread (version 1.6.4). "count" tool in UMI-tools was used to remove PCR duplicates based on barcode and UMI information. Gene-expression level was presented with log(TPM/10+1) unless specifically mentioned.

**Quality control of scRNA-seq data**. We filtered cells via the following two criteria: the number of detected genes is greater than 2,000; the number of detected transcripts is between 10,000 and 1,000,000. Overall, we captured 11,598/11,896 individual gonadal and testicular cells from healthy antenatal and postnatal male mice and 1,094/1,136 individual gonadal cells from $Helq^{-/-}$ male mice. On average, 9,413 genes and 302,772 transcripts were detected in normal cells and 8,370 genes and 263,296 transcripts were detected in $Helq^{-/-}$ cells.

**Dimension reduction and cell-type identification**. Seurat package (version 3.0.0)[34,35] was conducted to perform dimension reduction and clustering analysis.

For normal scRNA-seq data, firstly, raw UMI count matrix was used to create a Seurat object. We normalized the gene-expression measurements for each cell with the parameter "scale.factor = 100,000". Subsequently, 1000 highly variable genes were selected to perform PCA on the scaled data. Based on "JackStraw" results, top 30 significant PCs were chosen to construct a KNN graph and run non-linear dimensional reduction (UMAP). "FindClusters" function was used to cluster the cells with the parameter "resolution = 0.3" and 26 original clusters were merged into 11 clusters (C1-C10 and blood cells) based on specific markers. Notably, 20 single cells with abnormal correspondence between time-points and cell types assignment were treated as outliers. To explore finer heterogeneity among cells, certain clusters were reanalyzed with the similar pipelines and appropriate parameters. Differential gene-expression analysis was performed using "FindAllMarkers" function unless specifically mentioned in this paper.

**Cell-cycle analysis**. Cell-cycle analysis was performed using "CellCycleScoring" function in Seurat. Cell-cycle scores of each cell were assigned, and cell-cycle phase identities were also set by passing "set.ident = TRUE".

**Self-organizing map (SOM) algorithm**. The kohonen package in R was used to train a SOM for 282 pluripotency-related genes (downloaded from AmiGO: Version 2020-03-25 10.5281/zenodo.3727280). UMI count was first normalized to TPM and median value was calculated for each type of germ cells to generate a gene-cell-type matrix, which is used as the input data for SOM training. The total number of map units was set to 84 (determined as 5 * sqrt(N), where N is the number of input genes). Visualization was performed with custom R code.

**Pseudotime analysis**. Monocle package (version 2.10.1)[104] was used to construct cellular fine-grain trajectories in migrating PGCs (534 cells). "DDRTree" algorithm was used to reduce dimensionality. Then, 500 highly variable genes selected from migrating PGCs Seurat object were used to order cells in pseudotime along a trajectory. Root of the trajectory was assigned based on physiological development time.

**Calculation purity scores of cell populations**. In order to assess the purity of mitotic arrest PGCs, an entropy-based metric, ROGUE[51], was used to calculate purity scores of cell populations before and after re-clustering of mitotic arrest PGCs. We first filtered out low-abundance genes expressed less than 10 cells. We then used "SE_fun" function to calculate the expression entropy for each gene. Using "CalculateRogue" function, we calculated the ROGUE value of each cell type.

**Integration and comparative analysis of human and mouse scRNA-seq data**. To analyze the similarities and differences of germ cell development between human and mouse, scRNA-seq datasets of human PGCs (678 cells)[29] and spermatogonia (323 cells)[26] were used. The CCA anchor method from Seurat package was applied to integrate human and mouse datasets. We obtained UMI count matrix of 15,771 and 14,235 human-mouse one-to-one orthologous gene pairs corresponding to PGCs and spermatogonia. Standard preprocessing was performed with 1,000 highly variable genes. Then we identified anchors using top 30 dimensionalities. PCA was used to visualize the integrated results.

For differential gene-expression analysis, DEGs ($q$-value < 0.05) of PGCs and SPG stage in human and mouse were obtained using "FindAllMarkers" function in Seurat, respectively.

**Comparative analysis of human and mouse gene regulatory networks**. The SCENIC pipeline (as implemented in pySCENIC)[105] was performed to infer transcription factors, gene regulatory networks of germ cell development in human and mouse. TF-modules were first created relying on co-expression and then regulons were derived from adjacencies based on TF-motif enrichment analysis. After that, AUCell method was used to score the activity of regulons on single cells. A binary matrix was generated using a cutoff of the AUC score for each regulon. A set of 612, 335, 120, and 257 TF regulons from human PGCs, mouse PGCs, human SPG and mouse SPG were obtained, respectively. Network connections of the selected genes were visualized using Cytoscape (version 3.7.1).

**Treatment of γ-Secretase inhibitor−DAPT or LY411575 for Notch signaling pathway blocking in vitro**. The in vitro culture of mouse PGCs was performed with minor modifications[4]. Briefly, E14.5 male BVSC$^+$ PGCs were sorted by FACS and cultured in a well of 5,000 cells in a low-cell-binding U-bottom 96-well plate in GMEM (Invitrogen) containing 15% KSR, 0.1 mM NEAA, 1 mM sodium pyruvate, 0.1 mM 2-mercaptoethanol, 0.1 mg/ml streptomycin, 2 mM L-glutamine in the presence of LIF (1,000 U/ml; Merck Millipore), SCF (100 ng/ml; R&D Systems), and EGF (50 ng/ml; R&D Systems). The cells were collected after 24 h treatment by vehicle or γ-Secretase inhibitors: DAPT (100 μM, Selleck, S2215) and LY411575 (100 μM, Selleck, S2714).

**Quantitative PCR (q-PCR)**. Total RNA was extracted using TRIzol Reagent (Invitrogen). Total RNA was reverse transcribed into complementary DNA (cDNA) using HiScript QRT SuperMix (Vazyme). q-PCR was conducted with 2×PCR Master Mix (GenStar) on a LightCycler+96 Real-Time PCR system (Roche). All data were normalized to the expression of housekeeping gene *Gapdh* and calculated by ΔCq or ΔΔCq. All q-PCR primer pairs were listed in Supplementary Data 7.

**Western blot**. Cell extracted were prepared using cold RIPA buffer (Solarbio) supplemented with 1 mM phenylmethylsulfonyl fluoride and protease inhibitor cocktail (Roche) on ice for 30 min. The homogenates were centrifuged at 13,000 rpm for 10 min and the supernatant was boiled with 5 × loading buffer for 5 min at 98 °C. Protein lysates were subjected to SDS-PAGE and subsequently electrotransferred to polyvinylidene fluoride membranes. The membrane was incubated with the indicated primary antibodies overnight at 4 °C and HRP-conjugated secondary antibodies, followed by visualization using Super ECL Detection Reagent ECL (YEASON). The primary antibodies used were mouse anti-HELQ antibody (1:100, Santa Cruz Biotechnology #sc-81095), rabbit anti-HHEX antibody (1:500, R&D System #MAB83771), and mouse anti-α-Tubulin antibody (1:5000, Tianjin Sungene Biotech #KM9007T).

**Immunofluorescence**. The isolated mouse embryos at E7.5 and E9.5, male genital ridges at E11.5−E18.5, and testes at PND1-7 were fixed with 4% paraformaldehyde at 4 °C overnight, and washed extensively with phosphate-buffered saline (PBS). Then, the tissues were dehydrated and embedded in paraffin, and then cut into sections of 5-μm (genital ridges and testis)/10-μm (embryos) thickness, respectively. Immunostaining was performed after deparaffinization and rehydration as previously described[7]. The primary antibodies used were anti-DDX4 antibody (1:500, Abcam #ab27591), rabbit anti-DDX4 antibody (1:500, Abcam #ab13840), rabbit anti-IRF1 antibody (1:500, Abcam #ab186384), chicken anti-GFP antibody (1:500, Abcam #ab13970), rabbit anti-Ki67 antibody (1:500, Abcam #ab15580), rabbit anti-Cleaved PARP1 antibody (1:500, Abcam #ab32064), mouse anti-P53 antibody (1:200, Cell Signaling Technology #2524), rabbit ant-ATR (1:500, Abcam #ab227851), rabbit anti-ZXDC antibody (1:100, Proteintech Group, Inc #20530-1-AP), mouse anti-PLZF antibody (1:100, Santa Cruz Biotechnology #sc-28319), rabbit anti-Cytokeratin 18 (Krt18) antibody (1:200, elabscience # ESAP10261), mouse anti-TIE1 antibody (1:500, Abcam #ab201986), mouse anti-Neurofilament Medium antibody (1:250, Abcam #ab7794), mouse anti-Notch1 antibody (1:200, Santa Cruz Biotechnology #sc-373891), rabbit anti-Hes1 antibody (1:500, Abcam #ab108937), mouse anti-Ki67 antibody (1:500, Abcam #ab279653), rabbit anti-Phospho-Chk1 antibody (1:50, Cell Signaling Technology #2348), rabbit anti-Wee1 antibody (1:100, Cell Signaling Technology #13084), rabbit anti-KLF4 antibody (1:50, MilliporeSigma #HPA002926), rabbit anti-E-Cadherin antibody (1:50, Cell Signaling Technology #3195), rabbit anti-Notch2 antibody (1:200, Cell Signaling Technology #5732), rabbit anti-Notch3 antibody (1:150, Abcam #ab23426), rabbit anti-Notch4 antibody (1:150, Abcam #ab184742) and mouse anti-5mC antibody (1:300, Active Motif #39647). The secondary antibodies used were Goat Alexa Fluor 488 anti-rabbit IgG (1:500, Jackson ImmunoResearch #111-545-003), Goat Alexa Fluor 594 anti-rabbit IgG (1:500, Jackson ImmunoResearch #111-585-003), Goat Alexa Fluor 647 anti-rabbit IgG (1:500, Jackson ImmunoResearch #111-605-003), Goat Alexa Fluor 488 anti-mouse IgG (1:500, Jackson ImmunoResearch #115-545-003), Goat Alexa Fluor 594 anti-mouse IgG (1:500, Jackson ImmunoResearch #115-585-003), Goat Alexa Fluor 647 anti-mouse IgG (1:500, Jackson ImmunoResearch #115-605-003), Donkey Fluorescein (FITC) anti-Chicken IgG (1:500, Jackson ImmunoResearch #703-095-155). Images were captured with a ZEISS LSM880 confocal microscope. Five to ten fields were randomly selected.

**Simultaneous RNAscope and immunofluorescence assay**. To visualize the transcription of mRNA and distinguish subtypes of PGCs, the RNAscope probe targeting *Wee1* was designed and synthesized by Advanced Cell Diagnostics company, and the assay of RNAscope and IF were performed by using RNAscope® Multiplex Fluorescent Reagent Kit v2. For each experiment, *POLR2A*, *PPIB*, *UBC* and *HPRT* were used as the positive controls; *dapB* was used as the negative control.

Briefly, after fixing in 10% NBF (Neutral buffer formalin) for 24 h, fresh mouse male gonadal sections (5-μm-thin) were prepared and then pretreated with hydrogen peroxide solution, target retrieval solution, and stained with primary antibodies overnight at 4 °C. Sections were further treated with protease plus and finally hybridized with the RNA probe of target gene for 2 h at 40 °C in hybrid furnace, followed by a series of signal amplifications. After RNAscope, sections were stained with secondary antibody 30 min at room temperature. And Nuclei were counterstained with Hoechst for 10 min at room temperature. Images were obtained with ZEISS LSM880 confocal microscope.

The signal dots were visually counted following the RNAscope manual (as outlined in https://acdbio.com/how-correctly-interpret-your-rnascope%C2%AE-images), and single dot equals to single mRNA. Robust cutoff for determining the "High expression level" and "Low expression level" of *Wee1* mRNA was set up using the following criteria: (1) The number of dots in each cell is an integer, so the cutoff must also be an integer; (2) The mean dot in random PGCs is 5.35, which means PGCs with more than 5 (less than and nearest to 5.35) are *Wee1* mRNA high expression, otherwise are low expression; (3) The 25% percentile of the dot

numbers in WEE1[High] (WEE1 protein high expression) PGCs is 5.25, representing the lower threshold of expected expression is 5 (less than and nearest to 5.25). Overall, PGCs with less than or equal to 5 dots were defined as *Wee1*[Low] (*Wee1* mRNA low expression) ones, and PGCs with more than 5 dots were defined as *Wee1*[High] (*Wee1* mRNA high expression) ones. In all cases, statistics were performed following the above criteria.

**Germ cell counting**. Confocal images were used to count the numbers of DDX4+, MKI67+ and cPARA1+ cells in the gonads of WT and *Helq*−/− male mice. The percentage of MKI67+ and cPARA1+ cells with respect to DDX4+ cells was then calculated. For germ cell number comparison between the WT and *Helq*−/− male mice, the data were presented as the normalized number of DDX4+ cells ± SE per 1000 somatic cells[106]. Stained sections of at least four WT or *Helq*−/− mice were analyzed at each time-point. Unpaired two-tailed t-test was performed to calculate *P* values.

**Preparation of *Helq*−/− mice**. *Helq*−/− mice were generated by CRISPR/Cas9 technology. Briefly, one single-guide RNA (sgRNA) were designed targeting the first exon of *Helq* and cloned into pSpCas9(BB)-2A-GFP (PX458) vector as previously described[107]. The designed plasmid was electroporated into mouse BVSC ESCs using Neon[TM] Transfection System (Thermofisher) according to the manufacturer's instructions. Two days later, GFP-positive cells were sorted by FACS and transferred manually into Matrigel-coated 96-well-plates with one single cell per well. Genomic DNA was extracted from the harvested cells and used for Sanger Sequencing. The targeted ESCs with 8-nt deletion in the two alleles of *Helq* were injected into blastocysts to generate chimeric mice. The chimeric mice were self-crossed to generate *Helq*−/− mice. Genomic DNA was extracted to confirm the genotype.

**Construction of shRNA vectors and lentivirus preparation**. Two shRNAs against *Hhex* were designed using pLKO.1-TRC shRNA system as previously described[108]. For lentiviral vector production, HEK293T cells cultured in 10-cm dishes and transfect at 70–80% confluency using Lipofectamine® LTX & PLUS™ Reagent (Invitrogen) in Opti-MEM (Thermo Fisher). After incubation for 48 h in DMEM (containing 10% FBS), the supernatant was filtered and concentrated by ultracentrifugation. The pelleted viruses were stored at −80 °C for further use. The control in this test is a group with no shRNA added, the sh-NC is a group with lentivirus carrying shRNA against no target, and sh-*Hhex*-1 and sh-*Hhex*-2 are groups with lentivirus carrying shRNA against *Hhex*. Oligonucleotides used in this study were listed in Supplementary Data 7.

**Generation of knockdown mSSC lines**. For lentiviral transduction, mSSCs were dissociated into single cells and $2.5 \times 10^4$ cells per well were cultured in a 12-well plate. Lentivirus was added after 24 h and the medium was changed after 48 h. Puromycin was used to select transduced mSSCs at a final concentration of 0.4 μg/ml.

**Cell-cycle and cell death analysis**. For analysis of cell cycle in knockdown or knockout cells, the cell pellet was gently mixed with pre-cooled 70% ethanol and fixed at 4 °C for more than 2 h. After washing with cold PBS, cells were treated with Rnase A and stained with PI (Cell Cycle and Apoptosis Analysis Kit, Yeasen) and analyzed using flow cytometer (CytoFlex, Beckman) as depicted in Supplementary Fig. 16.

For analysis of cell death in knockdown mSSCs, freshly collected cells were stained with Annexin V and PI (Annexin V-FITC/PI Apoptosis Detection Kit, Keygen) and analyzed using flow cytometer (CytoFlex, Beckman). The DNA content of cells was analyzed by ModFit software (ModFit LT 5.0, Verity Software House).

**Fluorescent activated cell sorting (FACS)**. The sample preparation was performed as mentioned above. To distinguish the cell ploidy, the suspended cells were stained with 4 μg/ml Hoechst at 37 °C for 20 min. All samples were resuspended with PBS (containing 2% FBS). FACS was performed with a MoFlo XDP (Beckman Coulter) cell sorter. BV- and SC fluorescence were detected with the FITC- and 405 nm channels, respectively. Hoechst was detected with the 355 nm channel. The sorted cells were used for the subsequent scRNA-seq analysis.

**Bulk RNA-seq library construction and sequencing**. Total RNA was extracted from $1 \times 10^6$ cells using TRIzol Reagent (Invitrogen). 5 ng total RNA per sample was used to the construction of libraries using the STRT-seq method. Sequencing was performed on Illumina HiSeq 4000.

**Statistics and reproducibility**. The experimental data were statistically analyzed using unpaired two-tailed *t* test to compare differences between different groups with GraphPad Prism version 6.0.0. The experiments in Fig. 4a, b, f, g, h, i, 5e, f, g, l, 6b, c, d, e, f, h and Supplementary Figs. 3b, c, 4a, e, f, 5c, 6b, b, c, g, h, i, 7c, d, e, f, 8d, e, 9e, 10c, d, e, f, h were conducted at least two times independently, and similar results were adopted for further analysis to guarantee reproducibility.

In Supplementary Fig. 9c, the statistical significance of relative expression level between the WT- and *Helq*−/− male germ cells was evaluated using two-tailed Wilcoxon rank-sum test with ggpubr package in R and *P values* were indicated.

**Reporting summary**. Further information on research design is available in the Nature Research Reporting Summary linked to this article.

## Data availability
The scRNA-seq datasets generated in the present study are deposited and publicly available in the Gene Expression Omnibus (GEO) at NCBI under accession number "GSE148032, ". The mouse male germ cell scRNA-seq landscape can be browsed at https://tanglab.shinyapps.io/Mouse_Male_Germ_Cells/. Single-cell RNA-seq datasets of previous publication are publicly available: Li et al. ("GSE86146")[29], Wang et al. ("GSE106487")[26], Hermann et al. ("GSE108970", "GSE108974")[71], and Green et al. ("GSE112393")[109].

Mutant phenotypes of representative Human-mouse shared DEGs in mouse reproductive system were available from MGI Database (http://www.informatics.jax.org/), listed in Supplementary Data 6. Male infertility and testicular germ cell tumor-related genes downloaded were downloaded from the public database: Gene Ontology Resource (http://geneontology.org/), Online Mendelian Inheritance in Man database (OMIM, https://www.omim.org/), and NCBI ClinVar (https://www.ncbi.nlm.nih.gov/clinvar/) to identify genes critical for the mitotic to mitotic arrest transition of mouse PGCs.

A reporting summary for this article is available as Supplementary Information file. All other relevant data supporting the key findings of this study are available within the article and its Supplementary Information files or from the corresponding author upon reasonable request. Source data are provided with this paper.

## Code availability
Data analysis pipeline used in this paper is deposited at https://github.com/sunshine-lp0/MGC_project_code [110]. More detailed information is available on demand.

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

## Acknowledgements

The authors thank Prof. Ralf Jauch for helpful discussion and critical reading of the manuscript. We thank Beckman for supporting the FACS. This work was supported by grants from the National Key R&D Program of China (2017YFA0105001 to X.Z., 2019YFA0801802 to M.W., 2020YFA0113300 to M.W.), the National Natural Science Foundation of China (31625018 to F.T., 82071711 to X.Z., 31671544 to X.Z., 31970787 to G.C., 32170869 to G.C., 81901542 to M.W., 31601208 to Z.L.), Beijing Municipal Science & Technology Commission (Z181100001318001 to F.T.), the Natural Science Foundation of Guangdong Province (2019A1515010446 to G.C., 2021A1515010802 to M.W.), Guangzhou Laboratory of Regenerative Medicine and Health Guangdong Province Key R&D Project (2018GZR110104002 to X.Z.), Science and Technology Projects of Guangzhou City (201904020031 to X.Z.), the Natural Science Foundation of Shenzhen (JCYJ20180305163311448 to G.C., JCYJ20210324120212033 to G.C.) and Discipline Construction Funding of Shenzhen (2016-1452).

## Author contributions

X.Z., F.T. and G.C. conceived and supervised the project. J.Z., C.W., M.C., Y.H., G.C., Y.Z., M.W., Z.L., X.W., S.R., D.W. and Z.Y. performed the experiments. With the help of X.Y., J.D., S.Z., S.B. and R.W., P.L. performed bioinformatics analysis. X.Z., F.T., G.C., J.Z., P.L., M.C., Y.H. and C.W. wrote the manuscript with the help from all the authors.

## Competing interests

The authors declare no competing interests.

## Additional information

**Peer review information** *Nature Communications* thanks Tin-Lap Lee and the other anonymous 2 reviewer(s) for their contribution to the peer review this work. Peer reviewer reports are available.

