## [Peer Review File · Nature Communications]

Reviewers' Comments:

Reviewer #1:

Remarks to the Author:

In mammals, the developmental process of germ cells is complex and delicate, involving multiple key inflection points in cell fate determination. With the development of single-cell sequencing technology, although some studies have conducted transcriptome analysis on the development of male germ cells, there is still no exploration of genes related to cell fate determination. Due to the limitation of samples, there are relatively few studies on the development of human germ cells. This study uses mice as an animal model and an improved single-cell sequencing method to analyze transcriptomes of 11,598 cells (from E 6.5 to adult). It not only described the gene regulatory network from primordial germ cell (PGC) specification to spermatogenesis in great detail, but also studied the biological function and mechanism of the key candidate genes (Notch signaling pathway and *Helq*) of the determinant point of fate from mitotic arrest PGCs to post-arrest PGCs, which was termed in this research. Moreover, they performed human-mouse comparison at the PGC and spermatogonia stages and identified a set of conserved regulators involved in germ cell development of both human and mouse. In addition, through the integrated analysis of clinical azoospermia patients and germ cell tumor data and single-cell transcriptome data, further intersection analysis of genes at each stage was carried out. This work offers a transcriptome atlas of male germline cycle and may help understand the cell-fate transition landscape of male germ cells systematically.

Here I have some comments for this study and the carefully revisions are required for improving the quality of this paper.

Major comments:

1. The title of the study is "Cell-fate transition and determination analysis of mouse male germ cells throughout the whole developmental program". However, the article is mainly written on SPG transition and from mitotic arrest PGCs to post-arrest PGC during PGC development. Should the conversion from spermatogonia (SPG) to spermatocytes (SPC) and the conversion process from SPC to round spermatids (RS) be briefly described and analyzed?

2. There were two articles on transcriptome sequencing of spermatogenesis of human, mouse, and non-human primates (1,2). Compared with them, how innovative is your article? Can the post-arrest PGCs, Q-ProSPG, and T-ProSPG mentioned in your article be reproduced based on the integration of the data from your research and them?

(1). Hermann BP, Cheng K, Singh A, Roa-De La Cruz L, Mutoji KN, Chen IC, Gildersleeve H, Lehle JD, Mayo M, Westernströer B, Law NC, Oatley MJ, Velte EK, Niedenberger BA, Fritze D, Silber S, Geyer CB, Oatley JM, McCarrey JR. The Mammalian Spermatogenesis Single-Cell Transcriptome, from Spermatogonial Stem Cells to Spermatids. *Cell Rep.* 2018 Nov 6;25(6):1650-1667.e8. doi: 10.1016/j.celrep.2018.10.026. PMID: 30404016; PMCID: PMC6384825.

(2). Shami AN, Zheng X, Munyoki SK, Ma Q, Manske GL, Green CD, Sukhwani M, Orwig KE, Li JZ, Hammoud SS. Single-Cell RNA Sequencing of Human, Macaque, and Mouse Testes Uncovers Conserved and Divergent Features of Mammalian Spermatogenesis. *Dev Cell.* 2020 Aug 24;54(4):529-547.e12. doi: 10.1016/j.devcel.2020.05.010. Epub 2020 Jun 5. PMID: 32504559; PMCID: PMC7879256.

3. The spermatocytes in C7 were very heterogeneous (Fig. 1c), especially the group on the right, which only have R1 cells and not R2-4 (Supp. Fig. 1b). Please briefly describe the reason for this heterogeneity in the discussion or result part.

4. In page 7, Line 7-9, Please describe the method for determining the cell cycle in detail in Fig. 2b legend or in the method. It's confused that you said mitotic arrest PGCs, Q-ProSPG and spermatids were quiescent in Fig.2a, while in Fig.2b showed these three cell clusters were actually active? In order to avoid this doubt, please mark the Bar value [from 0 (blue) to 1 (red)] in the legend clearly as the ratio of cells rather than the change in expression level (Fig.2b).

5. In Figure 3e, the authors found that there were thirteen modules of DEGs, while the range of the ordinate is quite small, from -0.4 to 0.4. Was the trend significant? Did the three categories really make sense?

6. In page 9, Line 11-15, the conclusion "it was undergoing epigenetic reprogramming for the formation of SSC pool" was put forward by GO analysis. More evidences are needed to illustrate the epigenetic reprogramming process.

7. In page 10, Line 23, "a gradual increase of *Hes1*Bright/*Hes1*Dim cell ratio was observed from

E13.5 to E16.5". What is the measurement standard of Bright and Dim?

8.The inhibitory effect of DAPT on the Notch signaling pathway should have a certain dose effect. In the article, the author directly used the concentration of 200 mg/kg if there are other side effects on the spermatogenic cells of the testis? According to the literature, DAPT has a more approved concentration of 100 mg/kg. How did the author determine this concentration?

9.The author identified several misregulated mitotic to mitotic arrest transition-related genes, according to the change of expression levels between mitotic PGCs and post-arrest PGCs. However, the DEGs you found may not be caused by Helq knockout. It may be the heterogeneity caused by the small number of cells (there were only dozens of cells from WT), not the actual difference. Would you provide some functional verification?

10.In Figure 5f, WT mice only had mitotic PGCs in E12.5, but Helq^{-/-} mice had mitotic PGCs and transitional PGCs. The pseudo-time trajectory also showed that mitotic PGCs will develop into transitional PGCs, so I think the development was actually advanced, not delayed. The same was to E13.5 and E14.5. If only based on the presence of Post-arrest PGC in the WT at E15.5 days and the KO group does not have this cell type, then it cannot be expressed as delayed development but developmental arrest after Helq knockout. Need to re-describe this part of the results.

11.In Fig.7a, human PGC development was not continuous. Obviously, its sampling time was very close. whether the development of human PGC is really discontinuous, or it is because the analysis method is not suitable.

Minor comments:

1.In page 11, Line 1, the chemical name that can be written when DAPT (N-[N-(3,5-difluorophenacetyl-L-alanyl)]-S-phenylglycine t-butyl ester ?) first appears.

2.In page 12, Line 6, whether the name of mouse protein should be capitalized (HELQ)? please refer to the international standard for clarification.

3.In page 12, Line 30, the proportion of S population described in the article is 47.84%, and the proportion shown in the figure is 48.62%. Which data shall prevail?

4.In page 12, Line 14, the term SSC has already appeared before (page 9, Line 15), and the acronym [spermatogonial stem cells (SSCs)] should be clearly marked there.

5.In page 15, Line 14-16, Please describe exactly whether the Control in Fig. 6m is a group with only lentivirus or nothing added.

6.The order of the author's figures is quite chaotic, which may cause inconvenience to read, so it is suggested to improve it.

Reviewer #2:

Remarks to the Author:

The authors are the first to perform scRNA-seq capturing the germ cell development from embryo to adult in mice. In this study, they identified a significant number of new markers at different developmental stages. Moreover, they validated some of the targets by knockout animal model, immunostaining and other in vitro and in vivo experiments, which makes the results more convincing. For example, they demonstrated the critical role of Notch signaling for mitotic arrest in PGCs. Furthermore, they discovered a new subpopulation "transitional" PGCs. Lastly, they integrated the mouse scRNA-seq data with previous human scRNA-seq data to identify conserved regulators of the two species. Altogether, the high-resolution data serve as a very useful resource for future germ cell investigation.

Despite the well-known role of Gfra1 in SSC maintenance, Gfra1 expression has not been reported in round spermatids. Therefore, the authors could consider to provide more evidence at mRNA or protein level to validate the finding from scRNA-seq.

Although the author might be the first to demonstrate that the Notch signaling pathway was intrinsically activated in germ cells for the first time. It should be noted that Hes1/Notch have been proved to express in mouse spermatogonia by immunostaining and qRT-PCR (PMID: 29161703). Also, the activation of Hes1 and Notch signaling in PGCs has been reported in chicken (PMID: 29951200).

Fig. 3c showed a clear developmental trajectory. However, there were several small clusters of cells from E17.5, E18.5 and P0 locating away from the main trajectory. Do they represent different cell states? The authors could consider exploring more about this observation. In this regard, I would suggest the authors perform single cell RNA velocity analysis to see if it can add additional insights.

Fig. 6a showed that Hhex was upregulated at both T-proSPG and Undiff.ed SPG stages, suggesting it may have a role in the transition. The authors used the culture derived from PND5.5 testis to examine the function of Hhex, which did not test this aspect directly. The authors should consider using culture derived from a corresponding earlier stage or at least discuss this limitation.

The authors could consider including UMAP plots showing gene expression and cell annotation on their website.

Minor:

Supp fig. 1b - some clusters distinct from the main trajectory only present in 1 replicate

Fig. 2d appears before Fig. 2a-c

Page 7 line 28 and 30 - "IRF1" for protein

The appearance of figures of supp fig. 4 in text is not in order

Fig 3e - "signature"

Page 10 line 19 - "HES1"

All proteins should be in capital letter for mouse

Page 11 line 20 - Supplementary Data 2

Fig. 5c appears after Fig. 5f

In Fig. 5e and Supplementary Fig. 6f, the developmental delay in Helq null PGCs during E15.5 and E18.5 is obvious. However, it seems that in E12.5 and E13.5, the WT PGCs showed a delay compared with Helq null PGCs.

Page 12 line 26 - "CellCycleSocring"

The order of Supplementary Fig. 7a and 7b is reversed

Fig. 7l should be Fig. 7k in text and figures

Reviewer #3:

Remarks to the Author:

In this study, Zhao et al. performed single-cell RNA-seq analysis of testicular cells at 28 time-points throughout mouse germline development. Their data provide an integrated resource for understanding mouse germ cell development. Their analyses validate previous studies, as well as uncover some new gene markers and signaling pathways potentially involving in mouse germline development. They claim to have identified a new transitional germ cell stage between the mitotically active PGC and arrested pro-spermatogonia (ProSPG) stages. They provide evidence that the Notch signaling pathway promotes the transition of mitotic PGCs to mitotically arrested ProSPG. They also provide evidence that HELQ, a DNA damage-associated cell cycle regulator, drives the transition of mitotic PGCs to mitotically arrested ProSPG. Finally, they combine their mouse scRNA-seq datasets with analogous previously published human datasets to identify conserved gene regulatory networks.

Overall, this study provides a good integrated resource for the field. However, as explained below, novelty appears to be weak. In addition, there are several other concerns that need to be addressed.

Major concerns:

1) While the authors examined some time points not previously examined by scRNA-seq, most of their scRNA-seq analyses are focused on known cell populations previously identified by published single cell RNA-seq analyses. While they claimed some "new" subsets, those are not convincing or were previously defined by others (see below).

2) The new identified "transitional PGC" subset is not convincing. Based on Fig. 3c, it's hard to believe this transitional PGC is a real cell subpopulation, as it does not form a clearly defined "cell cluster." Following this logic, it would appear that cells can be divided into as many as stages as the authors like. To make this claim, the authors need to provide more evidence that this is a new identified cell subpopulation, as well as provide some evidence for their functional roles. If not, related statements need to be corrected.

3) The novelty of the identification of the role of Notch in fetal germ cells is limited. First, Notch signaling has previously been reported to be active in developing mouse testes, both in germ cells and Sertoli cells (PMID: 18801836). In addition, roles of the Notch pathway in PGCs have previously been reported in chicken, *C. elegans*, *Xenopus*, sea urchin, and *Drosophila* (PMID: 29951200, PMID: 32008902, PMID: 20151992, PMID: 23533178, PMID: 20660750). None of these studies are cited in this study. Second, finding expression of NOTCH signaling components in a cell does not demonstrate that the NOTCH pathway is activate in this cell (PMID: 23907117).

Indeed, the detection of Notch1 and Hes1 expression in mouse PGCs here does not mean the Notch pathway is activated in PGCs. Third, the in vivo effects observed by the authors in response to the Notch inhibitor, DAPT, do not necessarily reflect a defect in Notch signaling in germ cells. Indeed, it was previously shown that activating the Notch pathway in Sertoli cells influences fetal germ cell quiescence (PMID: 23907117, PMID: 23391689). In addition, several studies have shown that the Notch pathway is active exclusively in Sertoli cells in the perinatal testis, and that activating Notch is essential for fetal germ cell quiescence (PMID: 23907117, PMID: 23391689).

4) The study of the role of Helq in fetal germ cells is a novel work in this study, and thus this reviewer encourages the authors to study this further. Currently, the authors only show the cellular changes after Helq ko. While the authors tested potential mechanisms (as shown in Fig. 5l), these are mainly following previous publications (PMID: 24005329 and PMID: 24005041). Some further work can be done. For example, the authors could go deeper into how Helq influences fetal germ cell development. Currently, the authors stated that Helq depletion causes delayed development of fetal PGCs. However, based on the cell composition (Fig. 5f), another possibility is that Helq depletion impacts PGC growth or maintenance. This is supported by the fact that there are less mitotic PGCs at E13.5 and E14.5 in Helq ko mice compared to WT mice. In addition, their finding that apoptosis is upregulated in Helq^{-/-} PGCs also supports this hypothesis. The authors could go more deeply into other phenotypic defects. In addition, the authors could identify candidate HELQ-regulated genes from their scRNA-seq data and test their roles.

5) The scRNA-seq analysis of Helq^{-/-} germ cells has only one biological replicate; at least one more replicate should be provided for rigor. This is important, as the authors find opposite effects at two stages of development (Figs. 5e and S6f), which could be due to sampling bias. Even if their finding is correct, their interpretation that inhibited development at one stage causes an accumulation of germ cells at later time point, is not necessarily correct. Another possibility is that loss of Helq inhibits germ cell maturation at one stage and promotes germ cell events later. These two possibilities should be distinguished or at least acknowledged.

6) More data analyses should be done to identify conserved gene regulatory networks between humans and mice. The authors integrated their own previously published datasets with that of Wang et al., but there are several resources in the field, e.g., PMID: 30726734, PMID: 31928944, and PMID: 33453151. While these datasets are from different platforms, the authors should at least compare their enriched genes/signaling pathways to validate their own findings.

7) Fig. 5c. Is the % at E13.5 and E14.5 really significantly changed? With only 4 samples, it is hard to believe that there is a significant difference. Please also check the statistical test in Fig. 5h.

Minor concerns:

8) The authors often incorrectly refer to ProSPG as PGCs. PGCs convert into M ProSPG at ~E11.5-E13.5 (PMID: 23843236). Thus, what they call "mitotic PGCs" during this period should contain both PGCs and M-ProSPG. What they refer to as "Arrest PGCs" are typically called T1-ProSPG (PMID: 23843236). The authors also refer to other ProSG stages that they call "Q- and T-ProSPG,"

without defining them. If they want to use this system, they should not only define them, but refer to a publication where this system is used. It is suggested to use the M-, T1-, and T2-ProSPG nomenclature instead. A final note: further confusion is rendered by the fact that the authors refer to gonocytes in the Introduction and never explain that gonocytes are the same as ProSPG.

9) Pg 9, line 11. This "post-arrest PGC" has been identified by Law et al. 2019. The authors should perform a bioinformatic comparison to test their similarities.

10) Potential doublets contamination? Pg 7, line 20. It is surprising that the authors found that round spermatids (RS) express pluripotency and SPG markers, given that RS are post-meiotic and certainly not pluripotent. One explanation is this is an artifact, and that these cells are actually doublets. The authors should investigate this possibility.

11) The authors refer to "transcriptional" regulation throughout the manuscript when they presumably instead mean "transcriptome." Changes in mRNA levels as detected by RNAseq or scRNA-seq could be due to either alterations or transcription rate or RNA half-life, not necessarily the former. Thus, the authors are incorrect when they state that changes detected by these methods are transcriptional in the Abstract ("transcriptional reconfiguration" and "transcriptional reprogramming" should be revised), on pgs 5, top of pg 7, etc. Search and replace should be conducted throughout the entire MS.

12) Protein acronyms should be all caps and not italicized; e.g., HELQ, not Helq.

13) Pg 29, line 21. The ref 80 has no mSSC culture method.

Point-by-point response to the Reviewers' questions:

Below we have listed the point-by-point response to each of the questions, with the answers highlighted in blue.

Reviewer #1 (Remarks to the Author):

In mammals, the developmental process of germ cells is complex and delicate, involving multiple key inflection points in cell fate determination. With the development of single-cell sequencing technology, although some studies have conducted transcriptome analysis on the development of male germ cells, there is still no exploration of genes related to cell fate determination. Due to the limitation of samples, there are relatively few studies on the development of human germ cells.

This study uses mice as an animal model and an improved single-cell sequencing method to analyze transcriptomes of 11,598 cells (from E 6.5 to adult). It not only described the gene regulatory network from primordial germ cell (PGC) specification to spermatogenesis in great detail, but also studied the biological function and mechanism of the key candidate genes (Notch signaling pathway and Helq) of the determinant point of fate from mitotic arrest PGCs to post-arrest PGCs, which was termed in this research. Moreover, they performed human-mouse comparison at the PGC and spermatogonia stages and identified a set of conserved regulators involved in germ cell development of both human and mouse. In addition, through the integrated analysis of clinical azoospermia patients and germ cell tumor data and single-cell transcriptome data, further intersection analysis of genes at each stage was carried out. This work offers a transcriptome atlas of male germline cycle and may help understand the cell-fate transition landscape of male germ cells systematically.

Here I have some comments for this study and the carefully revisions are required for improving the quality of this paper.

Answer: We thank the reviewer for these positive comments.

Major comments:

1. The title of the study is "Cell-fate transition and determination analysis of mouse male germ

cells throughout the whole developmental program". However, the article is mainly written on SPG transition and from mitotic arrest PGCs to post-arrest PGC during PGC development. Should the conversion from spermatogonia (SPG) to spermatocytes (SPC) and the conversion process from SPC to round spermatids (RS) be briefly described and analyzed?

Answer: We appreciated the reviewer's suggestions and agree that more analyses should be included in this whole developmental program. To this aim, we have analyzed the global differentially expressed genes (DEGs) and gene ontology (GO) terms during spermatogenesis. As expected, "Cell cycle", "Gamete generation", "Synaptonemal complex assembly", "Cilium organization", and "Sperm axoneme assembly" were significantly enriched in different stages of spermatogenic cells (Rebuttal Fig. R1a, see also Supplementary Fig. 13a of the revised Manuscript). Of note, we found that transcriptional inactivation and reactivation occurred on sex chromosomes during the conversion processes from SPG to SPC, and from SPC to RS, respectively (Rebuttal Fig. R1b, see also Supplementary Fig. 13b of the revised Manuscript). The minimum transcriptional state on sex chromosomes was observed in SPC and the gene expression levels gradually increased, reaching chrX+Y/chrA ratios comparable to SPG in RS (Rebuttal Fig. R1b, see also Supplementary Fig. 13b of the revised Manuscript). Meanwhile, DEG analysis and GO analysis of each type of spermatogenic cells with the former- and later-stage germ cells were conducted (Supplementary Data 4 of the revised Manuscript).

Overall, these results characterized a dynamic developmental and cellular sequence of events during the conversion processes from SPG to SPC, and further to RS. And the main findings of this part were essentially consistent with previous studies^{1,2}.

Rebuttal Fig. R1. Transcriptome dynamics from SPG to SPC and further to RS. a, (Left) Heatmap of the top 50 DEGs in 12 cell types of mouse male germ cell development. The color key from blue to red indicates low to high gene expression levels. L, leptotene; Z, zygotene; P, pachytene; D, diplotene; MI, metaphase I; RS, round spermatids. **(Right)** Bubble plot shows top enriched GO terms of 12 cell type of mouse male germ cell development. The size and color of bubbles represent the number of genes and gene ratio in corresponding GO terms, respectively. **b, Boxplot** showing the ratio of average expression level of selected genes between two classes of chromosomes. Yellow represents the ratio of gene expression level between sex chromosomes and chromosome 2. Prasinous represents the ratio of gene expression level between sex chromosomes and chromosome 9. Rose represents the ratio of gene expression level between sex chromosomes and autosome.

2. There were two articles on transcriptome sequencing of spermatogenesis of human, mouse, and non-human primates (1,2). Compared with them, how innovative is your article? Can the post-arrest PGCs, Q-ProSPG, and T-ProSPG mentioned in your article be reproduced based on the integration of the data from your research and them?

(1). Hermann BP, Cheng K, Singh A, Roa-De La Cruz L, Mutoji KN, Chen IC, Gildersleeve H, Lehle JD, Mayo M, Westernströer B, Law NC, Oatley MJ, Velte EK, Niedenberger BA, Fritze D, Silber S, Geyer CB, Oatley JM, McCarrey JR. *The Mammalian Spermatogenesis Single-Cell Transcriptome, from Spermatogonial Stem Cells to Spermatids*. *Cell Rep*. 2018 Nov 6;25(6):1650-1667.e8. doi: 10.1016/j.celrep.2018.10.026. PMID: 30404016; PMCID: PMC6384825.

(2). Shami AN, Zheng X, Munyoki SK, Ma Q, Manske GL, Green CD, Sukhwani M, Orwig KE, Li JZ, Hammoud SS. *Single-Cell RNA Sequencing of Human, Macaque, and Mouse Testes Uncovers Conserved and Divergent Features of Mammalian Spermatogenesis*. *Dev Cell*. 2020 Aug 24;54(4):529-547.e12. doi: 10.1016/j.devcel.2020.05.010. Epub 2020 Jun 5. PMID: 32504559; PMCID: PMC7879256.

Answer: We thank the reviewer for the comments. Our study makes essential progress in the field of germ cell development. Firstly, although precious studies have illuminated specific stages of germ cell development¹⁻¹⁰, none of them monitored a full-term program of germ cell development in a single study. It is known that due to the batch effects which caused by different sequencing technologies, different library preparation protocols, and different data interpretation strategies, the integrated analyses of data from different platforms or labs was still technically challenging. It is therefore of profound significance to establish high-quality full-term transcriptome atlas of mouse male germ cell development at single-cell resolution. It would be convenient to know the dynamic expression patterns of any candidate genes in an accurate and consistent way. In addition, our dataset should be a high-quality comprehensive reference for comparison to inform the accurate cell type-specific transcriptional changes when any gene of interest is manipulated in male mouse germline.

Secondly, compared with 10×Genomics method, our single-cell single-tube amplification library construction method provides much higher gene detection rate (about 9,413 genes in each cell), and higher sensitivity in capturing RNA molecules (reviewed recently by directly comparing seven methods for single-cell and/or single-nucleus profiling)¹¹, which can dig out more lowly-expressed functional genes such as transcription factors. Our method also showed much lower cross-contaminations between different individual cells due to its single-tube amplification library construction strategy.

Lastly, how the cell-fate transition and determination of male germ cells is regulated remain largely unexplored, especially during prenatal male germ cell development. Based on the unbiased single-cell transcriptome reference established in the present study, we uncovered that Notch signaling pathway and HELQ played critical roles in the proper cell-fate transition from mitotic to mitotic arrest PGCs to ensure male fertility. We believe that our data would offer novel insights into this field and provide new clues for further study on male germ cell development.

For the second question that “*Can the post-arrest PGCs, Q-ProSPG, and T-ProSPG mentioned in your article be reproduced based on the integration of the data from your research and them?*”

According to the suggestion of the reviewer, we have tried to validate the robustness of our data by integrated analysis with these two publicly available scRNA-seq data of germ cells^{5,10}.

Firstly, by mapping the purified P6 (Postnatal day 6) and adult Id4-GFP positive SPG onto scRNA-seq based developmental trajectory constituted by the cells from post-arrest PGCs to diff.ed SPG, we found that the distribution of P6 and adult Id4-GFP positive SPG was generally merged with the corresponding cell clusters ranged from T-ProSPG to Diff.ing SPG, which confirmed our cell cluster assignment results (Rebuttal Fig. R2a, see also Supplementary Fig. 11a of the revised Manuscript). Further re-clustering results showed that T-ProSPG population could be further divided into two subpopulations, one of which was identified as “I-ProSPG

(Intermediate ProSPG)”, which largely lacked the expression of cell cycle genes and exhibited the migratory properties compared with T-ProSPG, which was consistent with a previous study⁹ (Rebuttal Fig. R2b-d, see also Supplementary Fig. 11b-d of the revised Manuscript).

Secondly, we obtained similar results by integrating our data and a previous study⁵ (Rebuttal Fig. R3, see also Supplementary Fig. 12 of the revised Manuscript) (PMID: 32504559, whose raw data were generated in their previously study⁶ (PMID: 30146481)). As a result, post-arrest PGCs, Q-ProSPG, and T-ProSPG were still mapped at the expected positions along the developmental trajectory reconstructed by no matter all spermatogenic cells or the cells including only late-stage PGCs, ProSPG, and SPG (Rebuttal Fig. R3a, b, see also Supplementary Fig. 11a, b of the revised Manuscript).

Overall, we provide an accurate, high-quality and more importantly, robust single-cell transcriptome atlas of male germline cycle, which provided the opportunity to uncover how the cell-fate transition and determination were regulated during male germ cell development in an unbiased way.

Rebuttal Fig. R2. Integrated analysis of the datasets of this study and a previous study¹⁰ (PMID: 30404016). a, UMAP plots showing integration results of post-arrest PGCs, Q-ProSPG, T-ProSPG, undiff.ed SPG, diff.ing SPG, and diff.ed SPG from this paper and 578 Id4-

GFP⁺ spermatogonia from Hermann *et al*¹⁰. Cells are colored based on cell types (left), data sources (middle), and time points (right) are shown. **b**, 3D PCA plots showing integration results of post-arrest PGCs, Q-ProSPG, T-ProSPG from this paper and 578 Id4-GFP⁺ spermatogonia from Hermann *et al*¹⁰. Cells are colored based on clusters (left), cell types (upper right), and time points (lower right) are shown. **c**, 3D PCA plots of *Dnmt3l*, *Tie1*, *Zbtb16*, and *Gfra1*. The color key from gray to blue indicates low to high expression levels. **d**, Violin plots showing expression levels of cell proliferation-related markers, actin cytoskeleton organization genes, and cell migration-related markers in each cluster of integration results between this study and Hermann *et al*¹⁰.

Rebuttal Fig. R3. Integrated analysis of the datasets of this study and a previous study⁶ (PMID: 30146481). a, UMAP plots showing integration results of germ cells from this study and Green *et al*⁶. Cells are colored based on cell types (left) and references (right). b, UMAP

plots showing integration results of germ cells from this study and Green *et al*⁶. Cells are colored based on cell types (left), clusters (middle), and time points (right). **c**, UMAP plots of germ cell markers. The color key from gray to blue indicates low to high expression levels.

3. *The spermatocytes in C7 were very heterogeneous (Fig. 1c), especially the group on the right, which only have R1 cells and not R2-4 (Supp. Fig. 1b). Please briefly describe the reason for this heterogeneity in the discussion or result part.*

Answer: Thank the reviewer for this question. We have checked our data carefully and re-analyzed cells in C7 by re-clustering (Rebuttal Fig. R4a, already shown in the original manuscript, see also Supplementary Fig. 1h of the revised Manuscript), and the reconstruction of developmental trajectory using Monocle (Rebuttal Fig. R4b). Our results showed that germ cells in C7 were developed along the ordered developmental trajectory. And the cells from Repeat 1 exhibited similar molecular characteristics like other cells of the same cell cluster (Rebuttal Fig. R4c, d). Based on these results, the certain heterogeneity of C7 does not affect our conclusions and the subsequent analysis.

Rebuttal Fig. R4. Reconstruction of the developmental trajectory by re-clustering analysis. a, 3D PCA plots of 12 cell types of mouse male germ cell development. Cells are colored based on cell types (left), replicates (middle), and time-points (right). undiff.ed SPG, undifferentiated spermatogonia; diff.ing SPG, differentiating spermatogonia; diff.ed SPG, differentiated spermatogonia; L, leptotene; Z, zygotene; P, pachytene; D, diplotene; MI, metaphase I; RS, round spermatids. **b**, The inferred trajectory of 11 cell types during mouse

male germ cell development by Monocle analysis. Cells are colored based on cell types (left), replicates (middle), and time-points (right). **c**, Heatmap showing expression levels of diff.ed SPG marker genes scaled among 18 germ cell types. **d**, Dotplot showing average expression levels for genes (expressed in at least 3 cells) between C7 diff.ed SPG-1 and C7 diff.ed SPG-2 (distinct from main cluster), respectively. The Pearson's correlation coefficient is 0.93.

4. In page 7, Line 7-9, Please describe the method for determining the cell cycle in detail in Fig. 2b legend or in the method. It's confused that you said mitotic arrest PGCs, Q-ProSPG and spermatids were quiescent in Fig.2a, while in Fig.2b showed these three cell clusters were actually active? In order to avoid this doubt, please mark the Bar value [from 0 (blue) to 1 (red)] in the legend clearly as the ratio of cells rather than the change in expression level (Fig.2b).

Answer: Thank the reviewer for this question. We are sorry that the original manuscript might somehow lead to misunderstanding and confusion due to lacking of essential description of Bar value and more details for determining the cell cycle. We have revised them in the revised manuscript accordingly (see Fig. 1c of the revised Manuscript).

5. In Figure 3e, the authors found that there were thirteen modules of DEGs, while the range of the ordinate is quite small, from -0.4 to 0.4. Was the trend significant? Did the three categories really make sense?

Answer: Thank the reviewer for this question. These genes were all obtained based on the DEG analysis (Fold change ≥ 1.5 , $P < 0.01$), and we scaled the expression level ($\log(\text{TPM}/10 + 1)$) of each gene among these cells (Mitotic PGCs, Transitional PGCs, Mitotic arrest PGCs, and Post-arrest PGCs), and calculated the average value within each cell type. Also, we have made a schematic diagram of gene classification to make it easier for the readers to understand (Rebuttal Fig. R5, see Supplementary Fig. 5d of the revised Manuscript).

Rebuttal Fig. R5. Schematic diagram of clustering analysis of dynamic gene expression during the mitotic to mitotic arrest transition.

6. In page 9, Line 11-15, the conclusion “it was undergoing epigenetic reprogramming for the formation of SSC pool” was put forward by GO analysis. More evidences are needed to illustrate the epigenetic reprogramming process.

Answer: We thank the reviewer for this question. To provide more evidences about the supposed epigenetic reprogramming process, immunostaining of 5mC was performed from E12.5 to E17.5 (Rebuttal Fig. R6, see also Supplementary Fig. 5b). The results showed that there was a significant accumulation of DNA methylation during the development of early gonadal germ cells into post-arrest PGCs. This finding confirmed our GO analysis result, in which the genes related to DNA methylation were highly expressed in post-arrest PGCs.

Rebuttal Fig. R6. Immunofluorescence of 5mC (5-Methylcytosine) co-stained with DDX4 in E12.5, E14.5, E15.5, and E17.5 mouse male gonads. Hollow and solid white arrowheads indicate 5mC^{Negative} and 5mC^{Positive} subtypes in DDX4⁺ cells, respectively. Scale bar, 10 μ m.

7. In page 10, Line 23, “a gradual increase of *Hes1*^{Bright}/*Hes1*^{Dim} cell ratio was observed from E13.5 to E16.5”. What is the measurement standard of Bright and Dim?

Answer: Firstly, confocal images were obtained in the same condition (the same laser intensity and detector gain). Then, the HES1 (GFP)^{Bright} and HES1 (GFP)^{Dim} cells were manually distinguished (Rebuttal Fig. R7a, see also Fig. 4d of the revised Manuscript) and quantified by at least two individuals. Finally, the relative protein expression level (which is indicated by the fluorescence intensity of the secondary antibody) was calculated using the ImageJ software, and statistical analysis verified that there was indeed a significant difference between the high (Bright) and low (Dim) groups (Rebuttal Fig. R7b).

Rebuttal Fig. R7. Image analysis of immunofluorescence results of HES1. a, Proportion of

HES1^{Dim} and HES1^{Bright} subtypes in DDX4⁺ cells in E13.5-E16.5 mouse male gonads. **b**, Quantification of fluorescence intensity of the secondary antibody in HES1^{Dim} and HES1^{Bright} subtypes. Mean \pm SEM, **** $P < 0.0001$, unpaired two-tailed t test.

8. The inhibitory effect of DAPT on the Notch signaling pathway should have a certain dose effect. In the article, the author directly used the concentration of 200 mg/kg if there are other side effects on the spermatogenic cells of the testis? According to the literature, DAPT has a more approved concentration of 100 mg/kg. How did the author determine this concentration?

Answer: We thank for this comment. As mentioned by the reviewer, DAPT has a more approved concentration of 100 mg/kg for oral administration. In fact, we used the concentration of 100 mg/kg by intraperitoneal injection and did observe a certain blocking of Notch signaling pathway; however, germ cell under this dose of treatment did not exhibit a clear phenotype (Rebuttal Fig. R8a-c and 8e-g). After increasing the dose to 200 mg/kg, a clear phenotype appeared, which was manifested by a significant blocking of cell cycle arrest at E15.5 and a SSC pool disturbance at PND0 (Rebuttal Fig. R8a-c and 8e-g, see also Fig. 6f-i of the revised Manuscript). As for side effects, we have observed a certain increase in the proportion of apoptosis using the concentration of 200 mg/kg by intraperitoneal injection (Rebuttal Fig. R8d and h). To further confirm this finding, another Notch signaling pathway inhibitor-LY411575 (a commercial chemical γ -Secretase inhibitor, N-[(1S)-2-[[[(7S)-6,7-dihydro-5-methyl-6-oxo-5H-dibenz[b,d]azepin-7-yl]amino]-1-methyl-2-oxoethyl]-3,5-difluoro- α S-hydroxy-benzeneacetamide, Selleck, S2714) was used to block Notch signaling pathway and we evaluated its function accordingly (Rebuttal Fig. R9a-f, see also Supplementary Fig. 6g-l of the revised Manuscript). Our results showed that the treatment by LY411575 could also result in a significant decrease of NOTCH1 and HES1 in germ cells, wherein the number of MKI67⁺ PGCs was significantly increased compared with the mock controls (Rebuttal Fig. R9c and 9f, see also Supplementary Fig. 6i and 6l of the revised Manuscript). Short-term treatment of the *in vitro* cultured mouse male PGCs by DAPT or LY411575 could also lead to the decreased expression of *Hes1*, demonstrating that PGCs could respond to the blocking of Notch signaling pathway properly (Rebuttal Fig. R9g, h, see also Supplementary Fig. 6e, f of the revised Manuscript). The results based on these two inhibitors illustrated that Notch signaling pathway

was indeed activated in gonadal germ cells, and played critical roles for the cell-fate transition of PGCs.

Rebuttal Fig. R8. The blocking of Notch signaling pathway *in vivo* using DAPT in concentration of 100 and 200 mg/kg. **a**, Immunofluorescence of NOTCH1 co-stained with DDX4 in DAPT-treatment (100 and 200 mg/kg)- and control mouse male gonads at E15.5. Scale bar, 10 μ m. **b**, Immunofluorescence of HES1 co-stained with DDX4 in DAPT-treatment (100 and 200 mg/kg)- and control mouse male gonads at E15.5. Scale bar, 10 μ m. **c**, Immunofluorescence of MKI67 co-stained with DDX4 and proportion of MKI67⁺ cells in

DDX4⁺ cells in DAPT-treatment (100 and 200 mg/kg)- and control mouse male gonads at E15.5. Solid yellow arrowheads indicate MKI67^{Positive} subtypes in DDX4⁺ cells. Scale bar, 10 μ m. **d**, Immunofluorescence of cPARP1 co-stained with DDX4 in DAPT-treatment (200 mg/kg)- and control mouse male gonads at E15.5. **e**, The quantification of relative fluorescence intensity of NOTCH1 in DAPT-treatment (100 and 200 mg/kg)- and control mouse male gonads at E15.5. Mean \pm SEM, n = 3 per group, **** $P < 0.0001$, unpaired two-tailed t test. **f**, The quantification of relative fluorescence intensity of HES1 in DAPT-treatment (100 and 200 mg/kg)- and control mouse male gonads at E15.5. Mean \pm SEM, n = 3 per group, **** $P < 0.0001$, unpaired two-tailed t test. **g**, Proportion of MKI67⁺ cells in DDX4⁺ cells in DAPT-treatment (100 and 200 mg/kg)- and control mouse male gonads at E15.5. Mean \pm SEM, n = 3 per group, ns, not significant, **** $P < 0.01$, unpaired two-tailed t test. **h**, Proportion of cPARP1⁺ cells in DDX4⁺ cells in DAPT-treatment (200 mg/kg)- and control mouse male gonads at E15.5. Mean \pm SEM, n = 3 per group, * $P < 0.05$, unpaired two-tailed t test

Rebuttal Fig. R9. The blocking of Notch signaling pathway *in vivo* by LY411575 (10 mg/kg) treatment and *in vitro* by treatment of DAPT (100 μM) or LY411575 (100 μM). a, Immunofluorescence of NOTCH1 co-stained with DDX4 in LY411575-treatment and control mouse male gonads at E15.5. Scale bar, 10 μm. b, Immunofluorescence of HES1 co-stained with DDX4 in LY411575-treatment and control mouse male gonads at E15.5. Scale bar, 10 μm. c, Immunofluorescence of MKI67 co-stained with DDX4 in LY411575-treatment and control mouse male gonads at E15.5. Solid yellow arrowheads indicate MKI67^{Positive} subtypes in DDX4⁺ cells. Scale bar, 10 μm. d, The quantification of relative fluorescence intensity of NOTCH1 in LY411575-treatment and control mouse male gonads at E15.5. Mean ± SEM, n = 3 per group, ** $P < 0.0001$, unpaired two-tailed t test. e, The quantification of relative fluorescence intensity of HES1 in LY411575-treatment and control mouse male gonads at**

E15.5. Mean \pm SEM, n = 3 per group, **** $P < 0.0001$, unpaired two-tailed t test. **f**, Proportion of MKI67⁺ cells in DDX4⁺ cells in LY411575-treatment and control mouse male gonads at E15.5. Mean \pm SEM, n = 3 per group, ** $P < 0.01$, unpaired two-tailed t test. **g**, Schematic showing the workflow of treatment by γ -Secretase inhibitor–DAPT or LY411575 for Notch signaling pathway blocking *in vitro*. The BVSC⁺ cells (PGCs) were cultured in a well of 5,000 cells in a low-cell-binding U-bottom 96-well plate in PGC medium (see Method in the revised Manuscript). **h**, Q-PCR analysis of the expression of PGCs marker genes (*Ddx4*, *Prdm1*, *Dppa3*) and Notch signaling pathway target gene *Hes1* in *in vitro* cultured PGCs after 24 h treated by DAPT or LY411575. To verify the germ cell identity, the sorted PGCs and somatic cells without treatment were used as the controls. Relative expression levels are shown with normalization to *Gapdh*. Error bars indicate mean \pm SEM from at least two independent biological replicates. ** $P < 0.01$, unpaired two-tailed t test.

9. The author identified several misregulated mitotic to mitotic arrest transition-related genes, according to the change of expression levels between mitotic PGCs and post-arrest PGCs. However, the DEGs you found may not be caused by *Helq* knockout. It may be the heterogeneity caused by the small number of cells (there were only dozens of cells from WT), not the actual difference. Would you provide some functional verification?

Answer: Thank the reviewer for this comment and suggestion. The cell number used for *Helq* knockout analysis was 652 (*Helq*^{-/-}) versus 825 (WT) during E11.5 and E15.5, which met the requirement for DEG analysis. And the mis-expression of mitotic to mitotic arrest transition-related genes (14% in all) might not be caused by the cell heterogeneity (Rebuttal Fig. R10a). As we know, *Helq* encodes a DNA helicase, which might regulate many downstream genes directly or indirectly. Thus, the DEGs revealed here should result from *Helq* knockout itself or further downstream effects.

Concerning on the novel role of HELQ in the regulation of germ cell development, we also tried to dissect certain interesting downstream target. In the DEG list when *Helq* was knocked out, we found that *Helq*^{-/-} germ cells highly expressed *Cdh1* (encoding E-cadherin) (Rebuttal Fig. R10b, see also Supplementary Fig. 9c of the revised Manuscript), which was gradually

downregulated in the WT control (involved in the module of “Loss” in Fig. 3e and Supplementary Data 2 of the revised Manuscript; Rebuttal Fig. R10c, see also Supplementary Fig. 9d of the revised Manuscript), and this downregulation trend was also verified by immunostaining of E-cadherin (Rebuttal Fig. R10d, see also Supplementary Fig. 9e of the revised Manuscript). Immunostaining results further showed the higher expression of E-cadherin protein in *Helq*^{-/-} germ cells compared with the WT control (Rebuttal Fig. R10e, see also Fig. 5l of the revised Manuscript). Moreover, GO analysis showed that *Cdh1* and other abnormally expressed genes were enriched in the processes such as “Apoptotic cleavage of cellular proteins”, “Apoptosis”, “Tight junction assembly”, “Hippo signaling pathway”, “Pathways in cancer”, and “Negative regulation of cell differentiation” (Supplementary Data 3 of the revised Manuscript), which were highly related to the phenotypes of *Helq*^{-/-} germ cells as we observed. Consistent with previous studies, the expression of E-cadherin (encoded by *Cdh1*) in male germ cells should be downregulated once sexual differentiation completed, which was critical for the differentiation of prenatal male germ cells^{12,13}. Based on these data, it is most likely that HELQ also functions in the regulation of PGC differentiation.

Rebuttal Fig. R10. *Helq* knockout caused abnormal maintenance of high expression level of *Cdh1* (E-cadherin). **a**, Statistics (top) and pie charts (bottom) of mitotic to mitotic arrest transition-related genes in *Helq*^{-/-} germ cells. **b**, Histograms of *Cdh1* expression in WT- and *Helq*^{-/-} germ cells, ns, not significant, ns, not significant, *** $P < 0.001$ and **** $P < 0.0001$, unpaired two-tailed t test. **c**, Violin plots showing the expression level of *Cdh1* at different stages in WT mouse male germ cells. **d**, (Top) Immunofluorescence of E-cadherin co-stained with MKI67 and BLIMP1 (GFP) in WT mouse male gonads at E13.5, E14.5 and E15.5. Scale bar, 10 μm . (Bottom) The quantification of relative fluorescence intensity of E-cadherin in WT mouse male germ cells at E13.5, E14.5, and E15.5. Mean \pm SEM, ns, not significant, **** $P < 0.0001$, unpaired two-tailed t test. **e**, Immunofluorescence of E-cadherin co-stained with DDX4 and the quantification of relative fluorescence intensity in the WT- and *Helq*^{-/-} mouse male gonads at E14.5. Scale bar, 10 μm . Mean \pm SEM, $n = 4$ per group, ** $P < 0.01$, unpaired two-

tailed t test.

10. In Figure 5f, WT mice only had mitotic PGCs in E12.5, but *Helq*^{-/-} mice had mitotic PGCs and transitional PGCs. The pseudo-time trajectory also showed that mitotic PGCs will develop into transitional PGCs, so I think the development was actually advanced, not delayed. The same was to E13.5 and E14.5. If only based on the presence of Post-arrest PGC in the WT at E15.5 days and the KO group does not have this cell type, then it cannot be expressed as delayed development but developmental arrest after *Helq* knockout. Need to re-describe this part of the results.

Answer: We thank for the reviewer for this suggestion. Our results demonstrated that *Helq* knockout resulted in the hyperactivation of p-ATR/p-CHK1 regulatory axis (Rebuttal Fig. R11a, b and d, e, see also Fig. 5e, f and h, i of the revised Manuscript), then might further cause the abnormal cell cycle switch of PGCs as exemplified by the emergence of the transitional PGCs at E12.5, which was traced by co-staining of MKI67 and WEE1 (Rebuttal Fig. R11c, f, see also Fig. 5g and j of the revised Manuscript). Therefore, the emergence of transitional PGCs at E12.5 when *Helq* was knocked out indicate that E12.5 *Helq*^{-/-} PGCs might be aberrantly enter the cell cycle arrest state by the aberrant hyperactivation of p-ATR/p-CHK1 regulatory axis, which exhibited as advanced development on the developmental trajectory. Although early-stage (E12.5) *Helq*^{-/-} PGCs was developmentally advanced than the WT control, the hyperactivation of p-ATR/p-CHK1 regulatory axis might result in the cell arrest for the prolonged cell cycle checkpoints at E15.5 as exemplified by the loss of post-arrest PGCs. Thus, we have re-described these results of this part in the revised manuscript.

Rebuttal Fig. R11. *Helq* knockout resulted in the hyperactivation of p-ATR/p-CHK1 regulatory axis. **a**, Immunofluorescence of p-ATR co-stained with DDX4 in the WT- and *Helq*^{-/-} mouse male gonads at E14.5. Scale bar, 10 μm. **b**, Immunofluorescence of p-CHK1 co-stained with MKI67 and BLIMP1 (GFP) in the WT- and *Helq*^{-/-} mouse male gonads at E14.5. Scale bar, 10 μm. **c**, Immunofluorescence of p-CHK1 co-stained with MKI67 and BLIMP1 (GFP) in the WT- and *Helq*^{-/-} mouse male gonads at E15.5. Scale bar, 10 μm. **d**, The quantification of relative fluorescence intensity of p-ATR in the WT- and *Helq*^{-/-} mouse male germ cells at E14.5. Mean ± SEM, n = 4 per group, **** $P < 0.0001$, unpaired two-tailed t test. **e**, Relative

proportions of of p-CHK1⁺ germ cells in the WT- and *Helq*^{-/-} mouse male germ cells at E14.5. Mean ± SEM, n = 4 per group, ** $P < 0.01$, unpaired two-tailed t test. **f**, Relative proportions of of p-CHK1⁺ germ cells in the WT- and *Helq*^{-/-} mouse male germ cells at E15.5. Mean ± SEM, n = 4 per group, * $P < 0.05$, unpaired two-tailed t test. **g**, (Top) Immunofluorescence of WEE1 co-stained with MKI67 and BLIMP1 (GFP) in WT- and *Helq*^{-/-} mouse male gonads at E14.5. Scale bar, 10 μm. (Bottom) Relative proportions of mitotic- (MKI67⁺ WEE1⁻), transitional- (MKI67⁺ WEE1⁺) and mitotic arrest PGCs (MKI67⁻ WEE1⁻) in the WT- versus *Helq*^{-/-} mouse male gonads at E14.5. Mean ± SEM, n = 4 per group, ns, not significant, * $P < 0.05$, unpaired two-tailed t test. **h**, Immunofluorescence of WEE1 co-stained with MKI67 and BLIMP1 (GFP) in the WT- and *Helq*^{-/-} mouse male gonads at E15.5. Scale bar, 10 μm.

11. In Fig. 7a, human PGC development was not continuous. Obviously, its sampling time was very close. whether the development of human PGC is really discontinuous, or it is because the analysis method is not suitable.

Answer: Thank the reviewer for this comment. PCA is a linear dimension reduction method that can preserve the global structures well, which has been widely used for the dimension reduction analysis. In fact, our previous study of human PGCs also presented a discontinuous developmental trajectory, which was reconstructed using both t-SNE and 3D PCA plots¹⁴ (Rebuttal Fig. R12). The discontinuity was probably caused by the sparse sampling time of human PGCs, which lacked the time points from 11 to 18 weeks. This sampling strategy might lead to the failed collection of certain transitional states between mitotic PGCs and mitotic arrest PGCs; by contrast, our study here was performed with much denser sampling points.

Rebuttal Fig. R12. Interspecies comparison of male PGCs development between human and mouse. a, PCA plots of four phases of mouse male PGCs and three phases of human male PGCs. Cells are colored based on the cell types are shown. **b,** PCA plots of four phases of mouse male PGCs and three phases of human male PGCs. Cells are colored based on the sampled time-points shown. **c,** t-SNE (left) and 3D PCA (right) plots of three phases of male PGCs from Li et al. (PMID: 28575695).

Minor comments:

1. In page 11, Line 1, the chemical name that can be written when DAPT (*N*-[*N*-(3,5-difluorophenacetyl-*l*-alanyl)]-*S*-phenylglycine *t*-butyl ester ?) first appears.

Answer: We thank for this suggestion. We have added the chemical name in the position when DAPT first appeared.

2. In page 12, Line 6, whether the name of mouse protein should be capitalized (*HELQ*)? please refer to the international standard for clarification.

Answer: We thank for this suggestion. We have checked all the names of mouse genes and proteins in our manuscript and modified them accordingly.

3. In page 12, Line 30, the proportion of S population described in the article is 47.84%, and the proportion shown in the figure is 48.62%. Which data shall prevail?

Answer: We are sorry for this error. We have checked the original data, and the proportion of S population was 47.84% as shown in the text. Accordingly, we have revised the number in the corresponding Figure. And the proportion of G1 and G2/M were correct. Our raw data of S phase in WT and *Helq*^{-/-} mESCs was shown below (powered by GraphPad Prism) (Rebuttal Fig. R13).

a

WT			Helq ^{-/-}		
G1	S	G2/M	G1	S	G2/M
38.6%	41.85%	19.55%	27.17%	48.82%	24.02%
41.14%	40.93%	17.93%	30.35%	49.5%	20.15%
48.69%	40.22%	11.09%	27.88%	46.44%	25.68%
			31.09%	46.6%	22.31%

b

t test			t test			t test		
1	Table Analyzed	G1 phase	1	Table Analyzed	S phase	1	Table Analyzed	G2/M phase
2			2			2		
3	Column B	Helq ^{-/-}	3	Column B	Helq ^{-/-}	3	Column B	Helq ^{-/-}
4	vs.	vs.	4	vs.	vs.	4	vs.	vs.
5	Column A	WT	5	Column A	WT	5	Column A	WT
6			6			6		
7	Unpaired t test		7	Unpaired t test		7	Unpaired t test	
8	P value	0.0043	8	P value	0.0010	8	P value	0.0453
9	P value summary	**	9	P value summary	**	9	P value summary	*
10	Significantly different? (P < 0.05)	Yes	10	Significantly different? (P < 0.05)	Yes	10	Significantly different? (P < 0.05)	Yes
11	One- or two-tailed P value?	Two-tailed	11	One- or two-tailed P value?	Two-tailed	11	One- or two-tailed P value?	Two-tailed
12	t, df	t=4.939 df=5	12	t, df	t=6.849 df=5	12	t, df	t=2.653 df=5
13			13			13		
14	How big is the difference?		14	How big is the difference?		14	How big is the difference?	
15	Mean ± SEM of column A	42.81 ± 3.030 N=3	15	Mean ± SEM of column A	41.00 ± 0.4718 N=3	15	Mean ± SEM of column A	16.19 ± 2.593 N=3
16	Mean ± SEM of column B	29.12 ± 0.9458 N=4	16	Mean ± SEM of column B	47.84 ± 0.7753 N=4	16	Mean ± SEM of column B	23.04 ± 1.184 N=4
17	Difference between means	-13.69 ± 2.771	17	Difference between means	6.840 ± 0.9987	17	Difference between means	6.850 ± 2.582
18	95% confidence interval	-20.81 to -6.564	18	95% confidence interval	4.273 to 9.407	18	95% confidence interval	0.2128 to 13.49
19	R square	0.8299	19	R square	0.9037	19	R square	0.5847

Rebuttal Fig. R13. Raw data for cell cycle analysis of the WT- and *Helq*^{-/-} mouse embryonic stem cells by flow cytometry. a, Cell cycle analysis of the WT- and *Helq*^{-/-} mouse embryonic stem cells by flow cytometry, 3 and 4 replicates, respectively. b, Statistical data of cell cycle analysis, * $P < 0.05$, ** $P < 0.01$, unpaired two-tailed t test.

4. In page 12, Line 14, the term SSC has already appeared before (page 9, Line 15), and the acronym [spermatogonial stem cells (SSCs)] should be clearly marked there.

Answer: Thank the reviewer for this suggestion. We have checked all the acronyms used in our paper and marked them in the position when they were first mentioned.

5. In page 15, Line 14-16, Please describe exactly whether the Control in Fig. 6m is a group with only lentivirus or nothing added.

Answer: Thank the reviewer for this suggestion. The control in this test is a group with no shRNA added, the sh-NC is a group with lentivirus carrying shRNA against no target, and sh-*Hhex-1* and sh-*Hhex-2* are groups with lentivirus carrying shRNA against *Hhex*. We have added these descriptions in the Method part, and oligonucleotides used in this study were listed in the Supplementary Data 7 of the revised Manuscript.

6. The order of the author's figures is quite chaotic, which may cause inconvenience to read, so it is suggested to improve it.

Answer: We thank the reviewer for the suggestion. We have fine-tuned the order of Figures and text to make them more logical, which will make it easier to understand.

Reviewer #2 (Remarks to the Author):

The authors are the first to perform scRNA-seq capturing the germ cell development from embryo to adult in mice. In this study, they identified a significant number of new markers at different developmental stages. Moreover, they validated some of the targets by knockout animal model, immunostaining and other in vitro and in vivo experiments, which makes the results more convincing. For example, they demonstrated the critical role of Notch signaling for mitotic arrest in PGCs. Furthermore, they discovered a new subpopulation “transitional” PGCs. Lastly, they integrated the mouse scRNA-seq data with previous human scRNA-seq data to identify conserved regulators of the two species. Altogether, the high-resolution data serve as a very useful resource for future germ cell investigation.

Answer: We thank the reviewer for these positive comments of our work.

2-1: Despite the well-known role of *Gfra1* in SSC maintenance, *Gfra1* expression has not been reported in round spermatids. Therefore, the authors could consider to provide more evidence at mRNA or protein level to validate the finding from scRNA-seq.

Answer: Thank the reviewer for this comment. We have validated the mRNA levels of these genes in round spermatids and spermatogonia as well as spermatocytes isolated from adult male *Oct4*-EGFP reporter mice. In brief, after testis dissociation and Hoechst staining, we sorted GFP-positive diploid, tetraploid (4N), and haploid (1N) cells, which were spermatogonia, spermatocytes and round spermatids, respectively. FACS-sorted populations were first confirmed by detecting the expression of well-known germ cell marker genes via q-PCR. Finally, we detected *Gfra1* expression in round spermatids, which was consistent with our scRNA-seq data (Rebuttal Fig. R14).

Rebuttal Fig. R14. Expression analysis of SPG-, SPC-, and RS marker genes by q-PCR of germ cells isolated from adult mouse testis. Relative expression levels are shown with normalization to *Gapdh*. Error bars indicate mean \pm SEM from at least two independent biological replicates. The purity of FACS-sorted SPG, SPC, and RS was verified by expression of known germ cell markers.

2-2: *Although the author might be the first to demonstrate that the Notch signaling pathway was intrinsically activated in germ cells for the first time. It should be noted that Hes1/Notch have been proved to express in mouse spermatogonia by immunostaining and qRT-PCR (PMID: 29161703). Also, the activation of Hes1 and Notch signaling in PGCs has been reported in chicken (PMID: 29951200).*

Answer: Thank the reviewer for this comment. As suggested by the reviewer, we have cited these references in the revised manuscript about the role of Notch signaling pathway in germ cell development, especially in non-mammalian animals such as chicken.

2-3: *Fig. 3c showed a clear developmental trajectory. However, there were several small clusters of cells from E17.5, E18.5 and P0 locating away from the main trajectory. Do they represent different cell states? The authors could consider exploring more about this observation. In this regard, I would suggest the authors perform single cell RNA velocity analysis to see if it can add additional insights.*

Answer: We thank the reviewer for these question and suggestion. Accordingly, we have re-analyzed our data by performing more analyses. Firstly, in our original dimension reduction and clustering results, all the Q-ProSPG were clustered as one cell cluster (C0) (Rebuttal Fig. R15a, b, see also Fig. 3c of the revised Manuscript), indicating that the heterogeneity of Q-ProSPG was negligible. Secondly, we found that there were only 22 and 63 genes, which were specifically expressed in these two sub-populations (C4 and C5) of post-arrest PGCs. In addition, C4 and C5 showed great similarity in terms of the expression of post-arrest PGCs' signature genes (Rebuttal Fig. R15c). Lastly, RNA velocity was also performed as the reviewer suggested (Rebuttal Fig. R15d, e), and we found that the directions of those cells locating away

from the main cluster were all point to the main developmental trajectory inferred by RNA velocity analysis. Based on these results, we think that the small cell clusters could be merged with the main clusters, which will not affect our findings and the obtained conclusions.

Rebuttal Fig. R15. Developmental trajectory of mouse male gonadal germ cells. a, UMAP (Uniform Manifold Approximation and Projection) plot of the mitotic PGCs, transitional PGCs, mitotic arrest PGCs, post-arrest PGCs, Q-ProSPG, and T-ProSPG. Cells are colored by indicated cell types. **b,** UMAP plot of the mitotic PGCs, transitional PGCs, mitotic arrest PGCs, post-arrest PGCs, Q-ProSPG, and T-ProSPG. Cell are colored by indicated automatically clustering results by the software. **c,** Heatmap showing expression levels of post-arrest PGCs markers scaled among the mitotic PGCs, transitional PGCs, mitotic arrest PGCs, C4 of post-arrest PGCs, C5 of post-arrest PGCs (away from the main trajectory), Q-ProSPG, and T-

ProSPG. **d**, UMAP projection of the mitotic PGCs, transitional PGCs, mitotic arrest PGCs, post-arrest PGCs, Q-ProSPG, and T-ProSPG, shown with velocity field. **e**, PCA projection of the mitotic PGCs, transitional PGCs, mitotic arrest PGCs, post-arrest PGCs, Q-ProSPG, and T-ProSPG, shown with velocity field.

2-4: *Fig. 6a showed that Hhex was upregulated at both T-proSPG and Undiff.ed SPG stages, suggesting it may have a role in the transition. The authors used the culture derived from PND5.5 testis to examine the function of Hhex, which did not test this aspect directly. The authors should consider using culture derived from a corresponding earlier stage or at least discuss this limitation.*

Answer: We thank for this comment. As far as we know, fetal germ cell are with low mitotic activity; thus, the *in vitro* culture of fetal germ cell is still very difficult. Although a previous study¹⁵ had reported that mouse male germ cells derived from E12.5 to E18.5 could be expanded *in vitro*, the offspring from the cultured cells showed growth abnormalities and were defective in genomic imprinting. As suggested by the reviewer, we have discussed this limitation in the Discussion part of the revised manuscript.

2-5: *The authors could consider including UMAP plots showing gene expression and cell annotation on their website.*

Answer: Thank for this suggestion. Accordingly, we have updated all the gene expression UMAP featureplots on our website. We believe it should be more convenient for researchers to access our data at https://tanglab.shinyapps.io/Mouse_Male_Germ_Cells/.

Minor:

Supp fig. 1b - some clusters distinct from the main trajectory only present in 1 replicate

Answer: Thank the reviewer for this question. We checked our data carefully and re-analyzed cells in C7 by re-clustering (Rebuttal Fig. R4a, already shown in our original manuscript). In addition, the developmental trajectory was reconstructed using Monocle (Rebuttal Fig. R4b).

Our results showed that germ cells in C7 were developed along the ordered developmental trajectory. And the cells from Repeat 1 exhibited similar molecular characteristics like other cells of the same cell cluster (Rebuttal Fig. R4c and 4d). Based on these results, the certain heterogeneity of C7 does not affect our conclusions and the subsequent analysis.

Fig. 2d appears before Fig. 2a-c

Answer: Thank the reviewer for this suggestion. We have revised it in the revised manuscript.

Page 7 line 28 and 30 - “IRF1” for protein

Answer: We thank the reviewer for this suggestion. We have modified all the names of mouse genes and mouse proteins in the revised manuscript according to the international standard.

The appearance of figures of supp fig. 4 in text is not in order

Answer: Thank the reviewer for this suggestion. We have revised this in the revised manuscript.

Fig 3e - “signature”

Answer: Thank the reviewer for this comment, we have revised this in the revised manuscript.

Page 10 line 19 - “HESI”

Answer: We thank for this suggestion. We have checked all the names of mouse genes and mouse proteins in our manuscript and revised them according to the international standard.

All proteins should be in capital letter for mouse

Answer: We thank for this suggestion. We have checked all the names of mouse genes and mouse proteins in our manuscript and revised them according to the international standard.

Page 11 line 20 - Supplementary Data 2

Answer: Thank the reviewer and we have revised it in the revised manuscript.

Fig. 5c appears after Fig. 5f

Answer: Thank the reviewer and we have revised it in the revised manuscript.

In Fig. 5e and Supplementary Fig. 6f, the developmental delay in Helq null PGCs during E15.5 and E18.5 is obvious. However, it seems that in E12.5 and E13.5, the WT PGCs showed a delay compared with Helq null PGCs.

Answer: We thank the reviewer for this comment. Our results demonstrated that *Helq* knockout resulted in the hyperactivation of p-ATR/p-CHK1 regulatory axis (Rebuttal Fig. R11), which might further cause the abnormal cell cycle switch of PGCs (Rebuttal Fig. R11). Therefore, the emergence of transitional PGCs at E12.5 when *Helq* was knocked out indicate that E12.5 *Helq*^{-/-} PGCs might aberrantly enter the cell cycle arrest state by the aberrant hyperactivation of p-ATR/p-CHK1 regulatory axis, which exhibited an advanced development on the developmental trajectory. Although E12.5 *Helq*^{-/-} PGCs was developmentally advanced than the WT control, the hyperactivation of p-ATR/p-CHK1 regulatory axis might result in the cell arrest for the prolonged cell cycle checkpoints at E15.5 as exemplified by the loss of post-arrest PGCs. Thus, we have re-described these results of this part in the revised manuscript.

Page 12 line 26 - "CellCycleSocring"

Answer: We thank and have revised it in the revised manuscript.

The order of Supplementary Fig. 7a and 7b is reversed

Answer: We thank and have revised it in the revised manuscript.

Fig. 7l should be Fig. 7k in text and figures

Answer: We thank and have revised it in the revised manuscript.

Reviewer #3 (Remarks to the Author):

In this study, Zhao et al. performed single-cell RNA-seq analysis of testicular cells at 28 time-points throughout mouse germline development. Their data provide an integrated resource for understanding mouse germ cell development. Their analyses validate previous studies, as well as uncover some new gene markers and signaling pathways potentially involving in mouse germline development. They claim to have identified a new transitional germ cell stage between the mitotically active PGC and arrested pro-spermatogonia (ProSPG) stages. They provide evidence that the Notch signaling pathway promotes the transition of mitotic PGCs to mitotically arrested ProSPG. They also provide evidence that HELQ, a DNA damage-associated cell cycle regulator, drives the transition of mitotic PGCs to mitotically arrested ProSPG. Finally, they combine their mouse scRNA-seq datasets with analogous previously published human datasets to identify conserved gene regulatory networks.

Overall, this study provides a good integrated resource for the field. However, as explained below, novelty appears to be weak. In addition, there are several other concerns that need to be addressed.

Answer: We thank the reviewer for the positive comments and his/her appreciation that our work provides a good integrated resource for the field.

Major concerns:

1) *While the authors examined some time points not previously examined by scRNA-seq, most of their scRNA-seq analyses are focused on known cell populations previously identified by published single cell RNA-seq analyses. While they claimed some “new” subsets, those are not convincing or were previously defined by others (see below).*

Answer: Thank the reviewer for this comment. First of all, considering the contribution of genetic background to the global gene expression variations¹⁶, and difficulties in integrating single-cell data across different platform¹⁷, it has profound significance to establish high-quality full-term transcriptome atlas of mouse male germ cell development at single-cell resolution. It is known that due to the "batch effect"¹⁸ which caused by different sequencing technologies, different library preparation protocols, and different data interpretation strategies,

it is quite difficult to integrate different single cell sequencing data, which is critical for understanding the cell-fate transition and determination of male germ cells. However, no tool so far can thoroughly overcome the insurmountable gap like “Integration analysis calculates differentially expressed genes with inaccuracy”¹⁹. This basic problem is discussed intensely by authoritative biologists and bioinformaticians (<https://github.com/satijalab/seurat/issues/>), but there is still no ideal solution. In this regard, the full single-cell transcriptome atlas of male germline cycle should be established in an unbiased way, allowing for an accurate overview of the cell-fate transition and determination underlying male germ cell development without significant batch effects.

Secondly, although precious studies illuminated specific stages of germ cell development¹⁻¹⁰, none of them monitored a full-term program of germ cell development in an independent study. To the best of our knowledge, our study is the first to resolve the whole developmental process of mouse male germ cells based on dense sampling time-points, covering essentially all stages throughout germ cell development. Compared with 10×Genomics method, our single-cell single-tube amplification library construction method provides much higher gene detection rate (about 9,413 genes in each cell), and higher sensitivity in capturing RNA molecules (reviewed recently by directly comparing seven methods for single-cell and/or single-nucleus profiling)¹¹, which can dig out more lowly-expressed functional genes such as transcription factors. It would be convenient to know the dynamic expression patterns of any candidate genes for functional study. In addition, our dataset would be a high-quality comprehensive reference for comparison to inform the accurate cell type-specific transcriptional changes when any gene of interest is manipulated in male mouse germline.

Lastly, with the unbiased single-cell transcriptome reference map for the full-term developmental process of male germ cells, we have systematically delineated the transcriptome reconfiguration from mitotic PGCs to post-arrest PGCs at single-cell resolution for the first time, wherein a transitional PGCs progression was uncovered to be crucial for this process. In addition, we have revealed the biological functions and mechanisms of Notch signaling pathway and HELQ in the cell-fate determination of mouse male gonadal germ cells in great

detail.

To summarize, our work offers novel insights into research of cell-fate transition and determination during male germ cell development.

2) The new identified “transitional PGC” subset is not convincing. Based on Fig. 3c, it's hard to believe this transitional PGC is a real cell subpopulation, as it does not form a clearly defined "cell cluster." Following this logic, it would appear that cells can be divided into as many as stages as the authors like. To make this claim, the authors need to provide more evidence that this is a new identified cell subpopulation, as well as provide some evidence for their functional roles. If not, related statements need to be corrected.

Answer: We appreciate for the comments of this reviewer. We are sorry for the misunderstanding due to the lack of essential descriptions and more details for cell cluster definition.

Firstly, for the comment “*it's hard to believe this transitional PGC is a real cell subpopulation, as it does not form a clearly defined "cell cluster."*”, we have added more experiments and new data in the revised manuscript to support the conclusion that the transitional PGCs is indeed an independent cell population. In detail:

(1) Automatically clustering results by the software showed that these cells (transitional PGCs) were first clustered as one part of the mitotic arrest PGCs based on dimensionality reduction of the global gene expression patterns (Fig 1c of the revised Manuscript); however, the proliferation status of the transitional PGCs was more similar to that of the mitotic PGCs rather than the mitotic arrest PGCs (Rebuttal Fig. R16a, see also Fig. 3a of the revised Manuscript). By using the gene set related to cell cycle, transitional PGCs can be clustered as an independent cell population (Rebuttal Fig. R16b, see also Supplementary Fig. 4a of the revised Manuscript). Further study found that these cells (transitional PGCs) shared the transcriptome signatures of both mitotic PGCs and mitotic arrest PGCs indeed (Rebuttal Fig. R16c, see also Supplementary Fig. 4f of the revised Manuscript); for instance, both of the

mitotic PGC's marker gene *Lefty2* and the mitotic arrest PGC's marker gene *Nanos2* were expressed in transitional PGCs. Based on these evidences, we defined these cells as transitional PGCs, which acted as the intermediate cell state between the mitotic PGCs and the mitotic arrest PGCs.

(2) On the other hand, to evaluate the heterogeneity of identified cell clusters, we performed ROGUE-guided analysis to calculate the purity score of cell types before and after re-clustering of the mitotic arrest PGCs as previously reported²⁰. Our results showed that the purity score of mitotic arrest PGCs was increased from 0.26 to 0.50 when re-clustering was performed (Rebuttal Fig. R16d, see also Supplementary Fig. 4b of the revised Manuscript), indicating that the transitional PGCs should be separated from the mitotic arrest PGCs for a better definition of these cell populations.

Secondly, the cellular characteristics of transitional PGCs were analyzed by comparing with the mitotic PGCs and mitotic arrest PGCs. We found that the transitional PGCs' signature genes were enriched in GO terms such as "Cell cycle regulation", "Cell cycle phase transition", and "APC/C-mediated degradation of cell cycle proteins" (Rebuttal Fig. R16e, see also Supplementary Fig. 4d of the revised Manuscript), suggesting that cell cycle might be involved in the regulation of PGC development at this stage. Among the genes associated with these terms, *Wee1* (encoding WEE1) was selected for further study. By co-staining of MKI67 (a marker for cell proliferation) and WEE1, we found that the developmental behavior of transitional PGCs (MKI67 and WEE1 double positive cells) could be traced, which were highly consistent with that of transitional PGCs as revealed by scRNA-seq data (Rebuttal Fig. R16f-h, see also Fig. 5e-j of the revised Manuscript). Based on these results, the developmental behavior of transitional PGCs could be traced by co-staining of MKI67 and WEE1.

Lastly, concerning on the function of transitional PGCs in PGC development, we traced the behavior of the transitional PGCs in *Helq*^{-/-} germ cells. Notably, we observed a significant increase of MKI67 and WEE1 double positive cells (transitional PGCs) at E14.5 and E15.5 in *Helq*^{-/-} group relative to the WT group (Fig. 16f-h, see also Fig. 5g and j, and Supplementary

Fig. 8e of the revised Manuscript). Based on these data, HELQ depletion could disrupt the mitotic to mitotic arrest transition of PGCs, wherein HELQ deficiency might hyperactivate the p-ATR/p-CHK1 regulatory axis, which then prolonged the cell cycle checkpoint and resulted in PGC arrest (Also see the answer to major question 10 raised by Reviewer #1).

In conclusion, based on all these data, we believe that transitional PGCs is a new cell population, which is critical for the cell-fate transition from mitotic to post-arrest PGCs.

Rebuttal Fig. R16. The identification and molecular characteristics of transitional PGCs.

a, (Top) Heatmap for expression levels of 97 cell cycle-specific genes (rows) at the four stages of mouse male PGC development (columns) ordered by the expression levels in each phase.

(Bottom) Heatmap of the top 20 genes of each stage with cells (columns) in the same order. The color key from blue to yellow indicates low to high expression levels. **b**, Heatmap showing hierarchical clustering result of mitotic arrest PGCs identified from Fig. 1c using cell cycle-related markers. **c**, Violin plots showing the relative expression levels ($\log(\text{TPM}/10+1)$) of representative genes related to pluripotency and cell cycle arrest. **d**, Purity evaluation of corresponding cell types before and after re-clustering using ROGUE tool. **e**, The enriched GO terms and representative signature genes of transitional PGCs compared with the mitotic PGCs and mitotic arrest PGCs. **f**, Immunofluorescence of WEE1 co-stained with MKI67 and BLIMP1 (GFP) in the WT- and *Helq*^{-/-} mouse male gonads at E14.5. Scale bar, 10 μm . **g**, Immunofluorescence of WEE1 co-stained with MKI67 and BLIMP1 (GFP) in the WT- and *Helq*^{-/-} mouse male gonads at E15.5. Scale bar, 10 μm . **h**, Relative proportions of mitotic- (MKI67⁺ WEE1⁻), transitional- (MKI67⁺ WEE1⁺), and mitotic arrest PGCs (MKI67⁻ WEE1⁻) in the WT- versus *Helq*^{-/-} mouse male gonads at E14.5. Mean \pm SEM, n = 4 per group, ns, not significant, * $P < 0.05$, unpaired two-tailed t test.

3) *The novelty of the identification of the role of Notch in fetal germ cells is limited. First, Notch signaling has previously been reported to be active in developing mouse testes, both in germ cells and Sertoli cells (PMID: 18801836). In addition, roles of the Notch pathway in PGCs have previously been reported in chicken, C. elegans, Xenopus, sea urchin, and Drosophila (PMID: 29951200, PMID: 32008902, PMID: 20151992, PMID: 23533178, PMID: 20660750). None of these studies are cited in this study. Second, finding expression of NOTCH signaling components in a cell does not demonstrate that the NOTCH pathway is activate in this cell (PMID: 23907117). Indeed, the detection of Notch1 and Hes1 expression in mouse PGCs here does not mean the Notch pathway is activated in PGCs. Third, the in vivo effects observed by the authors in response to the Notch inhibitor, DAPT, do not necessarily reflect a defect in Notch signaling in germ cells. Indeed, it was previously shown that activating the Notch pathway in Sertoli cells influences fetal germ cell quiescence (PMID: 23907117, PMID: 23391689). In addition, several studies have shown that the Notch pathway is active exclusively in Sertoli cells in the perinatal testis, and that activating Notch is essential for fetal germ cell quiescence (PMID: 23907117, PMID: 23391689).*

Answer: Thank the reviewer for this comment. We are sorry for our unintentional non-citing of parts of those references as listed by the reviewer. And we have cited them in the appropriate positions in the revised manuscript. Although these previous studies have reported the roles of Notch signaling pathway to be associated with germ cell development, especially in non-mammalian animals, our study here provided some new insights about the function of Notch signaling pathway in mammalian male germ cell development.

Firstly, Notch signaling pathway has been shown to be activated in developing mouse testis previously; however, to the best of our knowledge, there is not enough evidence to prove that the components or canonical targets of Notch signaling pathway were expressed in mouse male gonadal germ cells, and about whether the interruption of Notch signaling pathway was associated with germ cell development abnormality. Our immunostaining results of NOTCH1 and HES1 in the original manuscript and further immunostaining results of NOTCH2, NOTCH3, and NOTCH4 provided direct evidence that NOTCH1/3-HES1 axis was indeed activated in mouse male prenatal germ cells, while NOTCH2/4 might not involve in the transition from mitotic to mitotic arrest PGCs at E13.5-15.5 (Rebuttal Fig. R17a-d, see also Supplementary Fig. 6a-d of the revised Manuscript).

Secondly, to further confirm if Notch signaling pathway was activated in germ cells, another Notch signaling pathway inhibitor-LY411575 (a commercial chemical γ -Secretase inhibitor, N-[(1S)-2-[[[(7S)-6,7-dihydro-5-methyl-6-oxo-5H-dibenz[b,d]azepin-7-yl]amino]-1-methyl-2-oxoethyl]-3,5-difluoro- α S-hydroxy-benzeneacetamide, Selleck, S2714) was used to block Notch signaling pathway and we evaluated its function accordingly (Rebuttal Fig. R17g-l, see also Supplementary Fig. 6g-l of the revised Manuscript). Our results showed that the treatment by LY411575 could also result in a significant decrease of NOTCH1 and HES1 in germ cells, wherein the number of MKI67⁺ PGCs was significantly increased compared with the mock controls (Rebuttal Fig. R17i, l, see also Supplementary Fig. 6i, l of the revised Manuscript). Short-term treatment of the *in vitro* cultured mouse male PGCs by DAPT or LY411575 could also lead to the decreased expression of *Hes1*, demonstrating that PGCs could respond to the blocking of Notch signaling pathway properly (Rebuttal Fig. R17e, f, see also Supplementary Fig. 6e, f of

the revised Manuscript). The results demonstrated that Notch signaling pathway was indeed activated in mouse male gonadal germ cells, and played critical roles for the cell-fate transition of PGCs.

Lastly, we admitted that Notch signaling pathway in somatic cells exerted certain regulatory roles in germ cell development. We have checked these two references^{21,22} mentioned by Reviewer #3. These studies using gain-of-function strategy suggest that proper regulation of Notch signaling pathway in Sertoli cells is required for the maintenance of gonocytes (prospermatogonia) in an undifferentiated state during fetal development. Consistent with this finding, we also observed the gradual decrease of the protein expression of NOTCH1 and HES1 in gonadal somatic cells (Fig. 4a and 4b in the original manuscript, see also Fig. 4a, b of the revised Manuscript), suggesting that Notch signaling pathway should be downregulated in Sertoli cells for the maintenance of germ cells in the quiescent state. However, based on the acquired data in the present study, we believed that the intrinsic Notch signaling pathway in germ cells might also be important for its own development. Thus, the previous studies and our present study illustrated the roles of Notch signaling pathway in male gonad development by acting on the somatic cells and germ cells, respectively.

To summarize, our work uncovered the activation of Notch signaling pathway in mouse male PGCs and verified the role of Notch signaling pathway in regulating the transition from PGCs to prospermatogonia.

Rebuttal Fig. R17. Detection of Notch signaling pathway components and blocking of Notch signaling pathway *in vitro* or *in vivo*. a, Immunofluorescence of NOTCH2 co-stained

with DDX4 in E13.5 and E15.5 mouse male gonads. Scale bar, 10 μm . **b**, Immunofluorescence of NOTCH4 co-stained with DDX4 in E13.5 and E15.5 mouse male gonads. Scale bar, 10 μm . **c**, Immunofluorescence of NOTCH3 co-stained with DDX4 in E12.5-E15.5 mouse male gonads. Scale bar, 10 μm . **d**, Proportion of NOTCH3^{Negative} and NOTCH3^{Positive} subtypes in DDX4⁺ cells in E13.5 and E15.5 mouse male gonads. **e**, Schematic showing the workflow of treatment by γ -Secretase inhibitor–DAPT or LY411575 for Notch signaling pathway blocking *in vitro*. The BVSC⁺ cells (PGCs) were cultured in a well of 5,000 cells in a low-cell-binding U-bottom 96-well plate in PGC medium (see Method in the revised Manuscript). **f**, Q-PCR analysis of the expression of PGCs marker genes (*Ddx4*, *Prdm1*, *Dppa3*) and Notch signaling pathway target gene *Hes1* in *in vitro* cultured PGCs after 24 h treated by DAPT or LY411575. To verify the germ cell identity, the sorted PGCs and somatic cells without treatment were used as the controls. Relative expression levels are shown with normalization to *Gapdh*. Error bars indicate mean \pm SEM from at least two independent biological replicates. ** $P < 0.01$, unpaired two-tailed t test. **g**, Immunofluorescence of NOTCH1 co-stained with DDX4 in LY411575-treatment and control mouse male gonads at E15.5. Scale bar, 10 μm . **h**, Immunofluorescence of HES1 co-stained with DDX4 in LY411575-treatment and control mouse male gonads at E15.5. Scale bar, 10 μm . **i**, Immunofluorescence of MKI67 co-stained with DDX4 in LY411575-treatment and control mouse male gonads at E15.5. Solid yellow arrowheads indicate MKI67^{Positive} subtypes in DDX4⁺ cells. Scale bar, 10 μm . **j**, The quantification of relative fluorescence intensity of NOTCH1 in LY411575-treatment and control mouse male gonads at E15.5. Mean \pm SEM, $n = 3$ per group, **** $P < 0.0001$, unpaired two-tailed t test. **k**, The quantification of relative fluorescence intensity of HES1 in LY411575-treatment and control mouse male gonads at E15.5. Mean \pm SEM, $n = 3$ per group, **** $P < 0.0001$, unpaired two-tailed t test. **l**, Proportion of MKI67⁺ cells in DDX4⁺ cells in LY411575-treatment and control mouse male gonads at E15.5. Mean \pm SEM, $n = 3$ per group, ** $P < 0.01$, unpaired two-tailed t test.

4) *The study of the role of Helq in fetal germ cells is a novel work in this study, and thus this reviewer encourages the authors to study this further. Currently, the authors only show the cellular changes after Helq ko. While the authors tested potential mechanisms (as shown in*

Fig. 5l), these are mainly following previous publications (PMID: 24005329 and PMID: 24005041). Some further work can be done. For example, the authors could go deeper into how *Helq* influences fetal germ cell development. Currently, the authors stated that *Helq* depletion causes delayed development of fetal PGCs. However, based on the cell composition (Fig. 5f), another possibility is that *Helq* depletion impacts PGC growth or maintenance. This is supported by the fact that there are less mitotic PGCs at E13.5 and E14.5 in *Helq* ko mice compared to WT mice. In addition, their finding that apoptosis is upregulated in *Helq*^{-/-} PGCs also supports this hypothesis. The authors could go more deeply into other phenotypic defects. In addition, the authors could identify candidate HELQ-regulated genes from their scRNA-seq data and test their roles.

Answer: We thank the reviewer for these comments and suggestions. Accordingly, we have performed more experiments to investigate the potential mechanisms of HELQ in the regulation of germ cell development. We found that there were several interesting phenotypes and mechanisms about HELQ ablation in germ cells by combined analysis of the original data and the newly obtained data. The conclusions were listed below:

(1) Compared to the WT control, the higher ratio of MKI67-positive germ cells but lower ratio of total number of germ cells was detected in seminiferous cords of *Helq*^{-/-} mice (Rebuttal Fig. R18a, b, see also Fig. 5a and Supplementary Fig. 7g of the revised Manuscript). This finding suggests that the *Helq*^{-/-} germ cells do not proliferate faster; instead, they might be in the state of cell cycle arrest and cannot properly exit to G0 phase for cell quiescence (data from the original manuscript).

(2) Our immunostaining results showed that there was a remarkable increase of p-CHK1 positive cells in the *Helq*^{-/-} group compared with the WT control (Rebuttal Fig. R18d, h and i, see also Fig. 5f, i and Supplementary Fig. 8d of the revised Manuscript). p-CHK1 acts as the downstream effector of p-ATR in cell cycle checkpoint^{23,24}. Thus, this finding was consistent with our original data that ATR was hyperactivated in *Helq*^{-/-} germ cells (Rebuttal Fig. R18c and g, see also Fig. 5e and h of the revised Manuscript). These results suggest that *Helq* knockout might enforce the S/G2 checkpoint and result in the cell cycle arrest in S/G2 phase.

(3) By the co-staining of MKI67 and WEE1, the developmental behavior of transitional PGCs can be traced, which was critical for cell-fate transition of PGCs. Accordingly, we checked the behavior of transitional PGCs by co-staining of MKI67 and WEE1 (Rebuttal Fig. R18e, f and j, see also Fig. 6g, j and Supplementary Fig. 8e of the revised Manuscript). Our results showed that there was a significant increase of transitional PGCs (MKI67 and WEE1 double positive cells) at E14.5 (E14.5 is the time when transitional PGCs accounted for the largest proportion in the WT control) when *Helq* was knocked out (Rebuttal Fig. R18e, j, see also Fig. 6g, j of the revised Manuscript). Moreover, we also observed a significant increase of transitional PGCs in the *Helq*^{-/-} group at E15.5 (Rebuttal Fig. R18f, see also Supplementary Fig. 8e of the revised Manuscript), which was rare in the WT control. Based on these data, HELQ depletion could disrupt the mitotic to mitotic arrest transition of PGCs, wherein HELQ deficiency might hyperactivate the p-ATR/p-CHK1 regulatory axis, which then prolonged the cell cycle checkpoint and resulted in PGC arrest.

(4) Concerning on the novel role of HELQ in the regulation of germ cell development, we also tried to dissect certain interesting downstream target. In the DEG list when *Helq* was knocked out, we found *Helq*^{-/-} germ cells highly expressed *Cdh1* (encoding E-cadherin) (Rebuttal Fig. R10b, see also Supplementary Fig. 9c), which should be gradually downregulated in the WT control (involved in the module of “Loss” in Fig. 3e and Supplementary Data 2 of the revised Manuscript; Rebuttal Fig. R10c, see also Supplementary Fig. 9d of the revised Manuscript), and this downregulation trend was also verified by immunostaining of E-cadherin (Rebuttal Fig. R10d, see also Supplementary Fig. 9e of the revised Manuscript). Immunostaining results further validated the higher expression of E-cadherin protein in *Helq*^{-/-} germ cells compared with the WT control (Rebuttal Fig. R10e, see also Fig. 5l of the revised Manuscript). Moreover, GO analysis showed that *Cdh1* and other abnormally expressed genes were enriched in the processes such as “Apoptotic cleavage of cellular proteins”, “Apoptosis”, “Tight junction assembly”, “Hippo signaling pathway”, “Pathways in cancer”, and “Negative regulation of cell differentiation” (Supplementary Data 3 of the revised Manuscript), which were highly related to the phenotypes of *Helq*^{-/-} germ cells

as we observed. Consistent with previous studies, the expression of E-cadherin (encoded by *Cdh1*) in male germ cells should be downregulated once sexual differentiation completed, which was critical for the differentiation of prenatal male germ cells^{12,13}. Based on these data, it is most likely that HELQ also functions in the regulation of PGC differentiation. (Also see the answer to major question 9 raised by Reviewer #1).

(5) In the original manuscript, immunostaining of cleaved PARP1 (cPARP1) demonstrated that apoptosis was upregulated in *Helq*^{-/-} PGCs relative to the WT controls (Fig. 5b and Supplementary Fig. 7e of the revised Manuscript). This result suggests that *Helq*^{-/-} PGCs could not be properly maintained and certain null cells might be eliminated through apoptosis. Consistently, the p-ATR/p-CHK1 regulatory axis was abnormally activated in *Helq*^{-/-} PGCs, which might act as the upstream DNA cell cycle checkpoints for the induction of apoptosis.

In summary, based on the accumulated data of our study, HELQ plays an important role in male PGC development at least via two pathways: (1) the first one might be through the p-ATR/p-CHK1 regulatory axis, which might be necessary for cell cycle switch of PGCs as exemplified by abnormal behavior of the newly identified transitional PGCs when *Helq* was ablated; (2) the second one might be through other potential downstream targets like E-cadherin (encoded by *Cdh1*) to regulate further differentiation of PGCs. More studies are needed to elucidate other mechanisms of HELQ in regulating male germ cell development.

Rebuttal Fig. R18. Proper mitotic to mitotic arrest transition is crucial for the prenatal development of male germ cells. a, Numbers of DDX4⁺ cells from E11.5 to E16.5, and E18.5

WT versus *Helq*^{-/-} mouse male gonads. Mean ± SEM, n = 4 per time-point, ns, not significant, *** $P < 0.001$, unpaired two-tailed t test. **b**, Proportions of MKI67⁺ cells co-staining with DDX4⁺ from E11.5 to E16.5, in the WT and *Helq*^{-/-} mouse male gonads. Mean ± SEM, n = 4 per time-point, ns, not significant, ** $P < 0.01$, **** $P < 0.0001$, unpaired two-tailed t test. **c**, Immunofluorescence of p-ATR co-stained with DDX4 in the WT- and *Helq*^{-/-} mouse male gonads at E14.5. Scale bar, 10 μm. **d**, Immunofluorescence of p-CHK1 co-stained with MKI67 and BLIMP1 (GFP) in the WT- and *Helq*^{-/-} mouse male gonads at E14.5 and E15.5. Scale bar, 10 μm. **e**, Immunofluorescence of WEE1 co-stained with MKI67 and BLIMP1 (GFP) in the WT- and *Helq*^{-/-} mouse male gonads at E14.5. Scale bar, 10 μm. **f**, Immunofluorescence of WEE1 co-stained with MKI67 and BLIMP1 (GFP) in the WT- and *Helq*^{-/-} mouse male gonads at E15.5. Scale bar, 10 μm. **g**, The quantification of relative fluorescence intensity of p-ATR in the WT- and *Helq*^{-/-} mouse male germ cells at E14.5. Mean ± SEM, n = 4 per group, **** $P < 0.0001$, unpaired two-tailed t test. **h**, Relative proportions of p-CHK1⁺ germ cells in the WT- and *Helq*^{-/-} mouse male germ cells at E14.5. Mean ± SEM, n = 4 per group, ** $P < 0.01$, unpaired two-tailed t test. **i**, Relative proportions of p-CHK1⁺ germ cells in the WT- and *Helq*^{-/-} mouse male germ cells at E15.5. Mean ± SEM, n = 4 per group, * $P < 0.05$, unpaired two-tailed t test. **j**, Relative proportions of mitotic- (MKI67⁺ WEE1⁻), transitional- (MKI67⁺ WEE1⁺), and mitotic arrest PGCs (MKI67⁻ WEE1⁻) in the WT- versus *Helq*^{-/-} mouse male gonads at E14.5. Mean ± SEM, n = 4 per group, ns, not significant, * $P < 0.05$, unpaired two-tailed t test. **k**, Illustration of the mouse male germ cells development in HELQ defective condition.

5) *The scRNA-seq analysis of Helq^{-/-} germ cells has only one biological replicate; at least one more replicate should be provided for rigor. This is important, as the authors find opposite effects at two stages of development (Figs. 5e and S6f), which could be due to sampling bias. Even if their finding is correct, their interpretation that inhibited development at one stage causes an accumulation of germ cells at later time point, is not necessarily correct. Another possibility is that loss of Helq inhibits germ cell maturation at one stage and promotes germ cell events later. These two possibilities should be distinguished or at least acknowledged.*

Answer: Thanks the reviewer for this question. We are sorry that the original manuscript might

somehow lead to the misunderstanding and confusion due to the insufficient description of our data. Here, we explained this question based on several lines of evidence.

Firstly, two independent biological replicates were actually performed for *Helq*^{-/-} group at E13.5, E14.5, E15.5, and E18.5 (after quality control, there were 227 and 283 *Helq*^{-/-} germ cells at E13.5-15.5 in R1 and R2, versus 279 and 278 in the WT control, these data were shown in our original Supplementary Data 1 and Supplementary Data 3, and Supplementary Data 1 and Supplementary Data 3 of the revised Manuscript). Thus, the developmental defects of *Helq*^{-/-} germ cells were caused by its loss-of-function.

Secondly, to confirm the finding detected by scRNA-seq data, immunostaining assay was also performed. Our results showed that the *Helq*^{-/-} germ cells exhibited a clear developmental arrest and abnormal cell-fate transitions. (1) Compared to the WT control, the higher ratio of MKI67-positive germ cells but lower ratio of total number of germ cells were observed in the seminiferous cords of *Helq*^{-/-} mice. This finding suggests that the *Helq*^{-/-} germ cells do not proliferate faster; instead, they might be in the state of cell cycle arrest and cannot properly exit to G0 phase for cell quiescence. More importantly, by the co-staining of MKI67 and WEE1, the developmental behavior of the transitional PGCs can be traced, which was critical for cell-fate transition of PGCs. We found that *Helq*^{-/-} germ cells exhibited a clear developmental arrest phenotype; especially, an increased number of transitional PGCs were observed when *Helq* was knocked out (Rebuttal Fig. R18e, f and j, see also Fig. 6g, j and Supplementary Fig. 8e of the revised Manuscript; we counted at least 25 sections each, totally more than 3,000 cells). Furthermore, consistent with of the cell cycle analysis results in mESCs and germ cells, our immunostaining results showed that hyperactivation of the cell cycle checkpoints as exemplified by the activity of both p-ATR and p-CHK1. These results suggest that *Helq* knockout might enforce the S/G2 checkpoint and result in cell cycle arrest in S phase, thereby impeding the proper exit to G0 phase for PGC quiescence.

Lastly, in terms of the seemed opposite effects of HELQ in different stage of development, our data demonstrated that *Helq* knockout resulted in the hyperactivation of p-ATR/p-CHK1

regulatory axis, which might further cause the abnormal cell cycle switch of PGCs as exemplified by the increase of transitional PGCs. According to these results, it is possible that *Helq*^{-/-} early-stage PGCs (E12.5) might be aberrantly enter the cell cycle arrest state, which was also regulated by the aberrant hyperactivation of p-ATR/p-CHEK1 regulatory axis. In spite of this possibility, we admitted that HELQ might exert different function during the process of PGC development. And we have acknowledged the possibility in the Discussion part of the revised manuscript as suggested by the reviewer. In fact, HELQ might have different downstream effectors; for instance, the expression of *Cdh1* (encoding E-cadherin) was gradually decreased from E13.5 to E15.5, and genetic ablation of *Helq* resulted in the increased expression of E-cadherin protein. Given the fact that *Helq* encodes a DNA helicase, it was most likely that some genes were directly regulated by HELQ, while some genes were indirectly regulated by HELQ as listed in Supplementary Fig. 9c and Supplementary Data 3 of the revised Manuscript.

6) *More data analyses should be done to identify conserved gene regulatory networks between humans and mice. The authors integrated their own previously published datasets with that of Wang et al., but there are several resources in the field, e.g., PMID: 30726734, PMID: 31928944, and PMID: 33453151. While these datasets are from different platforms, the authors should at least compare their enriched genes/signaling pathways to validate their own findings.*

Answer: Thank the reviewer for this suggestion. Accordingly, we have integrated our dataset in this study with the human scRNA-seq datasets from different platforms, spanning infancy to adulthood²⁵ (Rebuttal Fig. R19a, b). Clustering analysis showed that human undiff.ed SPG-1 and undiff.ed SPG-2 were well-merged with mouse Q-ProSPG and T-ProSPG, respectively (Rebuttal Fig. R19a-c), suggesting that they represent a more ‘naïve’ state; while human undiff.ed SPG-3 were merged with mouse undiff.ed SPG. And human and mouse diff.ing SPG and diff.ed SPG could be also well-merged, respectively. Further analysis showed that the expression patterns of the highly conserved stage-by-stage matched regulators identified in our study were highly conserved between mouse and human (Rebuttal Fig. R19d). Overall, these analyses emphasized its potential conserved regulatory roles in mouse and human germ cell development.

Rebuttal Fig. R19. Interspecies comparison of male germ cell development between human and mouse from this study and a previous study²⁵ (PMID: 31928944). **a**, PCA plots showing integration results of germ cells from this study and Guo *et al.*²⁵. Cells are colored based on cell types. **b**, Cells are colored based on time-points. **c**, PCA plots of expression levels of *Zbtb16*, *Gfra1*, *Kit*, and *Stra8*. The color key from gray to blue indicates low to high expression levels. **d**, Heatmap showing expression levels of the highly conserved stage-by-stage matched regulators scaled among mouse ProSPG and SPG from this study (left), human SPG from Wang *et al.*³ (middle), and human SPG from Guo *et al.*²⁵ (right), respectively.

7) Fig. 5c. Is the % at E13.5 and E14.5 really significantly changed? With only 4 samples, it is

hard to believe that there is a significant difference. Please also check the statistical test in Fig. 5h.

Answer: Thank the reviewer for this comment. We have carefully checked our raw data. Firstly, for the statistics of the percent of MKI67⁺ cells in DDX4⁺ fraction, we have counted 4 independent samples from different embryos, and there were significant differences between the WT and *Helq*^{-/-} during E13.5 and E16.5 (Rebuttal Fig. R20). Secondly, we have checked the original data and the proportion of S population was 47.84% as shown in the text. And we have modified the number in the Supplementary Fig. 8c of the revised Manuscript. We make sure that the proportion of G1 and G2/M were correct. Our raw data of G1, S, and G2/M phase in the WT and *Helq*^{-/-} mESCs (powered by GraphPad Prism) were shown in Rebuttal Fig. R13. Indeed, the difference was significant between the WT and *Helq*^{-/-} group.

Rebuttal Fig. R20. Raw data for the immunofluorescence of MKI67⁺ cells in WT- and *Helq*^{-/-} DDX4⁺ cells. a, Quantitative data of MKI67⁺ cells co-staining with DDX4⁺ from E13.5

to E16.5, in the WT and *Helq*^{-/-} mouse male gonads. **b**, Proportions of MKI67⁺ cells co-staining with DDX4⁺ from E13.5 to E16.5, in the WT and *Helq*^{-/-} mouse male gonads. Mean ± SEM, n = 4 per time-point, ** *P* < 0.01, **** *P* < 0.0001, unpaired two-tailed t test. **c**, Statistical data of cell cycle analysis, * *P* < 0.05, ** *P* < 0.01, unpaired two-tailed t test.

Minor concerns:

8) *The authors often incorrectly refer to ProSPG as PGCs. PGCs convert into M ProSPG at ~E11.5-E13.5 (PMID: 23843236). Thus, what they call “mitotic PGCs” during this period should contain both PGCs and M-ProSPG. What they refer to as “Arrest PGCs” are typically called T1-ProSPG (PMID: 23843236). The authors also refer to other ProSPG stages that they call “Q- and T-ProSPG,” without defining them. If they want to use this system, they should not only define them, but refer to a publication where this system is used. It is suggested to use the M-, T1-, and T2-ProSPG nomenclature instead. A final note: further confusion is rendered by the fact that the authors refer to gonocytes in the Introduction and never explain that gonocytes are the same as ProSPG.*

Answer: We are sorry for not explaining our naming system in detail, which caused some misunderstandings for the readers.

(1) First of all, according to a classical naming system²⁶, germ cells before colonization in gonads are all called PGCs. After colonization, male germ cells are called ProSPG (prospermatogonia), but different stages of ProSPG have its own name. In detail, germ cells have just colonized in gonads, which are still rapidly multiplying, they are called M-ProSPG, but when male germ cells enter mitotic arrest, they are called T1-ProSPG (transitional 1 ProSPG). After birth, the male germ cells resume proliferation, at this time they are called T2-ProSPG, which finally develop into spermatogonia.

(2) In another typical naming system²⁷, the germ cells from the specification PGCs to the proliferating germ cell after colonization in gonads (~E13.5) are collectively referred to as PGCs, and the germ cells thereafter until the spermatogonia after birth are collectively referred to as gonocytes (prospermatogonia).

(3) Given the fact there are many naming systems in this field, which always make it not easy to accurately understand the meaning of each word. Therefore, in previous studies of our group about human germ cell development^{28,29}, human male germ cells were classified into migrating FGCs (fetal germ cells, 4W, before colonization in gonad), mitotic FGCs (~4W-35W, after colonization in gonad), and mitotic arrest FGCs (9W-25W, after colonization in gonad). Thus, we follow the similar principle in the present study.

(4) In this study, we focused on the transition of germ cells after colonization in gonads, especially the transition of germ cells from the mitotic to mitotic arrest stage. In order to ensure the continuity for understating the story, we divided the PGCs before colonization in gonads into specification PGCs and migrating PGCs; male germ cells upon the entrance into the gonads, which still multiplied, were defined as mitotic PGCs; germ cells that exited from cell cycle after colonization were divided into mitotic arrest PGCs, post-arrest PGCs, Q-ProSPG, and T-ProSPG. Furthermore, during the transition from mitotic to mitotic arrest PGCs, we discovered a new subpopulation- the transitional PGCs.

Based on the classical nomenclature systems and the scRNA-seq data as well as experimental verification data in this study, we named the germ cells of mice in more detail to ensure the completeness and continuity of this study, which was mainly consistent with previous studies of our group^{28,29}. At the same time, to make it easier for reviewers and readers to understand the development sequence of germ cells, we have compared our nomenclature with the classic nomenclatures aforementioned (Rebuttal Fig. R21).

Mouse					Human						
Source	PMID: 23843236	PMID: 19306346		This paper		Source	PMID: 26046443				
E6.5	Primordial germ cells (PGCs)	Pre-migratory PGCs		Specification PGCs		~2-3W	Specification PGCs				
E7.5				Migrating PGCs			~4W	Migating PGCs/FGCs	Mitotic PGCs/FGCs		
E8.5											
E9.5											
E10.5				M-prospermatogonia (M-ProSPG)	Post-migratory PGCs	Mitotic PGCs	Transitional PGCs	Mitotic arrest PGCs	~9W	Mitotic arrest PGCs/FGCs	
E11.5											
E12.5	T1-ProSPG	Mitotic gonocytes		Post-arrest PGCs		~25W					
E13.5				Quiescent gonocytes					Quiescent ProSPG (Q-ProSPG)	Intermediate ProSPG (I-ProSPG)	Transitional ProSPG (T-ProSPG)
E14.5											
E15.5				Mitotic gonocytes	Migratory gonocytes				Before birth	?	
E16.5											
E17.5				T2-ProSPG					After birth	PreSPG (PMID: 30726734)	
PND0											
PND1											
PND2											
PND3											
PND4											
PND5											

Rebuttal Fig. R21. Schematic representation of the time-lines of germ cell development in mouse and human, modified from available literatures. (PMID: 23843236 and PMID: 19306346.) M, mitotic; T1, transitional 1; T2, transitional 2.

9) Pg 9, line 11. This “post-arrest PGC” has been identified by Law *et al.* 2019. The authors should perform a bioinformatic comparison to test their similarities.

Answer: We thank for this suggestion. Accordingly, we have downloaded the data of Law *et al.*³⁰ and performed graph-based clustering to subdivide the E16.5 population. Consistent with this study³⁰, three major cell clusters were found (Rebuttal Fig. R22a and b). Then, we extracted the mitotic arrest PGCs and post-arrest PGCs from our data to compare their similarities. According to the expression of selective marker genes, E16.5 C1 (Law *et al.*³⁰) were more like the mitotic arrest PGCs in this study, which highly expressed the germline pluripotency and stem cell markers such as *Nanog*, *Id4*, *Etv5*, and *Ret* (Rebuttal Fig. R22c and d). In contrast, the differentiation-associated markers such as *Dnmt3a*, *Dnmt3b*, *Sohlh1*, and *Sox3* were highly expressed in post-arrest PGCs of this study as well as E16.5 C2 and C3 (especially C3) (Rebuttal Fig. R22c and d). In conclusion, C2 and especially C3 of E16.5 in the previous study were more similar to the post-arrest PGCs of our study.

Rebuttal Fig. R22. Comparison of the similarities of post-arrest PGCs from this study and a previous study (PMID: 31243281). **a**, Dimensionality reduction and clustering of E16.5 cells from Law *et al.* (PMID: 31243281), shown in PCA plot. **b**, PCA plots of E16.5 cells from Law *et al.* (PMID: 31243281) are colored by expression levels of *Dppa3*, *Sox2*, *Nanog* and *Pou5f1*. The color key from gray to blue indicates low to high expression levels. **c**, Heatmap showing expression levels of germline pluripotency markers, stem cell markers, and differentiation-associated markers scaled among the E16.5 cells from Law *et al.* (PMID: 31243281), and male gonadal germ cells from this study, respectively.

10) *Potential doublets contamination?* Pg 7, line 20. *It is surprising that the authors found that round spermatids (RS) express pluripotency and SPG markers, given that RS are post-meiotic and certainly not pluripotent. One explanation is this is an artifact, and that these cells are*

actually doublets. The authors should investigate this possibility.

Answer: Thank the reviewer for this comment. Firstly, compared with other commercial methods, our STRT-seq method was established based on the strategy of single-cell single-tube amplification library construction, which essentially had no doublets and no cross-contaminations. A schematic diagram was made to show the real workflow of single cell collection for subsequent single-cell single-tube amplification library construction (Rebuttal Fig. R23a). Next, we have verified the expression of KLF4, one of the well-known pluripotency markers by immunostaining in round spermatids (Rebuttal Fig. R23b). In addition, we have detected *Gfra1* expression in round spermatids (Rebuttal Fig. R23c), consistent with our scRNA-seq data.

Rebuttal Fig. R23. The workflow of single cell collection and expression confirmation of several marker genes by immunostaining or q-PCR. a, Schematic diagram of workflow of single cell collection. **b,** Immunofluorescence of KLF4 co-stained with PLZF (ZBTB16) and PNA in adult mouse testis. Yellow and white arrowheads indicate SPG and RS, respectively. Negative (no primary antibody) control of immunofluorescence are also shown. Scale bar, 10

µm. **c**, Expression analysis of SPG-, SPC-, and RS marker genes in germ cells derived from adult testis by q-PCR. Relative expression levels are shown with normalization to *Gapdh*. Error bars indicate mean ± SEM from at least two independent biological replicates. The purity of FACS-sorted SPG, SPC, and RS was verified by expression of known germ cell markers.

11) *The authors refer to “transcriptional” regulation throughout the manuscript when they presumably instead mean “transcriptome.” Changes in mRNA levels as detected by RNAseq or scRNA-seq could be due to either alterations or transcription rate or RNA half-life, not necessarily the former. Thus, the authors are incorrect when they state that changes detected by these methods are transcriptional in the Abstract (“transcriptional reconfiguration” and “transcriptional reprogramming” should be revised), on pgs 5, top of pg 7, etc. Search and replace should be conducted throughout the entire MS.*

Answer: We thank for this suggestion. We have checked and revised it in the revised manuscript.

12) *Protein acronyms should be all caps and not italicized; e.g., HELQ, not Helq.*

Answer: We thank for this suggestion. We have checked all the names of mouse gene and mouse protein in our manuscript and revised them according to the international standard.

13) *Pg 29, line 21. The ref 80 has no mSSC culture method.*

Answer: We are sorry for this mistake. We have checked and cited the correct references^{31,32} (PMID: 12700182 and PMID: 21915114) in the revised manuscript.

References

- 1 Ernst, C., Eling, N., Martinez-Jimenez, C. P., Marioni, J. C. & Odom, D. T. Staged developmental mapping and X chromosome transcriptional dynamics during mouse spermatogenesis. *Nat Commun* **10**, 1251 (2019).
- 2 Chen, Y. *et al.* Single-cell RNA-seq uncovers dynamic processes and critical regulators in mouse spermatogenesis. *Cell Res* **28**, 879-896 (2018).
- 3 Wang, M. *et al.* Single-Cell RNA Sequencing Analysis Reveals Sequential Cell Fate Transition during Human Spermatogenesis. *Cell Stem Cell* **23**, 599-614 e594 (2018).
- 4 Lau, X., Munusamy, P., Ng, M. J. & Sangrithi, M. Single-Cell RNA Sequencing of the Cynomolgus Macaque Testis Reveals Conserved Transcriptional Profiles during Mammalian Spermatogenesis. *Dev Cell* **54**, 548-566 e547 (2020).
- 5 Shami, A. N. *et al.* Single-Cell RNA Sequencing of Human, Macaque, and Mouse Testes Uncovers Conserved and Divergent Features of Mammalian Spermatogenesis. *Dev Cell* (2020).
- 6 Green, C. D. *et al.* A Comprehensive Roadmap of Murine Spermatogenesis Defined by Single-Cell RNA-Seq. *Dev Cell* **46**, 651-667 e610 (2018).
- 7 Lukassen, S., Bosch, E., Ekici, A. B. & Winterpacht, A. Characterization of germ cell differentiation in the male mouse through single-cell RNA sequencing. *Sci Rep* **8**, 6521 (2018).
- 8 Guo, J. *et al.* Chromatin and Single-Cell RNA-Seq Profiling Reveal Dynamic Signaling and Metabolic Transitions during Human Spermatogonial Stem Cell Development. *Cell Stem Cell* **21**, 533-546 e536 (2017).
- 9 Tan, K., Song, H. W. & Wilkinson, M. F. Single-cell RNAseq analysis of testicular germ and somatic cell development during the perinatal period. *Development* **147** (2020).
- 10 Hermann, B. P. *et al.* The Mammalian Spermatogenesis Single-Cell Transcriptome, from Spermatogonial Stem Cells to Spermatids. *Cell Rep* **25**, 1650-1667 e1658 (2018).
- 11 Ding, J. *et al.* Systematic comparison of single-cell and single-nucleus RNA-sequencing methods. *Nat Biotechnol* **38**, 737-746 (2020).
- 12 Krentz, A. D. *et al.* The DM domain protein DMRT1 is a dose-sensitive regulator of fetal germ cell proliferation and pluripotency. *Proc Natl Acad Sci U S A* **106**, 22323-22328 (2009).
- 13 Piperek, R. P., Kloc, M., Mizia, P. & Kubiak, J. Z. The Central Role of Cadherins in Gonad Development, Reproduction, and Fertility. *Int J Mol Sci* **21** (2020).
- 14 Li, L. *et al.* Single-Cell RNA-Seq Analysis Maps Development of Human Germline Cells and Gonadal Niche Interactions. *Cell Stem Cell* **20**, 891-892 (2017).
- 15 Lee, J. *et al.* Heritable imprinting defect caused by epigenetic abnormalities in mouse spermatogonial stem cells. *Biol Reprod* **80**, 518-527 (2009).
- 16 Pritchard, C., Coil, D., Hawley, S., Hsu, L. & Nelson, P. S. The contributions of normal variation and genetic background to mammalian gene expression. *Genome Biol* **7**, R26 (2006).
- 17 Lahnemann, D. *et al.* Eleven grand challenges in single-cell data science. *Genome Biol* **21**, 31 (2020).
- 18 Chen, W. *et al.* A multicenter study benchmarking single-cell RNA sequencing technologies using reference samples. *Nat Biotechnol* (2020).
- 19 Luecken, M. D. & Theis, F. J. Current best practices in single-cell RNA-seq analysis: a tutorial. *Mol Syst Biol* **15**, e8746 (2019).
- 20 Liu, B. *et al.* An entropy-based metric for assessing the purity of single cell populations. *Nat Commun* **11**, 3155 (2020).

- 21 Garcia, T. X. & Hofmann, M. C. NOTCH signaling in Sertoli cells regulates gonocyte fate. *Cell Cycle* **12**, 2538-2545 (2013).
- 22 Garcia, T. X., DeFalco, T., Capel, B. & Hofmann, M. C. Constitutive activation of NOTCH1 signaling in Sertoli cells causes gonocyte exit from quiescence. *Dev Biol* **377**, 188-201 (2013).
- 23 Saldivar, J. C., Cortez, D. & Cimprich, K. A. The essential kinase ATR: ensuring faithful duplication of a challenging genome. *Nat Rev Mol Cell Biol* **18**, 622-636 (2017).
- 24 Saldivar, J. C. *et al.* An intrinsic S/G2 checkpoint enforced by ATR. *Science* **361**, 806-810 (2018).
- 25 Guo, J. *et al.* The Dynamic Transcriptional Cell Atlas of Testis Development during Human Puberty. *Cell Stem Cell* (2020).
- 26 McCarrey, J. R. Toward a more precise and informative nomenclature describing fetal and neonatal male germ cells in rodents. *Biol Reprod* **89**, 47 (2013).
- 27 Culty, M. Gonocytes, the forgotten cells of the germ cell lineage. *Birth Defects Res C Embryo Today* **87**, 1-26 (2009).
- 28 Li, L. *et al.* Single-Cell RNA-Seq Analysis Maps Development of Human Germline Cells and Gonadal Niche Interactions. *Cell Stem Cell* **20**, 858-873.e854 (2017).
- 29 Guo, F. *et al.* The Transcriptome and DNA Methylome Landscapes of Human Primordial Germ Cells. *Cell* **161**, 1437-1452 (2015).
- 30 Law, N. C., Oatley, M. J. & Oatley, J. M. Developmental kinetics and transcriptome dynamics of stem cell specification in the spermatogenic lineage. *Nat Commun* **10**, 2787 (2019).
- 31 Kanatsu-Shinohara, M. *et al.* Long-term proliferation in culture and germline transmission of mouse male germline stem cells. *Biol Reprod* **69**, 612-616 (2003).
- 32 Sato, T. *et al.* In vitro production of fertile sperm from murine spermatogonial stem cell lines. *Nat Commun* **2**, 472 (2011).

Reviewers' Comments:

Reviewer #1:

Remarks to the Author:

Comments for Author

According to the revised manuscript and response files, the author has provided detailed answers to our proposed amendments, and added a lot of data analysis and experimental content, which is more conducive to understanding the development trajectory of male germ cells. Additionally, the author re-analyzed the dynamic development and cellular events during the transition from SPG to SPC and further to RS according to our suggestions. This part of the analysis allows readers to fully understand the transcriptome changes throughout the entire developmental process of mouse male germ cells. This work is of great significance for understanding the development process of male germ cells in detail. Compared with other work, this article focuses on the analysis of gene expression patterns in the transition process between different developmental stages, which is highly innovative. Moreover, according to the homology analysis of gene expression in human and mouse spermatogenesis, many conserved genes were obtained, and 435 genes related to male infertility and testicular germ cell tumors were screened out. The screening of these homologous genes helps in-depth understanding of the pathogenesis of human male infertility and testicular germ cell-related diseases.

In order to verify the accuracy of the transcriptome data, a variety of molecular biology methods were used to verify the work, and the construction of a knockout mouse model made it clear that the HELQ protein is essential for the cell fate transition from mitosis to PGC arrest. Furthermore, cell experiments verified that Notch signaling pathway regulates the mitotic arrest of male PGCs. The verification of these new results can already support its biological functions. There are two minor comments on data analysis (the last part), I hope the author will explain a little bit in the discussion. In the revised manuscript methodological description, the method steps and details are described in more detail.

Minor comments:

For the second question that "Can the post-arrest PGCs, Q-ProSPG, and T ProSPG mentioned in your article be reproduced based on the integration of the data from your research and them?"

Q: In Rebuttal Fig. R3, we can't see any post-arrest PGCs, Q-ProSPG, and T ProSPG, which you find in your article in paper (PMID:30146481), hope to explain it in the discussion.

In Figure 3e, the authors found that there were thirteen modules of DEGs, while the range of the ordinate is quite small, from -0.4 to 0.4. Was the trend significant? Did the three categories really make sense?

Q: We know you scaled TPM to $\log(\text{TPM}/10+1)$ of each gene among these cells and then scaled $\log(\text{TPM}/10+1)$ to z-score. You did too much calculation on expression levels, and the gene trend may not be the original expression trend. Meanwhile, there were spelling errors in Rebuttal Fig. R5.

Reviewer #2:

Remarks to the Author:

The revised version is much improved and has addressed the questions I raised earlier. As such, the manuscript can be released for publication.

Reviewer #3:

Remarks to the Author:

In this revised manuscript, the authors addressed some of my concerns, however, there are still couple that are not addressed well.

1) The evidence supporting "transitional PGCs" as a newly identified cell subset is not convincing. a) As shown in Fig. 3a, the defined "transitional PGCs" clearly contain some cells similar with mitotic PGCs (~80%) and other cells with mitotic arrest PGCs (~20%). That said, this transitional PGCs cells are a mixture that take part of mitotic PGCs and part of mitotic arrest PGCs. This is well supported by FIG. 3C, where the left part of the defined subcluster is close to mitotic PGCs, while the right part is close to mitotic arrest PGCs. b) If the authors further sub-cluster this defined

"transitional PGCs" based on cell cycle status, they will definitely increase their ROGUE score as well, thus this is not a clear evidence. c) With regard to the immunostaining result, it cannot support that WEE1+Ki67+ cells are transitional PGCs as well. This is because: i) whether Wee1 RNA level is specific for this subset+mitotic arrest PGCs is unclear; ii) whether WEE1 protein (or signal defined by the WEE1 Ab) has the same expression pattern with their RNA. If the authors do want to confirm this transitional PGC is a new cell subset, they should perform RNA-scope (identified marker genes) +IF (cell-cycle markers, such as EdU) to confirm that transitional PGCs share transcriptome with both mitotic PGCs and mitotic arrest PGCs. Otherwise, the authors should omit this statement.

2) While the authors provide some evidence that fetal germ cells have active Notch signaling, whether germ cell-Notch pathway plays the role in germ cell development is unclear, as the evidence is from the treatment with two global Notch inhibitors. This cannot exclude the possibility that Sertoli cell-Notch signaling pathway plays the role in fetal germ cell development, as reported previously (PMID: 23907117, PMID: 23391689), the authors should at least discuss about this in the MS.

3) With regard to the naming of germ cells, while the authors argued there are two naming system, this is not the reason they generated new ones here, which makes the field more confusing. Anyway, they should add the table (Rebuttal Fig. R21) to their MS, in case readers will understand the study better.

Point-by-point responses to the Reviewers' questions:

Below we have listed the point-by-point responses to each of the questions, with the answers highlighted in blue.

Reviewer #1 (Remarks to the Author):

Comments for Author

According to the revised manuscript and response files, the author has provided detailed answers to our proposed amendments, and added a lot of data analysis and experimental content, which is more conducive to understanding the development trajectory of male germ cells. Additionally, the author re-analyzed the dynamic development and cellular events during the transition from SPG to SPC and further to RS according to our suggestions. This part of the analysis allows readers to fully understand the transcriptome changes throughout the entire developmental process of mouse male germ cells. This work is of great significance for understanding the development process of male germ cells in detail. Compared with other work, this article focuses on the analysis of gene expression patterns in the transition process between different developmental stages, which is highly innovative. Moreover, according to the homology analysis of gene expression in human and mouse spermatogenesis, many conserved genes were obtained, and 435 genes related to male infertility and testicular germ cell tumors were screened out. The screening of these homologous genes helps In-depth understanding of the pathogenesis of human male infertility and testicular germ cell-related diseases.

In order to verify the accuracy of the transcriptome data, a variety of molecular biology methods were used to verify the work, and the construction of a knockout mouse model made it clear that the HELQ protein is essential for the cell fate transition from mitosis to PGC arrest. Furthermore, cell experiments verified that Notch signaling pathway regulates the mitotic arrest of male PGCs. The verification of these new results can already support its biological functions. There are two minor comments on data analysis (the last part), I hope the author will explain a little bit in the discussion. In

the revised manuscript methodological description, the method steps and details are described in more detail.

Answer: We thank the reviewer for these positive comments.

Minor comments:

For the second question that “Can the post-arrest PGCs, Q-ProSPG, and T ProSPG mentioned in your article be reproduced based on the integration of the data from your research and them?”

Q: In Rebuttal Fig. R3, we can't see any post-arrest PGCs, Q-ProSPG, and T ProSPG, which you find in your article in paper (PMID:30146481), hope to explain it in the discussion.

Answer: Thank the reviewer for this question. According to the suggestion of the reviewer in last review comments, we have validated the robustness of our data by integrated analysis with these two publicly available scRNA-seq data of mouse germ cells^{1,2}. We found that early-stage cell types ranged from T-ProSPG to Diff.ing SPG could be detected in a previous study (PMID: 30404016), but not the other dataset (PMID: 30146481). Late-stage cell types including the spermatogonia or other spermatogenic cells in these two studies could indeed merge well with our cells along the developmental trajectory. It is most likely that the lack of post-arrest PGCs, Q-ProSPG and T ProSPG in paper (PMID:30146481) is due to the testes for sequencing were obtained from adult mice (7 to 18 weeks old), in which the early-stage germ cells (such as post-arrest PGCs and Q-ProSPG at perinatal stages) were not covered. Accordingly, we have added corresponding explanations in the Discussion section of the revised Manuscript.

In Figure 3e, the authors found that there were thirteen modules of DEGs, while the range of the ordinate is quite small, from -0.4 to 0.4. Was the trend significant? Did the three categories really make sense?

Q: We know you scaled TPM to $\log(\text{TPM}/10+1)$ of each gene among these cells and then scaled $\log(\text{TPM}/10+1)$ to z-score. You did too much calculation on expression

levels, and the gene trend may not be the original expression trend. Meanwhile, there were spelling errors in Rebuttal Fig. R5.

Answer: Thanks for the question. According to the routine scRNA-seq analysis, we performed pre-processing (quality control, normalization, feature selection, scaling, and dimensionality reduction) and cell- and gene-level downstream analysis in our data³. In the Seurat pipeline⁴, gene scaling is always generally applied. In Fig. 3e, gene counts were scaled to have zero mean and unit variance (z scores); therefore, all genes were weighted equally. To avoid the similar concern that might be raised by the readers, we calculated the statistical significance between two adjacent cell types in each trend. The magnitudes of significance agreed with smooth or sharp changes of the DEGs (Fold change \geq 1.5, p -value $<$ 0.01) in the trends (Rebuttal 2 Fig. R1). In addition, the normalized data (unscaled) was used to show the original expression trend, and we found that it was consistent with the scaled data (Rebuttal 2 Fig. R1). Based on these data, the scaled data could reflect the gene expression patterns of DEG modules.

In addition, the spelling errors in Supplementary Fig. 5d have been corrected in the revised Manuscript.

Rebuttal 2 Fig. R1. Clustering analysis of dynamic gene expression during mitotic to mitotic arrest transition. Taken Module 4 as an example, the genes in this module are expressed stably from mitotic PGCs to transitional PGCs, but decrease significantly from transitional PGCs to post-arrest PGCs. The original expression trend showed by the scaled data is consistent with the unscaled data. NS, not significant, *** p -value $<$ 0.001.

Reviewer #2 (Remarks to the Author):

The revised version is much improved and has addressed the questions I raised earlier.

As such, the manuscript can be released for publication.

Answer: We thank the reviewer for the positive comments.

Reviewer #3 (Remarks to the Author):

In this revised manuscript, the authors addressed some of my concerns, however, there are still couple that are not addressed well.

1) The evidence supporting “transitional PGCs” as a newly identified cell subset is not convincing. a) As shown in Fig. 3a, the defined "transitional PGCs" clearly contain some cells similar with mitotic PGCs (~80%) and other cells with mitotic arrest PGCs (~20%). That said, this transitional PGCs cells are a mixture that take part of mitotic PGCs and part of mitotic arrest PGCs. This is well supported by FiG. 3C, where the left part of the defined subcluster is close to mitotic PGCs, while the right part is close to mitotic arrest PGCs. b) If the authors further sub-cluster this defined "transitional PGCs" based on cell cycle status, they will definitely increase their ROGUE score as well, thus this is not a clear evidence. c) With regard to the immunostaining result, it cannot support that WEE1+Ki67+ cells are transitional PGCs as well. This is because: i) whether Wee1 RNA level is specific for this subset+mitotic arrest PGCs is unclear; ii) whether WEE1 protein (or signal defined by the WEE1 Ab) has the same expression pattern with their RNA. If the authors do want to confirm this transitional PGC is a new cell subset, they should perform RNA-scope (identified marker genes) +IF (cell-cycle markers, such as EdU) to confirm that transitional PGCs share transcriptome with both mitotic PGCs and mitotic arrest PGCs. Otherwise, the authors should omit this statement.

Answer: Thank the reviewer for these questions and suggestions.

Firstly, for the comment “The evidence supporting “transitional PGCs” as a newly identified cell subset is not convincing. a) As shown in Fig. 3a, the defined "transitional PGCs" clearly contain some cells similar with mitotic PGCs (~80%) and other cells with mitotic arrest PGCs (~20%). That said, this transitional PGCs cells are a mixture that take part of mitotic PGCs and part of mitotic arrest PGCs. This is well supported by FiG. 3C, where the left part of the defined subcluster is close to mitotic PGCs, while the right part is close to mitotic arrest PGCs.”, we are sorry for this misunderstanding due to the lack of essential descriptions and more details for cell cluster definition

results. And we have performed more thorough analyses and experiments to provide evidences that the transitional PGC were an independent cell population. The evidence includes: Automatically dimension reduction (Fig. 1c of the revised Manuscript) could firstly cluster the gonadal PGCs at E11.5 to ~E16.5 into two cell clusters. According to previous reports⁵ and cell cycle status by immunostaining of MKI67 (Fig. 2b and Supplementary Fig. 4a of the revised Manuscript), the first one in earlier stages was mitotic PGCs, while the second one in later stages was mitotic arrest PGCs. Cell cycle analysis showed that almost all male mitotic PGCs were indeed actively proliferating, which was similar to the specification and migrating PGCs; but a part of mitotic arrest PGCs expressed a high level of cell cycle-related genes (Fig. 2b of the revised Manuscript), indicating that a subpopulation or an independent cell population existed in mitotic arrest PGCs. Hence, unsupervised hierarchical clustering analysis was performed on mitotic arrest PGCs using cell cycle-related genes (Rebuttal 2 Fig. R2a). Notably, two cell populations that expressed high- or low expression levels of cell cycle-related genes were identified; therefore, the cell population that highly expressed cell cycle-related genes was named transitional PGCs and the cell population that lowly expressed cell cycle-related genes was the ultimate mitotic arrest PGCs (Rebuttal 2 Fig. R2a). In addition, as this panel (Rebuttal 2 Fig. R2a) was important for the description of our analysis logics and clustering results, we have put this panel in the main figures of the revised Manuscript (Fig. 3a of the revised Manuscript). Further analysis showed that the transitional PGCs shared the transcriptome signatures of both mitotic PGCs and mitotic arrest PGCs (Rebuttal 2 Fig. R2c); for instance, both of mitotic PGC's marker gene *Mki67* and mitotic arrest PGC's marker gene *Nanos2* were expressed in the transitional PGCs. DEG analysis further showed that the transitional PGCs exhibited its unique gene expression signature (Rebuttal 2 Fig. R2d, see also Fig. 3b of the revised Manuscript). To confirm that transitional PGCs were an independent cell cluster, unbiased cell clustering was further performed on the mitotic PGCs, transitional PGCs, and mitotic arrest PGCs using dimension reduction and clustering analysis (Rebuttal 2 Fig. R2e and f). We found that these cells could also be automatically sub-divided into three independent cell populations, which was comparable with the previous results by

unsupervised hierarchical clustering analysis. In details, the mitotic PGCs, transitional PGCs and mitotic arrest PGCs corresponded to cluster C1, C2 and C3, respectively (Rebuttal 2 Fig. R2e and f). Coupled with these data and the newly acquired RNAscope and IF results (Rebuttal 2 Fig. R3, discussed below), the transitional PGCs were an independent cell population in the development of mouse male germ cells. By the way, we are sorry for the inaccurate crop of transitional PGCs in the previous Manuscript. In the revised Manuscript, we have re-plotted this panel to show the integral clustering and sub-dividing results clearly without superfluous rectangles (Rebuttal 2 Fig. R2a, see also Fig. 3a of the revised Manuscript).

Collectively, based on thorough bioinformatics analyses and more detailed description of our data, we confirmed that the transitional PGCs were indeed an independent cell population, which was further supported by the RNAscope and IF results (Rebuttal 2 Fig. R3, discussed below).

Rebuttal 2 Fig. R2. The identification of the transitional PGCs. **a**, Heatmap showing cell cycle-related genes based hierarchical clustering result of the mitotic arrest PGCs identified from Fig. 1c. The transitional PGCs are clearly subdivided as an independent cell cluster, see also Rebuttal 2 Fig. R2b. **b**, (Top) Heatmap for expression levels of cell cycle-related genes (rows) at the four stages of mouse male PGC development (columns) ordered by the expression levels in each phase. (Bottom) Heatmap of the top 20 DEGs of each stage (identified among the specification PGCs, migrating PGCs, mitotic PGCs,

and mitotic arrest PGCs) with cells (columns) in the same order. The annotation bars in the first line represent the cell clusters identified in Fig. 1c of the Manuscript by dimension reduction and clustering analysis according to the global gene expression profiles. In details, dark red, light red, orange and yellow bars indicate the specification PGCs, migrating PGCs, mitotic PGCs, and mitotic arrest PGCs, respectively, while the pink annotation bars in the second line represent the transitional PGCs sub-divided from mitotic arrest PGCs by clustering analysis according to the gene sets related to cell cycle, see also Rebuttal 2 Fig. R2b. **c**, Violin plots showing the relative expression levels ($\log(\text{TPM}/10+1)$) of representative genes related to proliferation, pluripotency, and cell cycle arrest. **d**, Heatmap of the top DEGs of each stage (identified among the mitotic PGCs, transitional PGCs, and mitotic arrest PGCs). The color key from blue to red indicates low to high expression levels. **e**, Dimension reduction and automatic clustering of the mitotic PGCs, transitional PGCs, and mitotic arrest PGCs. Cells are colored by the cell types in Rebuttal 2 Fig. R2a and b. **f**, Dimension reduction and automatic clustering of the mitotic PGCs, transitional PGCs, and mitotic arrest PGCs. Cells are colored by the clusters. The mitotic PGCs, transitional PGCs, and mitotic arrest PGCs correspond to cluster C1, C2 and C3, respectively.

Secondly, for the comment “*b) If the authors further sub-cluster this defined "transitional PGCs" based on cell cycle status, they will definitely increase their ROGUE score as well, thus this is not a clear evidence.*”, we have toned down this indirect evidence in the revised Manuscript although the ROGUE-guided analysis results indicated that the transitional PGCs should be separated from the mitotic arrest PGCs as an independent cell population for a better definition of its cellular characteristics to a certain extent.

Lastly, for the comment “*c) With regard to the immunostaining result, it cannot support that $WEE1+Ki67+$ cells are transitional PGCs as well. This is because: i) whether *Wee1* RNA level is specific for this subset+mitotic arrest PGCs is unclear; ii) whether *WEE1* protein (or signal defined by the *WEE1* Ab) has the same expression*”

pattern with their RNA. If the authors do want to confirm this transitional PGC is a new cell subset, they should perform RNA-scope (identified marker genes) +IF (cell-cycle markers, such as EdU) to confirm that transitional PGCs share transcriptome with both mitotic PGCs and mitotic arrest PGCs. Otherwise, the authors should omit this statement.”, we have performed more experiments and added the newly acquired data in the revised Manuscript to support the conclusion that the transitional PGCs were indeed an independent cell population which shared transcriptome signatures of both mitotic PGCs and mitotic arrest PGCs.

As we explained above, the transitional PGCs were an independent cell population according to scRNA-seq results, and *Wee1* was identified to be a marker for transitional PGCs (Rebuttal 2 Fig. R3a), which has been validated by immunostaining using the WEE1 antibody in our previous revised Manuscript. According to the suggestion of this reviewer, RNAscope (identified marker genes) +IF (cell-cycle markers) were performed to prove that the transitional PGCs were a new cell population. We have conducted IF (immunofluorescence) & RNAscope (mRNA expression level could be relatively quantified according to criteria following RNAscope manual, see Method). First, we confirmed that the expression patterns of WEE1 protein and *Wee1* mRNA were generally consistent (Rebuttal 2 Fig. R3c). This result indicated that our previous immunostaining of WEE1 co-stained with MKI67 could be used to distinguish the mitotic PGCs, transitional PGCs and mitotic arrest PGCs. To further confirm that transitional PGCs were a new cell population, IF (DDX4+MKI67) & RNAscope (*in situ* hybridization) (*Wee1* mRNA expression level could be relatively quantified according to the criteria following RNAscope manual, see Method) was then performed. The results showed that germ cells at E14.5 could be mainly divided into three cell types according to the expression of MKI67 and *Wee1* (Rebuttal 2 Fig. R3d and e): MKI67-positive and *Wee1* lowly expressed cells (MKI67⁺ *Wee1*^{Low}), MKI67-positive and *Wee1* highly expressed cells (MKI67⁺ *Wee1*^{High}), and MKI67-negative and *Wee1* lowly expressed cells (MKI67⁻ *Wee1*^{Low}). Notably, the proportions of these three cell types were consistent with the results using WEE1 and MKI67 antibodies (Rebuttal 2 Fig. R3e and f). Hence, based on the newly acquired data and other results in the

manuscript, we confirmed that transitional PGCs were indeed an independent cell population.

Rebuttal 2 Fig. R3. Simultaneous immunofluorescence and RNAscope assay of specific markers for distinguishing sub-populations of PGCs *in situ*. a, Violin plot shows the expression level of *Wee1* in the mitotic PGCs, transitional PGCs and mitotic arrest PGCs, which can act as a marker of transitional PGCs. *P* values (< 0.0001)

indicate the expression level of *Wee1* in the transitional PGCs is significantly higher than that of the mitotic PGCs and mitotic arrest PGCs. **b**, Positive and negative probes of RNAscope in E14.5 mouse male gonadal sections. These results indicate that the staining process and sections passed quality control measures. Scale bar, 10 μm . **c**, (Left) RNAscope of *Wee1* co-stained with immunofluorescence of DDX4 and WEE1 antibodies in E14.5 mouse male gonadal sections. Arrows indicate the dots of the probe signals, each representing one copy of the *Wee1* mRNA. Scale bar, 10 μm . Detailed images of the indicated germ cells are also shown. Scale bar, 5 μm . (Right) The *Wee1* mRNA staining signal dots in random, WEE1^{High} (WEE1 protein high expression, detected by WEE1 antibody), and WEE1^{Low} (WEE1 protein low expression, detected by WEE1 antibody) PGCs are quantitatively analyzed. In the left part, graphed results are presented as scatter dot plot and the mean dot counted per random PGC was 5.35; In the right part, graphed results are presented as box and whiskers and the 25% percentile of the dots in WEE1^{High} (WEE1 protein high expression, detected by WEE1 antibody) PGCs is 5.25. We set 5 as cutoff to determine the “High” or “Low” expression level of *Wee1* mRNA (see Method). WEE1 protein and *Wee1* mRNA have the relatively coincident expression pattern. Images are obtained using ZEISS LSM880 confocal microscope under a C-Apochromat 63 \times /1.20 W korr M27 objective lens with 2 \times scan zoom. **d**, RNAscope of *Wee1* co-stained with immunofluorescence of DDX4 and MKI67 antibodies in E14.5 mouse male gonadal sections. Arrows indicate the dots of the probe signals, each representing one copy of the *Wee1* mRNA. Scale bar, 10 μm . Detailed images of the indicated germ cells are also shown. Scale bar, 5 μm . **e**, Proportions of the MKI67⁺ *Wee1*^{Low}, MKI67⁺ *Wee1*^{High}, MKI67⁻ *Wee1*^{Low}, and MKI67⁻ *Wee1*^{High} sub-groups according to above RNAscope & IF staining results. Images are obtained using ZEISS LSM880 confocal microscope under a C-Apochromat 63 \times /1.20 W korr M27 objective lens with 2 \times scan zoom. Quantitative data of RNAscope and IF result is shown below, 4 experimental repeats, 40 sections, and 457 cells. **f**, Proportions of the MKI67⁺ WEE1^{Low}, MKI67⁺ WEE1^{High}, MKI67⁻ WEE1^{Low}, and MKI67⁻ WEE1^{High} sub-groups according to previous IF staining results (Fig. 5g and j of the revised Manuscript). Images are obtained using ZEISS LSM880 confocal microscope

under a C-Apochromat 63×/1.20 W korr M27 objective lens with 0.6× scan zoom. Quantitative data of IF result is shown below, 4 experimental repeats, 26 sections, and 1470 cells.

2) While the authors provide some evidence that fetal germ cells have active Notch signaling, whether germ cell-Notch pathway plays the role in germ cell development is unclear, as the evidence is from the treatment with two global Notch inhibitors. This cannot exclude the possibility that Sertoli cell-Notch signaling pathway plays the role in fetal germ cell development, as reported previously (PMID: 23907117, PMID: 23391689), the authors should at least discuss about this in the MS.

Answer: Thank the reviewer for this suggestion. We have discussed the roles that Sertoli cell-Notch signaling pathway plays in fetal germ cell development^{6,7} in the Results and Discussion sections of the revised manuscript.

3) With regard to the naming of germ cells, while the authors argued there are two naming system, this is not the reason they generated new ones here, which makes the field more confusing. Anyway, they should add the table (Rebuttal Fig. R21) to their MS, in case readers will understand the study better.

Answer: Thank the reviewer for this suggestion. We have added this table to the revised manuscript following the reviewer's suggestion.

Method

Simultaneous RNAscope and immunofluorescence assay

To visualize the transcription of mRNA and distinguish subtypes of PGCs, the RNAscope probe targeting *Wee1* was designed and synthesized by Advanced Cell Diagnostics company, and the assay of RNAscope and IF were performed by using RNAscope® Multiplex Fluorescent Reagent Kit v2. For each experiment, *POLR2A*, *PPIB*, *UBC* and *HPRT* were used as the positive controls; *dapB* was used as the negative control.

Briefly, after fixing in 10% NBF (Neutral buffer formalin) for 24 h, fresh mouse male gonadal sections (5- μ m-thin) were prepared and then pretreated with hydrogen peroxide solution, target retrieval solution, and stained with primary antibodies overnight at 4°C. Sections were further treated with protease plus and finally hybridized with the RNA probe of target gene for 2 h at 40°C in hybrid furnace, followed with a series of signal amplifications. After RNAscope, sections were stained with secondary antibody 30 min at room temperature. And Nuclei were counterstained with Hoechst for 10 min at room temperature. Images were obtained with ZEISS LSM880 confocal microscope.

The signal dots were visually counted following the RNAscope manual (as outlined at <https://acdbio.com/how-correctly-interpret-your-rnascope%C2%AE-images>), and single dot equals to single mRNA. Robust cutoff for determining the “High expression level” and “Low expression level” of *Wee1* mRNA was set up using the following criteria: (1) The number of dots in each cell is an integer, so the cutoff must also be an integer; (2) The mean dot in random PGCs is 5.35, which means PGCs with more than 5 (less than and nearest to 5.35) are *Wee1* mRNA high expression, otherwise are low expression; (3) The 25% percentile of the dot numbers in WEE1^{High} (WEE1 protein high expression) PGCs is 5.25, representing the lower threshold of expected expression is 5 (less than and nearest to 5.25). Overall, PGCs with less than or equal to 5 dots are defined as *Wee1*^{Low} (*Wee1* mRNA low expression) ones, and PGCs with more than 5

dots are defined as *Wee1*^{High} (*Wee1* mRNA high expression) ones. In all cases, statistics was performed following the above criteria.

Reference

- 1 Hermann, B. P. *et al.* The Mammalian Spermatogenesis Single-Cell Transcriptome, from Spermatogonial Stem Cells to Spermatids. *Cell Rep* **25**, 1650-1667 e1658 (2018).
- 2 Shami, A. N. *et al.* Single-Cell RNA Sequencing of Human, Macaque, and Mouse Testes Uncovers Conserved and Divergent Features of Mammalian Spermatogenesis. *Dev Cell* (2020).
- 3 Luecken, M. D. & Theis, F. J. Current best practices in single-cell RNA-seq analysis: a tutorial. *Mol Syst Biol* **15**, e8746 (2019).
- 4 Butler, A., Hoffman, P., Smibert, P., Papalexi, E. & Satija, R. Integrating single-cell transcriptomic data across different conditions, technologies, and species. *Nat Biotechnol* **36**, 411-420 (2018).
- 5 Ewen, K. A. & Koopman, P. Mouse germ cell development: from specification to sex determination. *Mol Cell Endocrinol* **323**, 76-93 (2010).
- 6 Garcia, T. X. & Hofmann, M. C. NOTCH signaling in Sertoli cells regulates gonocyte fate. *Cell Cycle* **12**, 2538-2545 (2013).
- 7 Garcia, T. X., DeFalco, T., Capel, B. & Hofmann, M. C. Constitutive activation of NOTCH1 signaling in Sertoli cells causes gonocyte exit from quiescence. *Dev Biol* **377**, 188-201 (2013).

Reviewers' Comments:

Reviewer #3:

Remarks to the Author:

The authors addressed most of my concerns in the revised manuscript. However, evidence supporting that the "transitional PGCs" is a cell population is still not convincing, as their used marker, *Wee1*, is not specifically expressed in this subset. Nonetheless, the revised manuscript is much improved and acceptable for publication.